# On Self-Adaptive Perception Loss Function for Sequential Lossy Compression

## Abstract

We consider causal, low-latency, sequential lossy compression, with mean squared error (MSE) as the distortion loss, and a perception loss function (PLF) to enhance the realism of reconstructions. As the main contribution, we propose and analyze a new PLF that considers the joint distribution between the current source frame and the previous reconstructions. We establish the theoretical rate-distortion-perception function for first-order Markov sources and analyze the Gaussian model in detail. From a qualitative perspective, the proposed loss can simultaneously avoid the error-permanence phenomenon and also better exploit the temporal correlation between high-quality reconstructions. The proposed loss is referred to as self-adaptive perception loss function (PLF-SA), as its behavior adapts to the quality of reconstructed frames. We provide a detailed comparison of the proposed perception loss function with previous approaches through both information theoretic analysis as well as experiments involving moving MNIST and UVG datasets.

## 1 Introduction

In recent years, the topic of lossy compression for videos has received significant attention, driven by the growing demand for producing visually appealing reconstructions even at lower bitrates. Early versions of compression algorithms relied on distortion measures, e.g., mean squared error (MSE), MS-SSIM (Golinski et al., 2020; Rippel et al., 2021; Li et al., 2021) and PSNR (Agustsson et al., 2020a; Yang et al., 2020; Rippel et al., 2021; Li et al., 2021). However, these metrics often resulted in outputs that were perceived as blurry and lacking *realism*. Consequently, there have been efforts to incorporate *perception*-based loss functions into compression systems to improve visual quality. These loss functions aim to quantify the divergence between the distributions of the source and the reconstruction, where achieving *perfect* perceptual quality means that the two distributions match with each other. Blau & Michaeli (2019) explored the rate-distortion-perception (RDP) tradeoff from a theoretical perspective. Subsequently, Zhang et al. (2021) introduced universal representations, wherein the representation remains fixed during encoding, and only the decoder can be adjusted to attain near-optimal performance.

With the multitude of frames in a video, there is no unique perception loss function (PLF) that is suitable in all cases. In fact, at least two different PLFs have been proposed in prior work. One choice is a PLF based on the framewise marginal distributions (FMD) of the source and reconstruction (Mentzer et al., 2022), where the perception loss function only preserves the marginal distribution of the reconstructed frames instead of the joint distribution. In contrast to this choice, other works such as Veerabadran et al. (2021) have proposed PLF based on the entire joint distribution (JD) of source frames. A recent study, Salehkalaibar et al. (2023), establishes the rate-distortion-perception (RDP) trade-off for both losses. It is shown that at low bitrates, PLF-JD encounters *error permanence phenomenon*, where errors propagate across all future reconstructions, leaving distortion unchanged across frames. On the other hand, at higher bit rates PLF-JD is more desirable, as PLF-FMD does not address temporal consistency between frames.

In this work, we study causal, low-latency, sequential video compression when the output is subject to both a mean squared error (MSE) distortion loss and a new perception loss function, which we refer to as *Self-Adaptive (SA)*. Our proposed loss function (PLF-SA) considers the joint distribution between the current

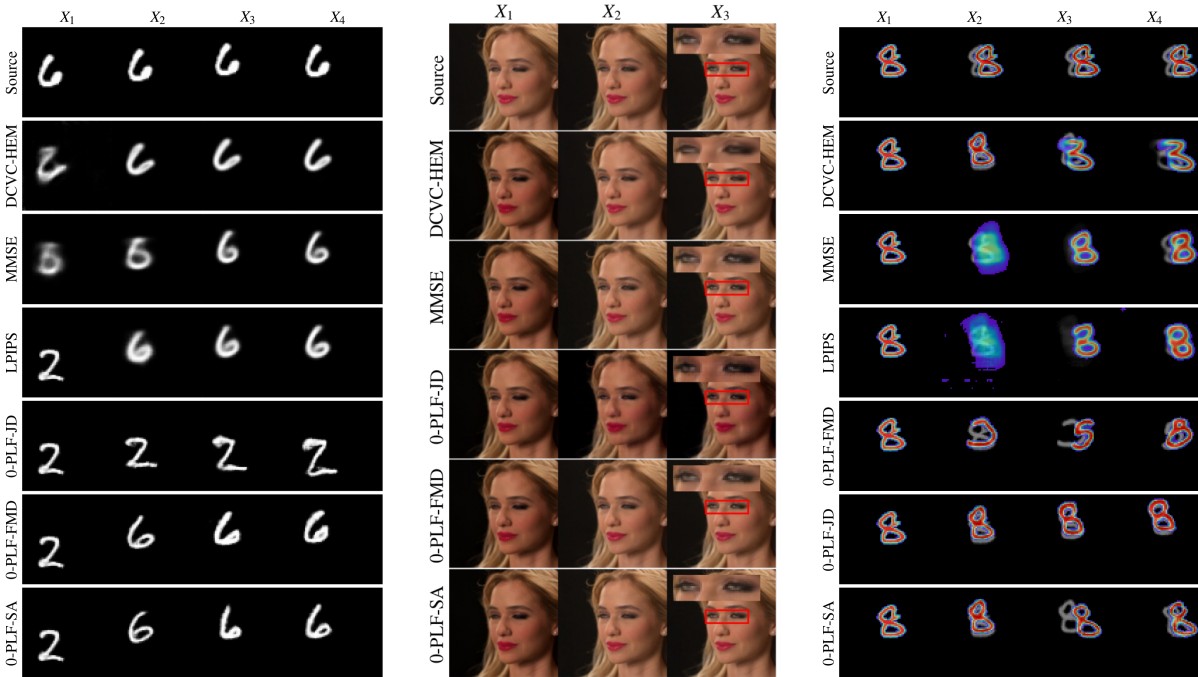

(a)       (b)       (c)

Figure 1: **(a)** Outputs for MovingMNIST with the first frame compressed at a low bitrate $R_1 = 12$ bits. PLF-SA and PLF-FMD recover from previous errors, while PLF-JD and DCVC-HEM exhibit error permanence. **(b)** Outputs for UVG with the first frame compressed at a low bitrate $R_1 = 0.144$ bpp. PLF-SA and PLF-FMD maintain color tone, whereas PLF-JD propagates color tone errors. DCVC-HEM struggles to reconstruct details like eye pupils, while PLF models perform better. **(c)** Outputs for MovingMNIST with the first frame compressed at a high bitrate $R_1 = \infty$ bits. PLF-FMD produces reconstruction error without maintaining the temporal correlation. PLF-JD propagates the trajectory error while PLF-SA rectifies the error preserves the temporal correlation across different frames.

source frame and the previous reconstructions. We establish the rate-distortion-perception function for first-order Markov sources and analyze the Gaussian source in detail for our proposed PLF. We also present experimental results involving moving MNIST and UVG datasets. Our key observation is that our proposed PLF mitigates the disadvantages of previously proposed loss functions: 1) when the previous reconstructions are of lower quality our proposed PLF does not suffer from the error permanence phenomenon observed with PLF-JD; 2) when the previous reconstructions are of higher quality our proposed PLF preserves the joint distribution with these frames and yields better temporal consistency than PLF-FMD. We summarize these below:

- *Resilience to error permanence phenomenon*: Using theoretical analysis of the rate-distortion-perception function of first-order Gauss-Markov sources and through experiments (see, e.g., Fig. 1a and Fig. 1b), we demonstrate that PLF-SA does not suffer from the error permanence phenomenon. In particular, when the first source frame is compressed at a low bitrate, PLF-JD fails to correct mistakes appearing in this frame in subsequent reconstructions. PLF-SA does not suffer from this effect.

- *Sensitivity to temporal correlation across frames*: Through both theoretical analysis and experimental findings (see Fig. 1c), we demonstrate that our proposed PLF-SA can better exploit temporal correlation across frames to yield improved reconstruction. In this setting we assume that the first frame is compressed at a higher bitrate while the second frame is compressed at a lower bitrate. We note that while PLF-FMD yields incorrect output in the second frame, PLF-SA is able to exploit the temporal correlation with the first frame to output the correct digit in the second frame. We also note that PLF-JD still suffers from error permanence as observed in the incorrect trajectory in the reconstruction of the third frame in Fig. 1c; PLF-SA also does not suffer from this effect.

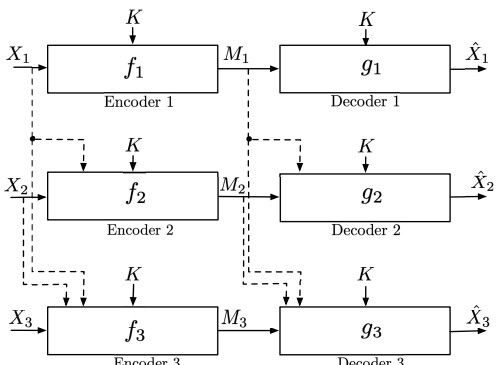

Figure 2: System model for a sequential lossy compression.

PLF-SA does not suffer from the error permanence phenomenon at low bitrates and maintains temporal correlation among frames, especially when the first frame undergoes high-rate compression. Consequently, it takes advantage of both loss functions (PLF-FMD or PLF-JD), depending on the operating rate regime. This adaptability to varying rates is the rationale behind naming this PLF as Self-Adaptive.

## 2  System Model

Assume that we have $T$ frames of video denoted by $(X_1, \ldots, X_T) \in \mathcal{X}_1 \times \ldots \times \mathcal{X}_T$ (where $\mathcal{X}_i \subseteq \mathbb{R}^d$) distributed according to joint distribution $P_{X_1 \ldots X_T}$. The encoders and decoders have access to a shared common randomness $K \in \mathcal{K}$. The (possibly stochastic) $j$th encoding function gets the sources $(X_1, \ldots, X_j)$ and the key $K$ and outputs a variable length message $M_j \in \mathcal{M}_j (= \{0,1\}^\star)$, i.e.,

$$f_j \colon \mathcal{X}_1 \times \ldots \times \mathcal{X}_j \times \mathcal{K} \to \mathcal{M}_j, \qquad j = 1, \ldots, T. \tag{1}$$

The $j$th decoding function receives the messages $(M_1, \ldots, M_j)$ and using the key $K$, it outputs a reconstruction $\hat{X}_j \in \mathcal{X}_j$, i.e.,

$$g_j \colon \mathcal{M}_1 \times \mathcal{M}_2 \times \ldots \times \mathcal{M}_j \times \mathcal{K} \to \mathcal{X}_j. \tag{2}$$

The mappings $\{f_j\}_{j=1}^T$ and $\{g_j\}_{j=1}^T$ induce the conditional distribution $P_{\hat{X}_1 \ldots \hat{X}_T | X_1 \ldots X_T}$ for the reconstructed video given the original video. The proposed framework illustrated in Fig. 2 is a *one-shot* setting i.e., a single sample of the source is compressed at a time.

Each frame $j$ must be reconstructed such that its distortion remains below a certain level, where the loss is assumed to be the mean squared error (MSE) function i.e. $d(x_j, \hat{x}_j) = ||x_j - \hat{x}_j||^2$, which is widely used in many applications. From the perceptual perspective, for given probability distributions $P_{\hat{X}_1 \ldots \hat{X}_{j-1} X_j}$ and $P_{\hat{X}_1 \ldots \hat{X}_{j-1} \hat{X}_j}$, let $\phi_j(P_{\hat{X}_1 \ldots \hat{X}_{j-1} X_j}, P_{\hat{X}_1 \ldots \hat{X}_{j-1} \hat{X}_j})$ denote the perception loss function capturing the divergence between them. We call this PLF, *self-adaptive (SA)*. If $\phi_j(P_{\hat{X}_1 \ldots \hat{X}_{j-1} X_j}, P_{\hat{X}_1 \ldots \hat{X}_{j-1} \hat{X}_j}) = 0$ for $j = 1, \ldots, T$, we get

$$P_{\hat{X}_1 \ldots \hat{X}_{j-1} X_j} = P_{\hat{X}_1 \ldots \hat{X}_{j-1} \hat{X}_j}, \qquad j = 1, \ldots, T, \tag{3}$$

which is called *self-adaptive zero-perception loss (0-PLF-SA)*. In the following, we define two other perception loss functions which are extensively used in many works. For given probability distributions $P_{X_1 \ldots X_j}$ and $P_{\hat{X}_1 \ldots \hat{X}_j}$, let $\xi_j(P_{X_1 \ldots X_j}, P_{\hat{X}_1 \ldots \hat{X}_j})$ be called *perception loss function based on joint distribution (PLF-JD)*. Alternatively, the *perception loss function based on framewise marginal distribution (PLF-FMD)* is shown by $\psi_j(P_{X_j}, P_{\hat{X}_j})$. Notice that 0-PLF-JD and 0-PLF-FMD imply that $P_{X_1 \ldots X_j} = P_{\hat{X}_1 \ldots \hat{X}_j}$ and $P_{X_j} = P_{\hat{X}_j}$ for $j = 1, \ldots, T$, respectively. In most of the paper, for simplicity of presentation, we provide some of our results for $T = 3$ frames. In that case, we use the shorthand notation $\mathsf{X}$ to denote the tuple $(X_1, X_2, X_3)$, e.g., $\mathsf{M} := (M_1, M_2, M_3)$, $\mathsf{D} := (D_1, D_2, D_3)$, $\mathsf{P} := (P_1, P_2, P_3)$, $\mathsf{f} := (f_1, f_2, f_3)$.

## 3 RDP Regions

In this section, we introduce several RDP regions, beginning with the operational RDP region, which is defined as follows.

**Definition 3.1 (Operational RDP region)** *An RDP tuple* $(\mathsf{R}, \mathsf{D}, \mathsf{P})$ *is said to be achievable for the one-shot setting if there exist encoders and decoders such that:*

$$\mathbb{E}[\ell(M_j)] \leq R_j, \tag{4}$$

$$\mathbb{E}[\|X_j - \hat{X}_j\|^2] \leq D_j, \tag{5}$$

$$\phi_j(P_{\hat{X}_1 \ldots \hat{X}_{j-1} X_j}, P_{\hat{X}_1 \ldots \hat{X}_{j-1} \hat{X}_j}) \leq P_j, \quad j = 1, 2, 3, \tag{6}$$

*where* $\ell(M_j)$ *denotes the length of the message* $M_j$*. The operational RDP region, denoted by* $\overline{RDP}^o$*, is the closure of the set of all achievable tuples. Moreover, for a given* $(\mathsf{D}, \mathsf{P})$*, the operational rate region, denoted by* $\overline{R}^o(\mathsf{D}, \mathsf{P})$*, is the closure of the set of all tuples* $\mathsf{R}$ *such that* $(\mathsf{R}, \mathsf{D}, \mathsf{P}) \in \overline{RDP}^o$*.*

We consider Gauss-Markov sources as follows. We assume that $X_1 \sim \mathcal{N}(0, \sigma^2)$ for some $\sigma^2 > 0$,

$$X_2 = \rho X_1 + N_1, \qquad X_3 = \rho X_2 + N_2, \tag{7}$$

for some $0 \leq \rho \leq 1$, where $N_j$ is Gaussian random variable independent of $X_j$ with mean zero and variance $(1 - \rho^2)\sigma^2$ for $j = 1, 2$. The model extends naturally to the case of $T$ time-steps. We assume that the perception loss is Wasserstein-2 distance, i.e.,

$$\phi_j(P_{\hat{X}_1 \ldots \hat{X}_{j-1} X_j}, P_{\hat{X}_1 \ldots \hat{X}_{j-1} \hat{X}_j}) := W_2^2(P_{\hat{X}_1 \ldots \hat{X}_{j-1} X_j}, P_{\hat{X}_1 \ldots \hat{X}_{j-1} \hat{X}_j}). \tag{8}$$

Next, we provide an approximation for the operational RDP region and then analyze it for Gauss-Markov source model. In general, it is not feasible to compute the region $\overline{RDP}^o$ directly since it involves searching over all possible encoding-decoding functions. But, for first-order Markov sources where the Markov chain $X_1 \to X_2 \to X_3$ holds, the following region can be used as an approximation.

**Definition 3.2 (Information RDP Region)** *For first-order Markov sources, let the information RDP region, denoted by* $\overline{RDP}$*, be the set of all tuples* $(\mathsf{R}, \mathsf{D}, \mathsf{P})$ *which satisfy the following*

$$R_1 \geq I(X_1; X_{r,1}), \tag{9}$$

$$R_2 \geq I(X_2; X_{r,2}|X_{r,1}), \tag{10}$$

$$R_3 \geq I(X_3; X_{r,3}|X_{r,1}, X_{r,2}) \tag{11}$$

$$D_j \geq \mathbb{E}[\|X_j - \hat{X}_j\|^2], \tag{12}$$

$$P_j \geq \phi_j(P_{\hat{X}_1 \ldots \hat{X}_{j-1} X_j}, P_{\hat{X}_1 \ldots \hat{X}_{j-1} \hat{X}_j}), \qquad j = 1, 2, 3, \tag{13}$$

*for auxiliary random variables* $(X_{r,1}, X_{r,2}, X_{r,3})$ *and* $(\hat{X}_1, \hat{X}_2, \hat{X}_3)$ *satisfying the following*

$$\hat{X}_1 = \eta_1(X_{r,1}), \quad \hat{X}_2 = \eta_2(X_{r,1}, X_{r,2}), \quad \hat{X}_3 = X_{r,3}, \tag{14}$$

$$X_{r,1} \to X_1 \to (X_2, X_3), \tag{15}$$

$$X_{r,2} \to (X_2, X_{r,1}) \to (X_1, X_3), \tag{16}$$

$$X_{r,3} \to (X_3, X_{r,1}, X_{r,2}) \to (X_1, X_2), \tag{17}$$

*for some deterministic functions* $\eta_1(.)$ *and* $\eta_2(.,.)$*. Moreover, for a given* $(\mathsf{D}, \mathsf{P})$*, the compression rate region, denoted by* $\overline{R}(\mathsf{D}, \mathsf{P})$*, is the closure of the set of all tuples* $\mathsf{R}$ *that* $(\mathsf{R}, \mathsf{D}, \mathsf{P}) \in \overline{RDP}$*. Similarly, for a given* $(\mathsf{R}, \mathsf{P})$*, the information distortion region, denoted by* $\overline{D}(\mathsf{R}, \mathsf{P})$*, is the closure of the set of all tuples* $\mathsf{D}$ *that* $(\mathsf{R}, \mathsf{D}, \mathsf{P}) \in \overline{RDP}$*.*

The following theorem provides outer and inner bounds on the operational RDP region.

**Theorem 3.3** *For first-order Markov sources, a given* $(\mathsf{D}, \mathsf{P})$ *and* $\mathsf{R} \in \overline{R}(\mathsf{D}, \mathsf{P})$*, we have*

$$\mathsf{R} + \log(\mathsf{R} + 1) + 5 \in \overline{R}^o(\mathsf{D}, \mathsf{P}) \subseteq \overline{R}(\mathsf{D}, \mathsf{P}). \tag{18}$$

*Proof:* This statement can be proved using similar lines to the proof of Theorem 3 in Salehkalaibar et al. (2023) which was originally proposed for PLF-JD and PLF-FMD. The proof for PLF-SA is provided in Appendix A for completeness. ∎

Thus, for sufficiently large rates, we can approximate $\overline{R}^o(\mathsf{D}, \mathsf{P})$ by $\overline{R}(\mathsf{D}, \mathsf{P})$.

The above theorem shows that the operational RDP region can be approximated by the information-theoretic RDP region at certain rates. The proof of the inner bound in Theorem 3.3, given in Appendix A, also provides an operational interpretation of the auxiliary random variables $\mathsf{X}_r = (X_{r,1}, X_{r,2}, X_{r,3})$ defined in the iRDP region in Definition 3.2. In particular, each $X_{r,j}$ represents a lossy version of the source sample $X_j$ generated by the encoder at step $j$. It is compressed and transmitted to the decoder at rate $R_j$ as specified in equation 9–equation 11. We refer to $\mathsf{X}_r$ as the *encoded representation* of the source $\mathsf{X}$. The Markov chains in equation 15–equation 17 indicate that without loss of optimality, $X_{r,j}$ can be computed from the source $X_j$ and past representations $X_{r,1}, \ldots, X_{r,j-1}$ without requiring access to the past source samples $X_1, \ldots, X_{j-1}$.

Building on this interpretation, we consider how the reliability of the first frame's encoded representation influences the 0-PLF-SA behavior. If $X_{r,1}$ is unreliable, meaning it is almost independent of both $X_1$ and $X_2$, then 0-PLF-SA imposes independence between $\hat{X}_1$ and $\hat{X}_2$. In this case, 0-PLF-SA effectively reduces to 0-PLF-FMD, discarding unreliable prior information. In contrast, if $X_{r,1}$ is a reliable encoded representation, i.e., $X_{r,1} = X_1$, then 0-PLF-SA simplifies to 0-PLF-JD, leveraging accurate past information in future reconstructions. These observations suggest that the information RDP region for 0-PLF-SA naturally prioritizes retaining the most reliable components of previous reconstructions while disregarding unreliable ones. Since rate allocation is central to video system design, employing a perception loss that adapts effectively to both low-rate and high-rate regimes becomes particularly advantageous.

Next, we analyze the region for the Gauss–Markov model. Specifically, we show that one can restrict to jointly Gaussian distributions over reconstructions and sources without loss of optimality.

**Theorem 3.4** *For the Gauss-Markov source model, any tuple* $(\mathsf{R}, \mathsf{D}, \mathsf{P}) \in \overline{RDP}$ *can be achieved by a jointly Gaussian distribution over* $(X_{r,1}, X_{r,2}, X_{r,3})$ *and identity functions for* $\eta_j(\cdot)$*. That is, for the Gauss-Markov source model, the* $\overline{RDP}$ *region of Definition 3.2 simplifies to the set of all* $(\mathsf{R}, \mathsf{D}, \mathsf{P})$ *tuples such that*

$$R_1 \geq I(X_1; \hat{X}_1), \tag{19a}$$
$$R_2 \geq I(X_2; \hat{X}_2 | \hat{X}_1), \tag{19b}$$
$$R_3 \geq I(X_3; \hat{X}_3 | \hat{X}_1, \hat{X}_2) \tag{19c}$$
$$D_j \geq \mathbb{E}[\|X_j - \hat{X}_j\|^2], \tag{19d}$$
$$P_j \geq \phi_j(P_{\hat{X}_1 \ldots \hat{X}_{j-1} X_j}, P_{\hat{X}_1 \ldots \hat{X}_{j-1} \hat{X}_j}), \qquad j = 1, 2, 3, \tag{19e}$$

*for some auxiliary random variables* $(\hat{X}_1, \hat{X}_2, \hat{X}_3)$ *which satisfy the following Markov chains*

$$\hat{X}_1 \to X_1 \to (X_2, X_3), \tag{20}$$
$$\hat{X}_2 \to (X_2, \hat{X}_1) \to (X_1, X_3), \tag{21}$$
$$\hat{X}_3 \to (X_3, \hat{X}_1, \hat{X}_2) \to (X_1, X_2). \tag{22}$$

*Proof:* The proof uses similar lines to the proof of Theorem 4 in Salehkalaibar et al. (2023). It is provided in Appendix B for completeness. ∎

Based on the above RDP region, for a given rate tuple $\mathsf{R} = (R_1, R_2, R_3)$ and 0-perception loss, we define the following distortions in the causal setting.

**Definition 3.5** *For the Gauss-Markov source model, a given rate tuple $(R_1, R_2, R_3)$ and 0-PLF-SA, distortions $D_{j,SA}(R_1, \ldots, R_j)$ for $j = 1, 2, 3$, are defined by the following optimization problems*

$$D_{1,SA}(R_1) = \min_{P_{\hat{X}_1|X_1}} \mathbb{E}[\|\hat{X}_1 - X_1\|^2] \tag{23a}$$

$$s.t.: \quad P_{\hat{X}_1} = P_{X_1}, \tag{23b}$$

$$I(\hat{X}_1; X_1) \le R_1, \tag{23c}$$

$$\hat{X}_1 \to X_1 \to (X_2, X_3), \tag{23d}$$

*where $P_{\hat{X}_1^*|X_1}$ denotes the optimal solution of equation 23, and*

$$D_{2,SA}(R_1, R_2) = \min_{P_{\hat{X}_2|X_2\hat{X}_1^*}} \mathbb{E}[\|\hat{X}_2 - X_2\|^2] \tag{24a}$$

$$s.t.: \quad P_{\hat{X}_2\hat{X}_1^*} = P_{X_2\hat{X}_1^*} \tag{24b}$$

$$I(\hat{X}_2; X_2|\hat{X}_1^*) \le R_2 \tag{24c}$$

$$\hat{X}_2 \to (\hat{X}_1^*, X_2) \to (X_1, X_3), \tag{24d}$$

*where $P_{\hat{X}_2^*|X_2\hat{X}_1^*}$ denotes the optimal solution of equation 24, and*

$$D_{3,SA}(R_1, R_2, R_3) = \min_{P_{\hat{X}_3|X_3\hat{X}_2^*\hat{X}_1^*}} \mathbb{E}[\|\hat{X}_3 - X_3\|^2] \tag{25a}$$

$$s.t.: \quad P_{\hat{X}_3\hat{X}_2^*\hat{X}_1^*} = P_{X_3\hat{X}_2^*\hat{X}_1^*} \tag{25b}$$

$$I(\hat{X}_3; X_3|\hat{X}_2^*, \hat{X}_1^*) \le R_3 \tag{25c}$$

$$\hat{X}_3 \to (\hat{X}_1^*, \hat{X}_2^*, X_3) \to (X_1, X_2). \tag{25d}$$

**Remark 3.6** *According to Theorem 3.4, for the Gauss–Markov source with MSE distortion loss, the optimizations in equation 23–equation 25 can be restricted to jointly Gaussian reconstructions without loss of optimality. In this case, the constraints in equation 23–equation 25 translate into a non-empty compact subset of a finite-dimensional Euclidean space of covariance parameters, and the distortion loss is a continuous function of these parameters. Hence, the minima in equation 23–equation 25 are attained.*

**Corollary 3.7** *For the Gauss-Markov source model, a given rate tuple $\mathsf{R} = (R_1, R_2, R_3)$ and 0-PLF-SA, we have*

$$(D_{1,SA}(R_1), D_{2,SA}(R_1, R_2), D_{3,SA}(R_1, R_2, R_3)) \in \overline{D}(\mathsf{R}, \mathbf{0}). \tag{26}$$

*Proof:* Let $P_{\hat{X}_1^*|X_1}$, $P_{\hat{X}_2^*|X_2,\hat{X}_1^*}$, and $P_{\hat{X}_3^*|X_3,\hat{X}_2^*,\hat{X}_1^*}$ denote the optimal solutions of equation 23–equation 25. Define the joint distribution of reconstructions conditioned on the sources as follows

$$P_{\hat{X}_1^*, \hat{X}_2^*, \hat{X}_3^*|X_1, X_2, X_3} = P_{\hat{X}_1^*|X_1} P_{\hat{X}_2^*|X_2, \hat{X}_1^*} P_{\hat{X}_3^*|X_3, \hat{X}_2^*, \hat{X}_1^*}.$$

By construction and by the constraints in equation 23–equation 25, this joint conditional distribution satisfies the 0-PLF-SA, the rate constraints

$$I(\hat{X}_1^*; X_1) \le R_1, \quad I(\hat{X}_2^*; X_2|\hat{X}_1^*) \le R_2, \quad I(\hat{X}_3^*; X_3|\hat{X}_2^*, \hat{X}_1^*) \le R_3,$$

and the Markov conditions required in the definition of $\overline{D}(\mathsf{R}, \mathbf{0})$. Moreover, the resulting distortions are

$$\mathbb{E}\|\hat{X}_1^* - X_1\|^2 = D_{1,\mathrm{SA}}(R_1), \quad \mathbb{E}\|\hat{X}_2^* - X_2\|^2 = D_{2,\mathrm{SA}}(R_1, R_2), \quad \mathbb{E}\|\hat{X}_3^* - X_3\|^2 = D_{3,\mathrm{SA}}(R_1, R_2, R_3).$$

Hence the distortion triple

$$\left(D_{1,\mathrm{SA}}(R_1), D_{2,\mathrm{SA}}(R_1, R_2), D_{3,\mathrm{SA}}(R_1, R_2, R_3)\right)$$

is induced by a joint conditional distribution that is feasible for $\overline{D}(\mathsf{R}, \mathbf{0})$, which proves the corollary. ∎

Analogous to the above definition and corollary, one can define $D_{j,\mathrm{FMD}}(R_1, \ldots, R_j)$ and $D_{j,\mathrm{JD}}(R_1, \ldots, R_j)$, and study the achievability of the associated information–distortion region by replacing 0-PLF-SA in equation 23–equation 25 with the corresponding 0-PLF-FMD and 0-PLF-JD, respectively.

We next discuss various insights from the analysis of the RDP region in Theorem 3.4. We will often consider asymptotic regimes as follows. When we set the compression rate $R_j = \epsilon$, it will indicate a low-rate regime, i.e., we will assume that $\epsilon > 0$ is a small constant. When we refer to high-rate compression we will assume that $R_j \to \infty$.

## 4 Distortion Analysis for Gauss-Markov Sources and Zero-Perception Loss

In this section, we present practical insights from analyzing the Gauss-Markov source model when we have a zero-perception loss. We study two criteria on different PLFs: resilience to error permanence phenomenon and sensitivity to temporal correlation across frames.

### 4.1 Resilience to Error Permanence Phenomenon

In this section, we analyze the Gauss-Markov model at low rates to investigate resilience of different PLFs to error permanence phenomenon, initially identified in Salehkalaibar et al. (2023). In our analysis, we assume that the first frame is compressed at a low rate, i.e., $R_1 = \epsilon$ for sufficiently small $\epsilon > 0$. The rates of the second and third steps, $R_2$ and $R_3$, can take on any nonnegative values. For simplicity, assume the case of $\rho = 1$, i.e., $X_2 = X_1$ where the error permanence phenomenon can be clearly demonstrated for PLF-JD (the results for other values of $\rho$ please see Appendix C).

We note that reconstructions of the first frame in all cases are identical. Using standard analysis of the rate-distortion-perception function for Gaussian sources, we have that when $R_1 = \epsilon$, the reconstruction is given by: $\hat{X}_1 = \sqrt{2\epsilon \ln 2}X_1 + Z_1$ where $Z_1 \sim \mathcal{N}(0, (1-2\epsilon \ln 2)\sigma^2)$ is independent of $X_1$ and the resulting distortion is given by $D_{1,\mathrm{FMD}}(\epsilon) = D_{1,\mathrm{JD}}(\epsilon) = D_{1,\mathrm{SA}}(\epsilon) = 2(1-\sqrt{2\epsilon \ln 2})\sigma^2$.

For the second step, the achievable reconstructions of different 0-PLFs are shown in Table 1. Most strikingly, we note that regardless of the value of $R_2$, the reconstruction of 0-PLF-JD is of the form $\hat{X}_2 = \hat{X}_1$ when $\rho = 1$. Intuitively once $\hat{X}_1$ is generated and $\rho = 1$, the PLF-JD forces all the future reconstructions to be an identical copy and ignore any new information available to the decoder. This is referred to as the error permanence phenomenon Salehkalaibar et al., 2023. In contrast note that the reconstructions associated with PLF-SA and PLF-FMD are of the form: $\hat{X}_2 = \omega_1 \hat{X}_1 + \omega_2 X_2 + Z_2$, where the coefficients $\omega_1$ and $\omega_2$ are stated in Table 1. In this case the reconstruction $\hat{X}_2$ incorporates new information available to the decoder in the second step as reflected in the coefficient $\omega_2$. Interestingly, as $\epsilon$ approaches 0, the coefficient $\omega_2$ in both cases becomes identical, indicating that both losses capture similar information.

Table 1: Achievable reconstructions and distortions for $R_1 = \epsilon$ and a nonnegative $R_2 \gg \epsilon$.

| | Second Frame: $\hat{X}_2 = \omega_1 \hat{X}_1 + \omega_2 X_2 + Z_2$ | | |
| --- | --- | --- | --- |
| | Coefficients $\omega_1$, $\omega_2$ | Distortion of the second frame | $Z_2$ |
| 0-PLF-FMD | $\omega_1 = \dfrac{\sqrt{2\epsilon \ln 2}}{\sqrt{1-2^{-2R_2}+2\epsilon \ln 2}}$, $\omega_2 = \dfrac{1-2^{-2R_2}}{\sqrt{1-2^{-2R_2}+(2\epsilon \ln 2)}}$ | $2\sigma^2(1 - \sqrt{1 - 2^{-2R_2} + \rho^2 2\epsilon \ln 2}) + O(\epsilon)$ (Appendix C.2) | $\mathcal{N}(0, (1 - \omega_1^2 - \omega_2^2 - 2\omega_1\omega_2\sqrt{2\epsilon \ln 2})\sigma^2)$ |
| 0-PLF-SA | $\omega_1 = \sqrt{2\epsilon \ln 2}(1 - \sqrt{1 - 2^{-2R_2}})$, $\omega_2 = \sqrt{1 - 2^{-2R_2}}$ | $2\sigma^2(1 - \sqrt{1 - 2^{-2R_2}}) + O(\sqrt{\epsilon})$ (Appendix C.1) | $\mathcal{N}(0, (2^{-2R_2} - (1 - \sqrt{1 - 2^{-2R_2}})^2(2\epsilon \ln 2))\sigma^2)$ |
| 0-PLF-JD | $\omega_1 = 1$, $\omega_2 = 0$ | $2\sigma^2(1 - \sqrt{2\epsilon \ln 2}) = D_{1,\mathrm{JD}}(\epsilon)$ (Appendix C.3) | 0 |

Although our discussion above is limited to the case when the compression rate of the first frame is very small, similar conclusions also appear to hold for moderate compression rates. We illustrate this behavior numerically in Fig. 11 in Appendix C. In particular for $R_1 = 0.1$ and $R_2 \geq 0.05$, the distortion of the second

frame for 0-PLF-SA outperforms that of 0-PLF-JD. We also discuss the reconstruction associated with the third frame in the same Appendix. While, by design, PLF-FMD achieves a lower distortion than PLF-SA, it does not always output the most desired reconstructions. As discussed in the next section, PLF-FMD fails to effectively preserve temporal correlation across frames.

## 4.2 Sensitivity to Temporal Correlation Across Frames

In this section, we show that the choice of PLF affects the temporal correlation across different frames. Specifically, we consider the case where the first and thirds frames are compressed at a high rate, i.e., $R_1, R_3 \to \infty$, and the rate of the second frame is small enough, i.e., $R_2 = \epsilon$ for a sufficiently small $\epsilon > 0$. In order to develop a full qualitative understanding, we also consider the case of case of $R_2 = R_3 = \epsilon$. This case is more involved and discussed in Appendix D. In the first step, the high rate assumption implies that $\hat{X}_1 = X_1$. The achievable reconstructions of all 0-PLFs for the second and third steps are discussed in the following and summarized in Table 2.

Table 2: Achievable reconstructions and distortions for $R_1, R_3 \to \infty$ and $R_2 = \epsilon$.

| | Second Step | Third Step |
|---|---|---|
| 0-PLF-FMD ($\sqrt{\epsilon} \ll \rho < 1$) | $\hat{X}_2 = (1 - O(\epsilon))\hat{X}_1 + O(\epsilon)X_2 + Z_{2,\text{FMD}}$ $Z_{2,\text{FMD}} \sim \mathcal{N}(0, O(\epsilon)\sigma^2)$ $D_{2,\text{FMD}}(\infty, \epsilon) = 2(1 - \rho - O(\epsilon))\sigma^2$  Table 2 in Salehkalaibar et al. (2023) | $\hat{X}_3 = X_3$ (Appendix D.3) |
| 0-PLF-FMD ($0 < \rho \ll \sqrt{\epsilon}$) | $\hat{X}_2 = O(\sqrt{\epsilon})X_2 + Z'_{2,\text{FMD}}$ $Z'_{2,\text{FMD}} \sim \mathcal{N}(0, (1 - O(\epsilon))\sigma^2)$ $D_{2,\text{FMD}}(\infty, \epsilon) = 2\sigma^2(1 - O(\sqrt{\epsilon}))$ (Appendix D.3) | $\hat{X}_3 = X_3$ (Appendix D.3) |
| 0-PLF-JD | $\hat{X}_2 = (\rho - O(\sqrt{\epsilon}))\hat{X}_1 + O(\sqrt{\epsilon})X_2 + Z_{2,\text{JD}}$ $Z_{2,\text{JD}} \sim \mathcal{N}(0, (1 - \rho^2 + O(\epsilon))\sigma^2)$ $D_{2,\text{JD}}(\infty, \epsilon) = 2\sigma^2(1 - \rho^2 - O(\sqrt{\epsilon}))$  Table 2 in Salehkalaibar et al. (2023) | $\hat{X}_3 = (\rho - O(\sqrt{\epsilon}))\hat{X}_2 + \frac{1}{\sqrt{1+\rho^2}}(\rho N_1 + N_2) + Z_{3,\text{JD}}$ $Z_{3,\text{JD}} \sim \mathcal{N}(0, O(\sqrt{\epsilon})\sigma^2)$ $D_{3,\text{JD}}(\infty, \epsilon, \infty) = 2\sigma^2 \left(1 - \rho^4\right) \left(1 - \frac{1}{\sqrt{1+\rho^2}}\right) + O(\sqrt{\epsilon})$ (Appendix D.2) |
| 0-PLF-SA | $\hat{X}_2 = (\rho - O(\sqrt{\epsilon}))\hat{X}_1 + O(\sqrt{\epsilon})X_2 + Z_{2,\text{SA}}$ $Z_{2,\text{SA}} = Z_{2,\text{JD}}$ $D_{2,\text{SA}}(\infty, \epsilon) = D_{2,\text{JD}}(\infty, \epsilon)$ (Appendix D.1) | $\hat{X}_3 = X_3$ (Appendix D.1) |

*Achievable Reconstructions of* 0*-PLF-FMD*:

*Large Correlation Coefficient*: As it can be observed from the first row of Table 2, for a sufficiently large correlation coefficient, $\sqrt{\epsilon} \ll \rho < 1$, i.e., the movements between frames are smooth, the reconstruction based on 0-PLF-FMD for the second frame is given by $\hat{X}_2 \approx (1 - O(\epsilon))\hat{X}_1 + O(\epsilon)X_2$, implying that the first frame is *copied* in the future reconstruction. In the third frame we have that $\hat{X}_3 = X_3$ as $R_3 \to \infty$. On the other hand PLF-FMD also exhibits a tendency to copy the first frame when $R_3$ is small, as shown in Appendix D.3. In our experiments we observe that the output of 0-PLF-FMD looks more *static* when compared to the other PLFs.

*Small Correlation Coefficient*: The case when $0 < \rho \ll \sqrt{\epsilon}$, operationally captures the scenario when there are some sharp movements in frames. In this case we have $\hat{X}_2 = O(\sqrt{\epsilon})X_2 + Z'_{2,\text{FMD}}$ where $Z'_{2,\text{FMD}} \sim \mathcal{N}(0, (1 - O(\epsilon))\sigma^2)$ is independent of $X_2$ and $\hat{X}_3 = X_3$. Note that the reconstruction $\hat{X}_2$ largely ignores any correlation with $X_1$, which is undesirable in practice. We will demonstrate that this property of PLF-FMD leads to temporal inconsistency in the reconstructed frames in our experiments.

*Achievable Reconstructions of* 0*-PLF-JD*:

According to the third row of Table 2, the reconstruction of 0-PLF-JD in the second and third frames are given by $\hat{X}_2 = (\rho - O(\sqrt{\epsilon}))\hat{X}_1 + O(\sqrt{\epsilon})X_2 + Z_{2,\text{JD}}$ and $\hat{X}_3 = (\rho - O(\sqrt{\epsilon}))\hat{X}_2 + \frac{1}{\sqrt{1+\rho^2}}(\rho N_1 + N_2) + Z_{3,\text{JD}}$, which mimic the correlation structure of the source model. One weakness of this decoder is that the noise $Z_{2,\text{JD}}$ introduced in the second step continues to propagate in the third step through the term $\rho Z_{2,\text{JD}}$ as seen from $\hat{X}_3 = (\rho^2 - O(\sqrt{\epsilon}))\hat{X}_1 + O(\sqrt{\epsilon})X_2 + \rho Z_{2,\text{JD}} + \frac{1}{\sqrt{1+\rho^2}}(\rho N_1 + N_2) + Z_{3,\text{JD}}$. We will see in our experiements that this can lead to undesirable errors in the reconstruction, indicating error propagation effect.

*Achievable Reconstructions of* 0*-PLF-SA*:

The 0-PLF-SA in the second frame is expressed as $P_{\hat{X}_1 X_2} = P_{\hat{X}_1 \hat{X}_2}$. When combined with the high compression rate for the initial frame (i.e., $R_1 \rightarrow \infty$), it reduces to $P_{X_1 X_2} = P_{\hat{X}_1 \hat{X}_2}$, which is equivalent to the constraint in the 0-PLF-JD framework. Thus, the reconstruction of the second frame for 0-PLF-SA is similar to that of 0-PLF-JD, i.e., $\hat{X}_2 = (\rho - O(\sqrt{\epsilon}))\hat{X}_1 + O(\sqrt{\epsilon})X_2 + Z_{2,\text{SA}}$ where $Z_{2,\text{SA}} = Z_{2,\text{JD}}$. For the third frame, the reconstruction is given by $\hat{X}_3 = X_3$ due to the high rate. Thus, the decoder based PLF-SA differs from PLF-JD in that the reconstruction is not strongly dependent on the noise in the second step. In our experiments, we also demonstrate that PLF-SA indeed has an improved reconstruction over PLF-JD.

## 5 Experimental Results

Our theoretical results for PLF-SA show that PLF-SA is a new perceptual loss that inherits advantages from both PLF-JD and PLF-FMD. In this section we provide experimental results to further demonstrate this effect.

### 5.1 Implementation Details

Expanding upon the experimental framework established in Salehkalaibar et al. (2023), we merge the scale-space-flow neural video coding architecture introduced by Agustsson et al. (2020b) with Wasserstein GANs for perceptual quality enhancement, as proposed in Gulrajani et al. (2017). We employ three datasets: the 1-digit MovingMNIST (Srivastava et al., 2015), KTH (Schuldt et al., 2004) and UVG dataset (Mercat et al., 2020), offering varying levels of video resolution and scene complexity. The MovingMNIST dataset consists of low-complexity synthetic sequences with dimensions of $64 \times 64$. The KTH dataset is for action recognition which contains video clips of six human actions at $120 \times 160$. The UVG dataset comprises high-definition real-life video patches sized at $256 \times 256$. In general, the preference for certain deep learning structures and datasets aims at confirming the suggested theory rather than developing the most advanced neural network architectures.

To evaluate the compression performance of the proposed PLF-SA, we compare it with prior perception loss models, namely PLF-FMD and PLF-JD (Salehkalaibar et al., 2023). We also compare with another baseline, DCVC-HEM (Li et al., 2022), which makes use of MS-SSIM loss during training and its manually designed module for capturing strong temporal correlations through multi-scale features from previously decoded frames. Further experimental details can be found in Appendix E.

### 5.2 Main Results

We first present the results validating the low-rate regime analysis described in Section 4.1. Following that, we provide the complementary results for the high-rate regime analysis discussed in Section 4.2.

#### 5.2.1 Low-rate Case $R_1 = 12$ bits

We first validate the achievable reconstructions and distortions for $R_1 = \epsilon$ discussed in Section 4.1. Fig. 1a and Fig. 3 show samples of 3-frame MovingMNIST sequences where the first frame is encoded at a low bitrate $R_1 = 12$ bits.

As shown in third row of Table 1, given an incorrect reconstruction in $\hat{X}_1$, the decoder with 0-PLF-JD exhibit the error permanence phenomenon for future frame reconstructions, as it tends to replicate the reconstructed first frame as discussed in Table 1. Furthermore, as in the first and second rows of Table 1, the decoders with 0-PLF-FMD and proposed 0-PLF-SA utilize new information from $X_2$ to recover from wrongly predicted $\hat{X}_1$ with $\hat{X}_2 = \omega_1 \hat{X}_1 + \omega_2 X_2 + Z_2$. This highlights their capability to rectify previous mistakes. Results for DCVC-HEM and MMSE-based are also presented. Due to the low bitrate setting, the MMSE reconstructions tend to be blurry. DCVC-HEM also suffers from error propagation with digit "7" wrongly decoded as "3" in Fig. 1a.

Analogous results for UVG dataset are shown in Fig. 1b and Fig. 4. When the first frame is compressed at a low rate $R_1 = 0.144$ bpp, the reconstructed frame $\hat{X}_1$ exhibits a noticeable degradation in overall color tone. For the decoder with 0-PLF-JD, this error propagates to future reconstructions, $\hat{X}_2$ and $\hat{X}_3$. In contrast,

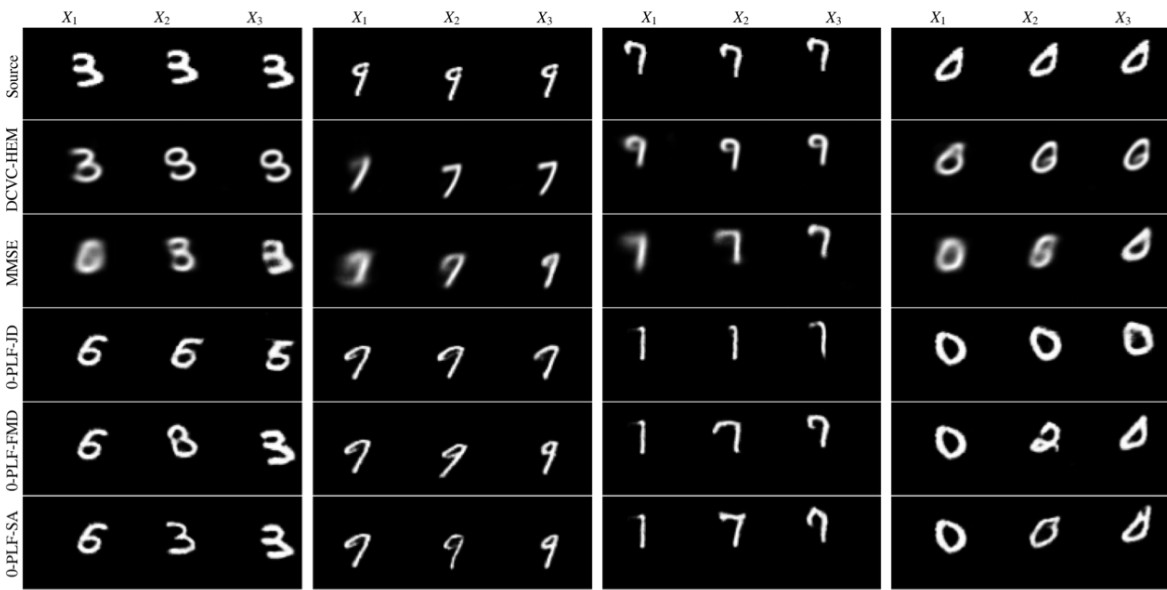

Figure 3: The reconstruction results on the MovingMNIST dataset when the first frame is compressed at a low rate $R_1 = 12$ bits. Similar to the Guass-Markov case presented in Section 4.1, both PLF-SA and PLF-FMD demonstrate resilience to prior errors (digit contour errors) by incorporating new information from $X_2$ and $X_3$, while PLF-JD suffers from error permanence phenomenon as it tends to ignore new information. DCVC-HEM exhibits a comparable tendency for error permanence.

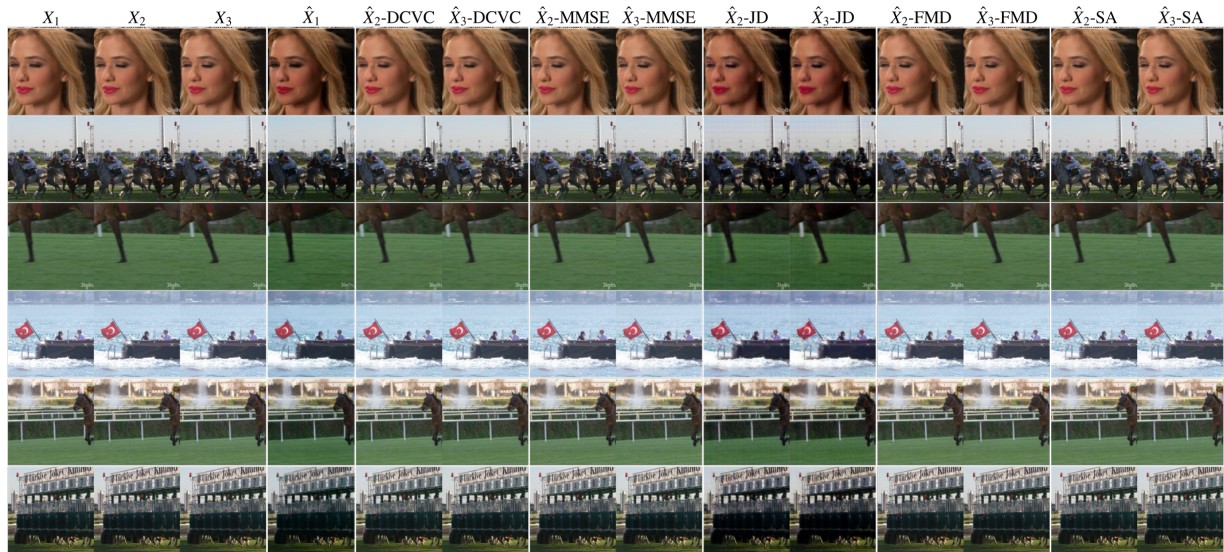

Figure 4: The reconstruction results on the UVG dataset when the first frame is compressed at a low rate $R_1 = 0.144$ bpp. $\hat{X}_1$ is shared across all models. Similar to the Gauss-Markov case and MovingMNIST results, PLF-SA and PLF-FMD exhibit robustness to first-frame errors (color tone mismatches) while PLF-JD suffers from error permanence.

0-PLF-FMD and 0-PLF-SA correct the color tone in the reconstructions of $\hat{X}_2$ and $\hat{X}_3$. Additionally, DCVC-HEM preserves the correct color tone but struggles to reconstruct fine details, such as eye pupils in Fig.1b, where the PLF models demonstrate better performance.

Furthermore, PSNR and perceptual loss comparisons on UVG dataset are presented in Table 3. All models are evaluated across 2000 frames for $\hat{X}_2$ and 2000 frames for $\hat{X}_3$ under the same low-bitrate setting ($R_1 = 0.144$ bpp). As discussed in Section 4.1, PLF-FMD achieves the lowest distortion, with PLF-SA closely following. In contrast, PLF-JD exhibits the worst performance due to the error permanence phenomenon. For perceptual loss, PLF-SA and PLF-FMD exhibit similar performance.

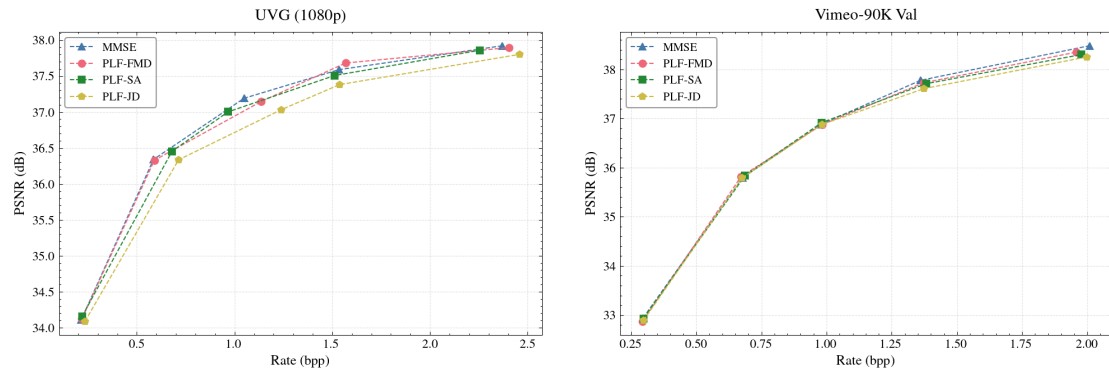

(a)                                                                    (b)

Figure 5: Rate-PSNR curves on the second frame of both the UVG (1080p) and Vimeo-90K ($448 \times 256$p) validation to provide a fair, rate-matched comparison under $R_1$-$\epsilon$ setting. For different PLFs, FMD yields the lowest PSNR, with SA closely matching its behavior, while JD produces the lowest PSNR. These results offer stronger empirical support for our theoretical claims.

Table 3: PSNR and LPIPS comparisons under the low-bitrate setting $R_1 = 0.144$ bpp on the UVG dataset. Among these models, PLF-FMD achieves the lowest distortion across two frames $\hat{X}_2$ and $\hat{X}_3$, with PLF-SA closely following. PLF-JD performs the worse due to error permanence.

|  | PSNR↑ | | LPIPS↓ | |
| --- | --- | --- | --- | --- |
|  | $\hat{X}_2$ | $\hat{X}_3$ | $\hat{X}_2$ | $\hat{X}_3$ |
| DCVC-HEM | 28.11 | 28.74 | 0.039 | 0.028 |
| PLF-JD | 22.38 | 21.99 | 0.049 | 0.053 |
| PLF-SA | 30.59 | 30.67 | 0.0039 | 0.0043 |
| PLF-FMD | 31.02 | 30.72 | 0.0036 | 0.0041 |

### 5.2.2   High-rate Case $R_1 = \infty$ bits

We now validate the achievable reconstructions and distortions for $R_1 = \infty$ ($\hat{X}_1 = X_1$) discussed in Section 4.2. and Table 5 in Appendix D. For MovingMNIST dataset, Fig. 1c and Fig. 6 show results with 3-frame architecture where $R_2 = 2$ bits and $R_3 = 16$ bits represent low and medium rates. Fig. 7 show results with 4-frame architecture where $R_2 = 2$ bits, $R_3 = 12$ bits and $R_4 = 2$ bits. The source digit maintains its motion direction across all frames. We evaluate each model's performance on reconstruction of $\hat{X}_2$ and $\hat{X}_3$ and analyze the digit moving trajectory across frames. Both small and large values of the correlation coefficient $\rho$ correspond to scenarios with high and low video sampling rates, respectively.

**Results on MovingMNIST.**   Fig. 1c, Fig. 6a and Fig. 7 show results for a small correlation coefficient, i.e., $0 < \rho \ll \sqrt{\epsilon}$. Both 0-PLF-SA and 0-PLF-JD fail to identify the correct direction in the second frame, producing identical reconstructions ($\hat{X}_{2,\text{SA}} = \hat{X}_{2,\text{JD}}$) as shown in third and fourth rows of Table 2. By the third frame, 0-PLF-JD exhibits error permanence, propagating second frame noise ($\rho Z_{2,\text{JD}}$) to third frame (see the third row of Table 2), while 0-PLF-SA reduces noise dependence and accurately reconstructs $\hat{X}_3 = X_3$ when $R_3 \to \infty$ (see the fourth row of Table 2). For 0-PLF-FMD, temporal correlation is less effectively preserved. It introduces synthetic noise in the second frame ($\hat{X}_2 = O(\sqrt{\epsilon})X_2 + Z'_{2,\text{FMD}}$) as in the second row of Table 2, decoding correct direction but often changing digit contours. In contrast, 0-PLF-SA balances content preservation and error correction, maintaining the digit's identity and direction under low bitrate conditions. The MMSE model produces blurry $\hat{X}_2$ at low rates but retains correct direction. By the third frame, $\hat{X}_3$ improves with a medium bitrate but lacks fine details. DCVC-HEM, using $\hat{X}_{2,\text{SA}}$ from 0-PLF-SA as input, corrects the direction in $\hat{X}_3$ but struggles with digit contours. In Fig. 7, results with additional 4-th frame are presented to demonstrate consistency with 3-frame setting shown in Fig. 6a, thereby validating the model's ability to effectively reconstruct longer sequences. Numerical results in Table 4 show that in the second frame, 0-PLF-JD and 0-PLF-SA exhibit higher distortion than 0-PLF-FMD due to trajectory errors. By the third frame, 0-PLF-JD propagates these errors to $\hat{X}_3$, whereas 0-PLF-SA corrects them, approaching 0-PLF-FMD's distortion. Additionally, 0-PLF-FMD struggles with temporal correlation in $\hat{X}_2$, resulting in the worst LPIPS score, while 0-PLF-SA achieves the best perceptual quality for both $\hat{X}_2$

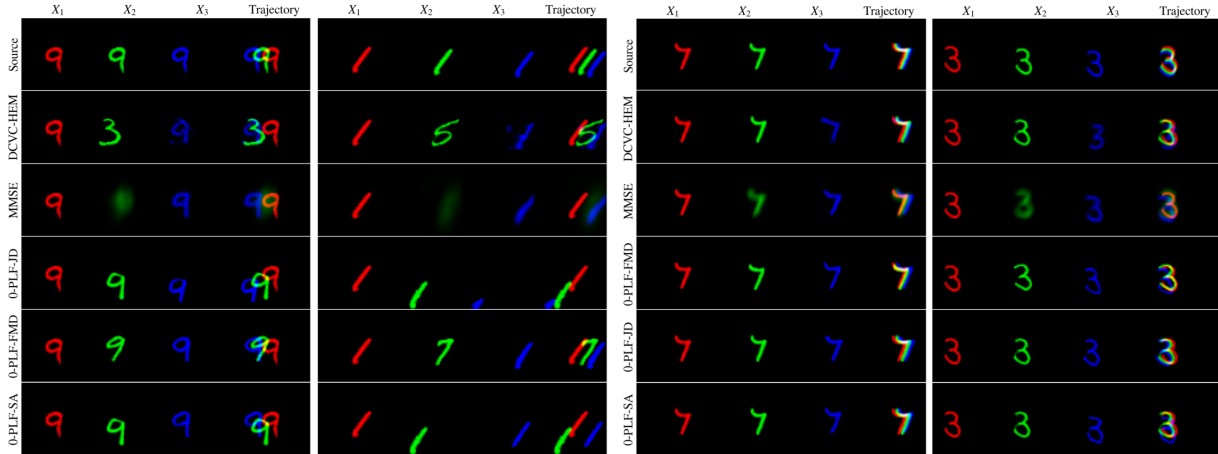

(a) Sharp movement scenario.          (b) Slow movement scenario.

Figure 6: The reconstruction results on the MovingMNIST dataset for $\infty$-$R_2$-$R_3$ with $R_2 = 2$ bits and $R_3 = 16$ bits. Colored digits highlight trajectory across frames. (a) With small correlation coefficient $0 < \rho \ll \sqrt{\epsilon}$, PLF-FMD preserves direction but loses temporal consistency in digits' contour. PLF-JD and PLF-SA fail to identify the direction in the second frame, but PLF-SA rectifies the error in the third frame. (b) With large correlation coefficient $\sqrt{\epsilon} \ll \rho < 1$, PLF-FMD tends to replicate the first frame without capturing motion effectively, while PLF-JD and PLF-SA show greater generative diversity.

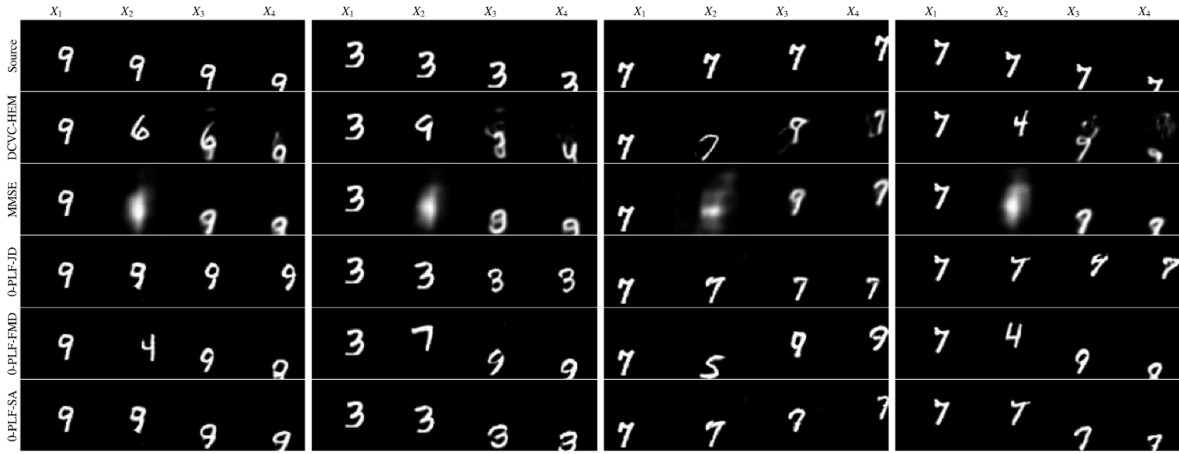

Figure 7: The reconstruction results on the MovingMNIST dataset using the $\infty$-$R_2$-$R_3$-$R_4$ 4-frame architecture, with $R_2 = 2$ bits, $R_3 = 16$ bits, and $R_4 = 16$ bits. Consistent with the results observed in the 3-frame architecture, both PLF-JD and PLF-SA produce direction errors, and PLF-SA is able to eventually correct them. PLF-FMD cannot obviously preserve the temporal correlation as it reconstructs incorrect digits.

and $\hat{X}_3$. These results highlight PLF-SA's robustness to small $\rho$ under low-bitrate settings ($R_2 = 2$ bits, $R_3 = 16$ bits).

Fig. 6b shows results for large correlation coefficient $\sqrt{\epsilon} \ll \rho < 1$. Corresponding to the first row of Table 2, 0-PLF-FMD tends to *copy* the first-frame $\hat{X}_1$ when reconstructing the second frame with $\hat{X}_2 \approx (1 - O(\epsilon))\hat{X}_1 + O(\epsilon)X_2$, resulting in a lack of generative diversity. In contrast, 0-PLF-JD and 0-PLF-SA do not exhibit such "static" reconstruction behavior for the second frame. MMSE and DCVC-HEM perform better compared with small $\rho$ case. However, issues such as blurriness and discrepancies in image details still persist. Overall, PLF-SA demonstrates a superior ability to balance reconstruction distortion and perceptual quality across various bitrate settings.

**Results on KTH.** As shown in Fig. 8, we further evaluate models on the KTH dataset, which contains grayscale video sequences of human actions such as walking and running. For both $\hat{X}_2$ and $\hat{X}_3$, PLF-SA is able to successfully reconstruct both the actions (walking versus running) and directions (left versus right),

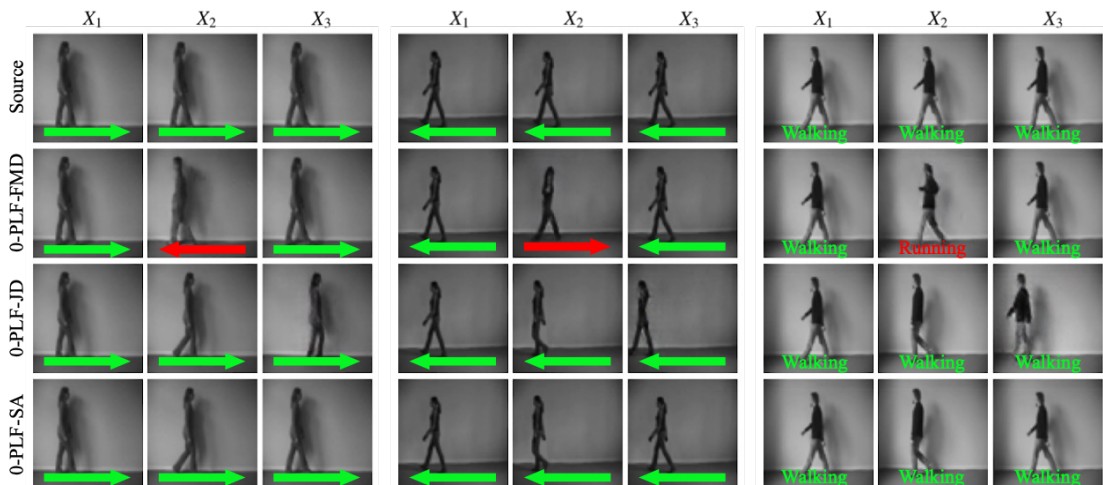

Figure 8: The reconstruction results on the KTH dataset for $\infty$-$R_2$-$R_3$ with $R_2 = 0$ bits and $R_3 = 32$ bits. PLF-SA successfully reconstructs human motions (running/walking) and directions (left/right) and PLF-JD exhibits a subtle bias in estimating the exact position. PLF-FMD also struggles to accurately capture both the motion type and direction.

Table 4: PSNR, LPIPS and FloLPIPS comparisons under the high-bitrate setting $R_1 = \infty$ on MovingMNIST with small $\rho$. PLF-JD and PLF-SA exhibit higher distortion than PLF-FMD due to trajectory errors. PLF-SA achieves the best perceptual quality on $\hat{X}_2$, $\hat{X}_3$, while PLF-FMD struggles with temporal correlation.

|  | PSNR↑ | | LPIPS↓ | | FloLPIPS↓ | |
|---|---|---|---|---|---|---|
|  | $\hat{X}_2$ | $\hat{X}_3$ | $\hat{X}_2$ | $\hat{X}_3$ | $\hat{X}_2$ | $\hat{X}_3$ |
| DCVC-HEM (9.83-86.21) | 14.50 | 20.42 | 0.115 | 0.073 | 0.141 | 0.104 |
| PLF-JD (2-16) | 13.54 | 14.82 | 0.026 | 0.077 | 0.154 | 0.263 |
| PLF-SA (2-16) | 13.54 | 20.47 | 0.026 | 0.021 | 0.126 | 0.061 |
| PLF-FMD (2-16) | 14.74 | 20.73 | 0.114 | 0.024 | 0.188 | 0.105 |

demonstrating its ability to capture temporal consistency under low and moderate rates. Meanwhile, PLF-JD also captures the correct action and direction, but exhibits a slight bias in estimating the precise spatial location of the human in $\hat{X}_3$. In contrast, PLF-FMD fails to reliably preserve the action or the directional consistency in $\hat{X}_2$ due to extremely low rate $R_2 = 0$ bits, generating visually implausible sequences. These results validate the effectiveness of our proposed method in handling realistic motion dynamics across various bitrate budgets.

## 6 Conclusions

We observe that previously proposed perception loss functions (PLF) in video compression can have disadvantages in different operating regimes. In particular, the PLF-JD that preserves the joint distribution of all the frames suffers from the effect of error permanence, where mistakes made in previously reconstructed frames carry over in subsequent frames. On the other hand, the PLF-FMD that only preserves marginal distribution of frames and achieves the lowest distortion, does not effectively exploit the temporal correlation during reconstruction. Motivated by these observations, we propose a new loss, PLF-SA, that mitigates the disadvantages of each. When the previously reconstructed frames are of lower quality, our proposed loss avoids the error permanence phenomenon in PLF-JD. When the previously reconstructed frames are of higher quality, the decoder based on PLF-SA effectively exploits temporal correlation between frames. We validate the merits of our proposed loss through experimental results involving moving-MNIST, KTH, and UVG datasets in a variety of operating regimes. We also provide information theoretic analysis of the first order Gauss-Markov source model to further explain the qualitative behavior of each PLF.

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

# A  Operational RDP Region

It is not feasible to compute the region $\overline{RDP}^o$ directly since it involves searching over all possible encoding-decoding functions. But, for first-order Markov sources where the Markov chain $X_1 \to X_2 \to X_3$ holds, the following region can be used as an approximation. So, with this motivation, we introduce the information RDP region as follows.

**Definition A.1 (Information RDP Region)** *For first-order Markov sources, let the information RDP region, denoted by $\overline{RDP}$, be the set of all tuples $(\mathsf{R}, \mathsf{D}, \mathsf{P})$ which satisfy the following*

$$R_1 \geq I(X_1; X_{r,1}), \tag{27}$$
$$R_2 \geq I(X_2; X_{r,2}|X_{r,1}), \tag{28}$$
$$R_3 \geq I(X_3; X_{r,3}|X_{r,1}, X_{r,2}), \tag{29}$$
$$D_j \geq \mathbb{E}[\|X_j - \hat{X}_j\|^2], \qquad j = 1, 2, 3, \tag{30}$$
$$P_j \geq \phi_j(P_{\hat{X}_1 \ldots \hat{X}_{j-1} X_j}, P_{\hat{X}_1 \ldots \hat{X}_{j-1} \hat{X}_j}), \qquad j = 1, 2, 3, \tag{31}$$

*for auxiliary random variables $(X_{r,1}, X_{r,2}, X_{r,3})$ and $(\hat{X}_1, \hat{X}_2, \hat{X}_3)$ such that*

$$\hat{X}_1 = \eta_1(X_{r,1}), \quad \hat{X}_2 = \eta_2(X_{r,1}, X_{r,2}), \quad \hat{X}_3 = X_{r,3}, \tag{32}$$
$$X_{r,1} \to X_1 \to (X_2, X_3), \tag{33}$$
$$X_{r,2} \to (X_2, X_{r,1}) \to (X_1, X_3), \tag{34}$$
$$X_{r,3} \to (X_3, X_{r,1}, X_{r,2}) \to (X_1, X_2), \tag{35}$$

*for some deterministic functions $\eta_1(.)$ and $\eta_2(.,.)$. Moreover, for a given $(\mathsf{D}, \mathsf{P})$, the compression rate region, denoted by $\overline{R}(\mathsf{D}, \mathsf{P})$, is the closure of the set of all tuples $\mathsf{R}$ that $(\mathsf{R}, \mathsf{D}, \mathsf{P}) \in \overline{RDP}$.*

**Proposition A.2** *For first-order Markov sources, a given $(\mathsf{D}, \mathsf{P})$ and $\mathsf{R} \in \overline{R}(\mathsf{D}, \mathsf{P})$, we have*

$$\mathsf{R} + \log(\mathsf{R} + 1) + 5 \in \overline{R}^o(\mathsf{D}, \mathsf{P}). \tag{36}$$

*Moreover, the following holds:*

$$\overline{R}^o(\mathsf{D}, \mathsf{P}) \subseteq \overline{R}(\mathsf{D}, \mathsf{P}). \tag{37}$$

To prove the above statement, we first discuss the achievable scheme that results in equation 36. Then, we will provide the proof of outer bound in equation 37.

Before stating the achievable scheme, we recall the strong functional representation lemma (SFRL) from Li & El Gamal (2018). It states that for jointly distributed random variables $X$ and $Y$, there exists a random variable $U$ independent of $X$, and function $\phi$ such that $Y = \phi(X, U)$. Here, $U$ is not necessarily unique. The strong functional representation lemma states further that $U$ should be chosen to satisfy

$$H(Y|U) \leq I(X; Y) + \log(I(X; Y) + 1) + 4. \tag{38}$$

Notice that the strong functional representation lemma can be applied conditionally. Given $P_{XY|W}$, we can represent $Y$ as a function of $(X, W, U)$ such that $U$ is independent of $(X, W)$ and

$$H(Y|W, U) \leq I(X; Y|W) + \log(I(X; Y|W) + 1) + 4. \tag{39}$$

*Proof of equation 36 (Inner bound)*:

For a given $(\mathsf{D}, \mathsf{P})$ and $\mathsf{R} \in \overline{R}(\mathsf{D}, \mathsf{P})$, let $\mathsf{X}_r = (X_{r,1}, X_{r,2}, X_{r,3})$ be jointly distributed with $\mathsf{X} = (X_1, X_2, X_3)$ where the Markov chains equation 33–equation 35 hold and the rate constraints in equation 27–equation 29

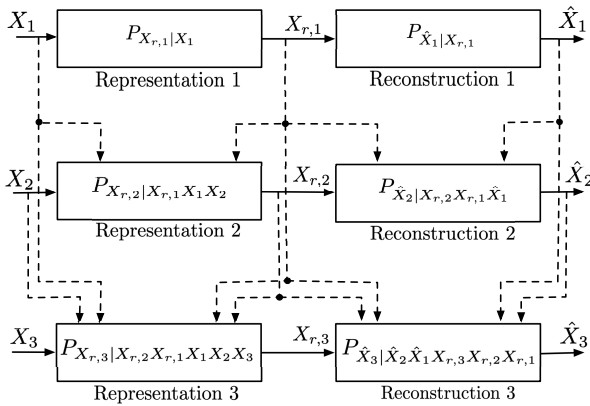

Figure 9: Encoded representations and reconstructions of the iRDP region $\overline{RDP}$.

are satisfied such that there exist $(\hat{X}_1, \hat{X}_2, \hat{X}_3)$ for which distortion-perception constraints equation 30–equation 31 hold. Denote the joint distribution of $(\mathsf{X}, \mathsf{X}_r, \hat{\mathsf{X}})$ by $P_{\mathsf{X}\mathsf{X}_r\hat{\mathsf{X}}}$ and notice that according to the Markov chains in equation 33–equation 35, it factorizes as the following

$$P_{\mathsf{X}\mathsf{X}_r\hat{\mathsf{X}}} = P_{X_1X_2X_3} \cdot P_{X_{r,1}|X_1} \cdot P_{X_{r,2}|X_{r,1}X_2} \cdot$$
$$P_{X_{r,3}|X_{r,2}X_{r,1}X_3} \cdot \mathbb{1}\{\hat{X}_1 = g_1(X_{r,1})\} \cdot \mathbb{1}\{\hat{X}_2 = g_2(X_{r,1}, X_{r,3})\} \cdot \mathbb{1}\{\hat{X}_3 = X_{r,3}\}. \tag{40}$$

For an illustration of encoded representations $\mathsf{X}_r$ and reconstructions $\hat{\mathsf{X}}$ in $\overline{R}(\mathsf{D}, \mathsf{P})$ which are induced by distribution $P_{\mathsf{X}\mathsf{X}_r\hat{\mathsf{X}}}$, see Fig. 9.

Now, we show that $\mathsf{R} + \log(\mathsf{R}+1) + 5 \in \overline{R}(\mathsf{D}, \mathsf{P})$. The achievable scheme is as follows. We construct the codebook according to the joint distribution in equation 40. By marginalizing this distribution over $(\mathsf{X}, \hat{\mathsf{X}})$, we obtain the following distribution $P_{\mathsf{X}_r}$

$$P_{\mathsf{X}_r} = P_{X_{r,1}} P_{X_{r,2}|X_{r,1}} P_{X_{r,3}|X_{r,2}X_{r,1}}. \tag{41}$$

From the SFRL (Li & El Gamal, 2018), we know that

- there exist a random variable $V_1$ independent of $X_1$ and a deterministic function $q_1$ such that $X_{r,1} = q_1(X_1, V_1)$ with the distribution given in 41, and

$$H(X_{r,1}|V_1) \leq I(X_1; X_{r,1}) + \log(I(X_1; X_{r,1}) + 1) + 4. \tag{42}$$

  Thus, the first encoder observes the source $X_1$ and applies the function $q_1$ to get $X_{r,1}$;

- according to the conditional SFRL, there exist a random variable $V_2$ independent of $(X_2, X_{r,1})$ and a deterministic function $q_2$ such that $X_{r,2} = q_2(X_{r,1}, X_2, V_2)$ with its conditional distribution given $X_{r,1}$ specified in equation 41, and

$$H(X_{r,2}|X_{r,1}, V_2) \leq I(X_2; X_{r,2}|X_{r,1}) + \log(I(X_2; X_{r,2}|X_{r,1}) + 1) + 4. \tag{43}$$

  At the second step, the representation $X_{r,1}$ is available at the second encoder. So, upon observing the source $X_2$, it applies the function $q_2$ to get $X_{r,2}$ (see Fig. 10);

- according to the conditional SFRL, there exist a random variable $V_3$ independent of $(X_3, X_{r,1}, X_{r,2})$ and a deterministic function $q_3$ such that $X_{r,3} = q_3(X_{r,1}, X_{r,2}, X_3, V_3)$ and

$$H(X_{r,3}|X_{r,1}, X_{r,2}, V_3) \leq I(X_3; X_{r,3}|X_{r,1}, X_{r,2}) + \log(I(X_3; X_{r,3}|X_{r,1}, X_{r,2}) + 1) + 4. \tag{44}$$

Now, the encoding and decoding are as follows

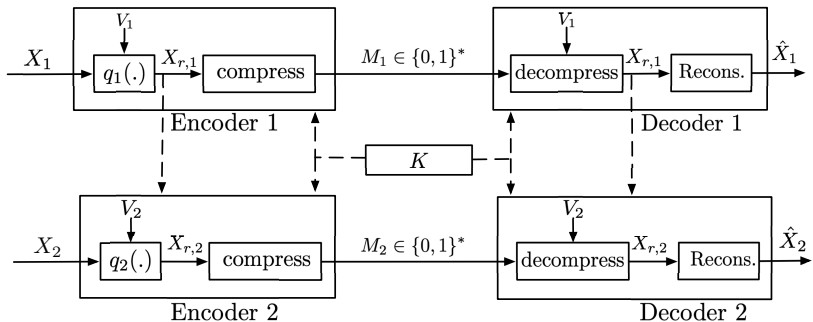

Figure 10: Strong functional representation lemma for $T = 2$ frames.

- With $V_1$ available at all encoders and decoders, we can have a class of prefix-free binary codes indexed by $V_1$ with the expected codeword length not larger than $I(X_1; X_{r,1}) + \log(I(X_1; X_{r,1}) + 1) + 5$ to encode this index losslessly (see Fig. 10).

- With $V_2$ available at the encoders and decoders, we can design a set of prefix-free binary codes indexed by $(V_2, X_{r,1})$ with expected codeword length not larger than $I(X_2; X_{r,2}|X_{r,1}) + \log(I(X_2; X_{r,2}|X_{r,1}) + 1) + 5$ to encode this index losslessly (see Fig. 10).

- Similarly, with $V_3$ available at the third encoder and decoder, one can represent the discrete index corresponding to $X_{r,3}$ losslessly.

- The decoders can use functions $\hat{X}_1 = \eta_1(X_{r,1})$, $\hat{X}_2 = \eta_2(X_{r,1}, X_{r,2})$ and $\hat{X}_3 = X_{r,3}$ to get the reconstruction $\hat{X}$.

Taking these facts together with the rate constraints in equation 27-equation 29 shows that $R + \log(R+1) + 5 \in \overline{R}^o(D, P)$.

*Proof of equation 37 (Outer Bound)*:

For any $(D, P)$, $R \in \overline{R}^o(D, P)$, shared randomness $K$, encoding functions $f_j \colon \mathcal{X}_1 \times \ldots \times \mathcal{X}_j \times \mathcal{K} \to \mathcal{M}_j$ and decoding functions $g_j \colon \mathcal{M}_1 \times \mathcal{M}_2 \times \ldots \times \mathcal{M}_j \times \mathcal{K} \to \mathcal{X}_j$ such that

$$R_j \geq \mathbb{E}[\ell(M_j)], \qquad j = 1, 2, 3, \tag{45}$$

and

$$D_j \geq \mathbb{E}[\|X_j - \hat{X}_j\|^2], \qquad j = 1, 2, 3, \tag{46}$$

$$P_j \geq \phi_j(P_{\hat{X}_1 \ldots \hat{X}_{j-1} X_j}, P_{\hat{X}_1 \ldots \hat{X}_j}), \qquad j = 1, 2, 3, \tag{47}$$

we lower bound the expected length of the messages. Define

$$X_{r,1} := (M_1, K), \tag{48}$$

$$X_{r,2} := (M_1, M_2, K), \tag{49}$$

and recall that according to the decoding functions, we have

$$\hat{X}_j = g_j(M_1, \ldots, M_j, K), \qquad j = 1, 2, 3. \tag{50}$$

We can write

$$R_1 \geq \mathbb{E}[\ell(M_1)] \tag{51}$$

$$\geq H(M_1|K) \tag{52}$$

$$= I(X_1; M_1|K) \tag{53}$$

$$= I(X_1; M_1, K) \tag{54}$$

$$= I(X_1; X_{r,1}). \tag{55}$$

Now, consider the following set of inequalities

$$R_2 \geq \mathbb{E}[\ell(M_2)] \tag{56}$$
$$\geq H(M_2|M_1, K) \tag{57}$$
$$= I(X_1, X_2; M_2|M_1, K) \tag{58}$$
$$= I(X_1, X_2; X_{2,r}|X_{r,1}). \tag{59}$$

Similarly, we have

$$R_3 \geq \mathbb{E}[\ell(M_3)] \tag{60}$$
$$\geq H(M_3|M_1, M_2, K) \tag{61}$$
$$= I(X_1, X_2, X_3; M_3|M_1, M_2, K) \tag{62}$$
$$\geq I(X_1, X_2, X_3; \hat{X}_3|X_{r,1}, X_{r,2}). \tag{63}$$

Notice that the definitions in equation 48–equation 49 imply the following Markov chains

$$X_{r,1} \to X_1 \to (X_2, X_3), \tag{64}$$
$$X_{r,2} \to (X_1, X_2, X_{r,1}) \to X_3. \tag{65}$$

On the other hand, the decoding functions of the first and second steps are as follows

$$\hat{X}_1 = g_1(M_1, K), \tag{66}$$
$$\hat{X}_2 = g_2(M_1, M_2, K), \tag{67}$$

where together with definitions in equation 48 and equation 49, we can write

$$\hat{X}_1 = g_1(M_1, K) := \eta_1(X_{r,1}), \tag{68}$$
$$\hat{X}_2 = g_2(M_1, M_2, K) := \eta_2(X_{r,1}, X_{r,2}), \tag{69}$$

such that $\eta_1(.)$ and $\eta_2(.,.)$ are deterministic functions.

Now, consider the fact that the set of constraints in equation 46–equation 47, equation 55, equation 59, equation 63 with Markov chains in equation 64–equation 65 and deterministic functions in equation 68–equation 69 constitute an iRDP region, denoted by $\overline{RDP}^{\text{OB}}$, which is the set of all tuples $(\mathsf{R}, \mathsf{D}, \mathsf{P})$ such that

$$R_1 \geq I(X_1; X_{r,1}), \tag{70}$$
$$R_2 \geq I(X_1, X_2; X_{r,2}|X_{r,1}), \tag{71}$$
$$R_3 \geq I(X_1, X_2, X_3; \hat{X}_3|X_{r,1}, X_{r,2}), \tag{72}$$
$$D_j \geq \mathbb{E}[\|X_j - \hat{X}_j\|^2], \qquad j = 1, 2, 3, \tag{73}$$
$$P_j \geq \phi_j(P_{\hat{X}_1 \ldots \hat{X}_{j-1} X_j}, P_{\hat{X}_1 \ldots \hat{X}_j}), \quad j = 1, 2, 3, \tag{74}$$

for auxiliary random variables $(X_{r,1}, X_{r,2})$ and $(\hat{X}_1, \hat{X}_2, \hat{X}_3)$ satisfying the following

$$\hat{X}_1 = \eta_1(X_{r,1}), \quad \hat{X}_2 = \eta_2(X_{r,1}, X_{r,2}), \tag{75}$$
$$X_{r,1} \to X_1 \to (X_2, X_3), \tag{76}$$
$$X_{r,2} \to (X_1, X_2, X_{r,1}) \to X_3. \tag{77}$$

for some deterministic functions $\eta_1(.)$ and $\eta_2(.,.)$.

Comparing the two regions $\overline{RDP}^{\text{OB}}$ and $\overline{RDP}$, we identify the following differences. The Markov chain in equation 34 is more restrictive than the one in equation 77. Moreover, the Markov chain in equation 35 does not appear in $\overline{RDP}^{\text{OB}}$. Despite these distinctions, the following lemma shows that $\overline{RDP}^{\text{OB}} = \overline{RDP}$. Now, for a given $(\mathsf{D}, \mathsf{P})$, let $\overline{R}^{\text{OB}}(\mathsf{D}, \mathsf{P})$ denote the set of rate tuples $\mathsf{R}$ such $(\mathsf{R}, \mathsf{D}, \mathsf{P}) \in \overline{RDP}^{\text{OB}}$, then this lemma implies that $\overline{R}^{\text{OB}}(\mathsf{D}, \mathsf{P}) = \overline{R}(\mathsf{D}, \mathsf{P})$ which completes the proof of the outer bound.

We conclude this section by the following lemma.

**Lemma A.3** *For first-order Markov sources, we have*

$$\overline{RDP} = \overline{RDP}^{OB}. \tag{78}$$

*Proof:* This result for the scenario without perception constraint has been similarly observed in Eq. (12) of Stavrou et al. (2022). The proof in this section is provided for completeness.

First, observe that the Markov chain constraints in equation 33–equation 35 are more restrictive than those in equation 76–equation 77. Consequently, we have $\overline{RDP} \subseteq \overline{RDP}^{OB}$. Now, it remains to show that $\overline{RDP}^{OB} \subseteq \overline{RDP}$. To establish this, we need to verify that imposing the Markov chains in equation 33–equation 35 does not alter the constraints in equation 70–equation 74. Consider the following facts

1. The distortion constraints in equation 73 depend only on the joint distribution of $(X_j, \hat{X}_j)$, and thus on the joint distribution of $(X_j, X_{r,1}, \ldots, X_{r,j})$, since $\hat{X}_1 = \eta_1(X_{r,1}), \hat{X}_2 = \eta_2(X_{r,1}, X_{r,2}), \hat{X}_3 = X_{r,3}$. So, imposing the Markov chain $X_{r,2} \to (X_2, X_{r,1}) \to X_1$ does not affect the expected distortion $\mathbb{E}[\|X_2 - \hat{X}_2\|^2]$, as this quantity is independent of the joint distribution between $X_1$ and $(X_{r,1}, X_{r,2}, X_2)$. The same reasoning applies to the remaining frames;

2. The perception constraints in equation 74 depend on the joint distributions $P_{\hat{X}_1 \ldots \hat{X}_{j-1} X_j}$ and $P_{\hat{X}_1, \ldots, \hat{X}_j}$, and hence on $P_{X_{r,1} \ldots X_{r,j}}$, since $\hat{X}_1 = \eta_1(X_{r,1}), \hat{X}_2 = \eta_2(X_{r,1}, X_{r,2}), \hat{X}_3 = X_{r,3}$. Thus, imposing $X_{r,2} \to (X_2, X_{r,1}) \to X_1$ does not affect $\phi_2(P_{\hat{X}_1 X_2}, P_{\hat{X}_1 \hat{X}_2})$ since it does not depend on the joint distribution of $X_1$ with $(X_{r,1}, X_{r,2}, X_2)$. A similar argument holds for other frames;

3. Moreover, the rate constraints in equation 71 and equation 72 would be further lower bounded by

$$R_2 \geq I(X_1, X_2; X_{r,2}|X_{r,1})$$
$$\geq I(X_2; X_{r,2}|X_{r,1}), \tag{79}$$
$$R_3 \geq I(X_1, X_2, X_3; \hat{X}_3|X_{r,1}, X_{r,2})$$
$$\geq I(X_3; \hat{X}_3|X_{r,1}, X_{r,2}). \tag{80}$$

Furthermore, the mutual information terms $I(X_1; X_{r,1})$, $I(X_2; X_{r,2}|X_{r,1})$ and $I(X_3; \hat{X}_3|X_{r,1}, X_{r,2})$ depend on distributions $P_{X_1 X_{r,1}}$, $P_{X_{r,1} X_{r,2} X_2}$ and $P_{X_3 \hat{X}_3 X_{r,1} X_{r,2}}$, respectively. These distributions must therefore be preserved under the imposed Markov chains equation 33–equation 35. The first two are maintained by the structure in equation 33–equation 34. Moreover, because the sources form a first-order Markov chain, preserving $P_{X_{r,1} X_1}$ and $P_{X_{r,1} X_{r,2} X_2}$ is sufficient to preserve $P_{X_{r,1} X_{r,2} X_3}$. As a result, ensuring that the joint distribution $P_{\hat{X}_3 X_{r,1} X_{r,2}}$ is preserved is enough to keep $I(X_3; \hat{X}_3|X_{r,1}, X_{r,2})$ unchanged.

Considering the above four facts, without loss of optimality, one can impose the following Markov chains

$$X_{r,1} \to X_1 \to (X_2, X_3), \tag{81}$$
$$X_{r,2} \to (X_2, X_{r,1}) \to (X_1, X_3), \tag{82}$$
$$\hat{X}_3 \to (X_3, X_{r,1}, X_{r,2}) \to (X_1, X_2). \tag{83}$$

This concludes the proof of the lemma.

∎

# B    Gauss-Markov Source Model

In this section, we prove that for Gaussian sources, jointly Gaussian reconstructions are optimal.

**Proposition B.1** *For the Gauss-Markov source model, any tuple* $(\mathsf{R}, \mathsf{D}, \mathsf{P}) \in \overline{RDP}$ *can be attained by a jointly Gaussian distribution over* $(X_{r,1}, X_{r,2}, X_{r,3})$ *and identity mappings for* $\eta_j(\cdot)$ *in Definition A.1.*

First, notice that a proof for the setting without perception constraint is provided in Khina et al. (2019). The following proof is different from Khina et al. (2019) in some steps and also involves the perception constraint.

For a given tuple $(\mathsf{R}, \mathsf{D}, \mathsf{P}) \in \overline{RDP}$, let $X_{r,1}^*$, $X_{r,2}^*$, $\hat{X}_1^* = \eta_1(X_{r,1}^*)$, $\hat{X}_2^* = \eta_2(X_{r,1}^*, X_{r,2}^*)$ and $\hat{X}_3^*$ be random variables satisfying equation 32–equation 34. Let $P_{\hat{X}_1^G|X_1}$, $P_{\hat{X}_2^G|\hat{X}_1^G X_2}$ and $P_{\hat{X}_3^G|\hat{X}_1^G \hat{X}_2^G X_3}$ be jointly Gaussian distributions such that the following conditions are satisfied:

$$\text{cov}(\hat{X}_1^G, X_1) = \text{cov}(\hat{X}_1^*, X_1), \tag{84}$$

$$\text{cov}(\hat{X}_1^G, \hat{X}_2^G, X_2) = \text{cov}(\hat{X}_1^*, \hat{X}_2^*, X_2), \tag{85}$$

$$\text{cov}(\hat{X}_1^G, \hat{X}_2^G, \hat{X}_3^G, X_3) = \text{cov}(\hat{X}_1^*, \hat{X}_2^*, \hat{X}_3^*, X_3). \tag{86}$$

In general, the Gaussian random variables which satisfy the constraints in equation 84–equation 86 can be written in the following format

$$X_1 = \nu \hat{X}_1^G + Z_1, \tag{87}$$

$$\hat{X}_2^G = \omega_1 \hat{X}_1^G + \omega_2 X_2 + Z_2, \tag{88}$$

$$\hat{X}_3^G = \tau_1 \hat{X}_1^G + \tau_2 \hat{X}_2^G + \tau_3 X_3 + Z_3, \tag{89}$$

for some real $\nu$, $\omega_1$, $\omega_2$, $\tau_1$, $\tau_2$, $\tau_3$ where $\hat{X}_1^G \sim \mathcal{N}(0, \sigma_{\hat{X}_1^G}^2)$, $\hat{X}_2^G \sim \mathcal{N}(0, \sigma_{\hat{X}_2^G}^2)$, $Z_1$, $Z_2$ and $Z_3$ are Gaussian random variables with zero mean and variances $\alpha_1^2, \alpha_2^2, \alpha_3^2$, independent of $\hat{X}_1^G$, $(\hat{X}_1^G, X_2)$ and $(\hat{X}_1^G, \hat{X}_2^G, X_3)$, respectively.

We explicitly derive the coefficients $\nu$, $\omega_1$, $\omega_2$, $\tau_1$, $\tau_2$ and $\tau_3$ in the following. Multiplying both sides of equation 87 by $\hat{X}_1^G$ and taking an expectation, we get

$$\mathbb{E}[X_1 \hat{X}_1^G] = \nu \sigma_{\hat{X}_1^G}^2. \tag{90}$$

According to equation 84, the above equation can be written as follows

$$\mathbb{E}[X_1 \hat{X}_1^*] = \nu \mathbb{E}[\hat{X}_1^{*2}]. \tag{91}$$

Multiplying both sides of equation 88 by the vector $[\hat{X}_1^G \ X_2]$ and taking an expectation, we have

$$[\mathbb{E}[\hat{X}_1^G \hat{X}_2^G] \ \mathbb{E}[X_2 \hat{X}_2^G]] = [\omega_1 \ \omega_2] \begin{pmatrix} \sigma_{\hat{X}_1^G}^2 & \mathbb{E}[X_2 \hat{X}_1^G] \\ \mathbb{E}[X_2 \hat{X}_1^G] & \sigma_2^2 \end{pmatrix}. \tag{92}$$

Considering the fact that $\mathbb{E}[X_2 \hat{X}_1^G] = \rho_1 \mathbb{E}[X_1 \hat{X}_1^G]$ and according to equation 85, the above equation can be written as follows

$$[\mathbb{E}[\hat{X}_1^* \hat{X}_2^*] \ \mathbb{E}[X_2 \hat{X}_2^*]] = [\omega_1 \ \omega_2] \begin{pmatrix} \mathbb{E}[\hat{X}_1^{*2}] & \rho_1 \mathbb{E}[X_1 \hat{X}_1^*] \\ \rho_1 \mathbb{E}[X_1 \hat{X}_1^*] & \sigma_2^2 \end{pmatrix}. \tag{93}$$

Similarly, multiplying both sides of equation 89 by the vector $[\hat{X}_1^G \ \hat{X}_2^G \ X_3]$, taking an expectation and considering equation 86, we get

$$[\mathbb{E}[\hat{X}_1^* \hat{X}_3^*] \ \mathbb{E}[\hat{X}_2^* \hat{X}_3^*] \ \mathbb{E}[X_3 \hat{X}_3^*]] = [\tau_1 \ \tau_2 \ \tau_3] \begin{pmatrix} \mathbb{E}[\hat{X}_1^{*2}] & \mathbb{E}[\hat{X}_1^* \hat{X}_2^*] & \rho_1 \rho_2 \mathbb{E}[X_1 \hat{X}_1^*] \\ \mathbb{E}[\hat{X}_1^* \hat{X}_2^*] & \mathbb{E}[\hat{X}_2^{*2}] & \rho_2 \mathbb{E}[X_2 \hat{X}_2^*] \\ \rho_1 \rho_2 \mathbb{E}[X_1 \hat{X}_1^*] & \rho_2 \mathbb{E}[X_2 \hat{X}_2^*] & \mathbb{E}[\hat{X}_3^{*2}] \end{pmatrix}. \tag{94}$$

Solving equations equation 91, equation 93 and equation 94, we get

$$\sigma^2_{\hat{X}^G_1} = \mathbb{E}[\hat{X}^{*2}_1], \tag{95}$$

$$\nu = \frac{\mathbb{E}[X_1\hat{X}^*_1]}{\mathbb{E}[\hat{X}^{*2}_1]}, \tag{96}$$

$$\alpha^2_1 = \sigma^2_1 - \frac{\mathbb{E}[X_1\hat{X}^*_1]}{\mathbb{E}[\hat{X}^{*2}_1]}, \tag{97}$$

$$\omega_1 = \frac{\nu\rho_1\mathbb{E}[\hat{X}^*_1\hat{X}^*_2] - \mathbb{E}[X_2\hat{X}^*_2]}{\nu^2\rho^2_1\sigma^2_{\hat{X}^G_1} - \sigma^2_2}, \tag{98}$$

$$\omega_2 = \frac{\nu\rho_1\sigma^2_{\hat{X}^G_1}\mathbb{E}[X_2\hat{X}^*_2] - \sigma^2_2\mathbb{E}[\hat{X}^*_1\hat{X}^*_2]}{\nu^2\rho^2_1\sigma^4_{\hat{X}^G_1} - \sigma^2_2\sigma^2_{\hat{X}^G_1}}, \tag{99}$$

$$\alpha^2_2 = \mathbb{E}[\hat{X}^{*2}_2] - \alpha^2_2\sigma^2_{\hat{X}^G_1} - \omega^2_2\sigma^2_2 - 2\omega_1\omega_2\rho_1\nu\sigma^2_{\hat{X}^G_1}. \tag{100}$$

For the third step, the coefficients and noise variance of equation 89 are given as follows

$$[\tau_1 \ \ \tau_2 \ \ \tau_3] = [\mathbb{E}[\hat{X}^*_1\hat{X}^*_3] \ \ \mathbb{E}[\hat{X}^*_2\hat{X}^*_3] \ \ \mathbb{E}[X_3\hat{X}^*_3]] \cdot \begin{pmatrix} \mathbb{E}[\hat{X}^{*2}_1] & \mathbb{E}[\hat{X}^*_1\hat{X}^*_2] & \rho_1\rho_2\mathbb{E}[X_1\hat{X}^*_1] \\ \mathbb{E}[\hat{X}^*_1\hat{X}^*_2] & \mathbb{E}[\hat{X}^{*2}_2] & \rho_2\mathbb{E}[X_2\hat{X}^*_2] \\ \rho_1\rho_2\mathbb{E}[X_1\hat{X}^*_1] & \rho_2\mathbb{E}[X_2\hat{X}^*_2] & \mathbb{E}[\hat{X}^{*2}_3] \end{pmatrix}^{-1}, \tag{101}$$

$$\alpha^2_3 = \mathbb{E}[\hat{X}^{*2}_3] - \tau^2_1\mathbb{E}[\hat{X}^{*2}_1] - \tau^2_2\mathbb{E}[\hat{X}^{*2}_2] - \tau^2_3\mathbb{E}[X^2_3] - 2\tau_1\tau_2\mathbb{E}[\hat{X}^*_1\hat{X}^*_2] - 2\tau_1\tau_3\rho_1\rho_2\mathbb{E}[X_1\hat{X}^*_1] - 2\tau_2\tau_3\rho_2\mathbb{E}[X_2\hat{X}^*_2], \tag{102}$$

where $(.)^{-1}$ denotes the inverse of a matrix.

Now, we look at the rate constraints.

*Rate Constraints*:

Consider the rate constraint of the first step as follows

$$R_1 \geq I(X_1; X^*_{r,1}) \tag{103}$$

$$= h(X_1) - h(X_1|X^*_{r,1}) \tag{104}$$

$$\geq h(X_1) - h(X_1|\hat{X}^*_1) \tag{105}$$

$$= h(X_1) - h(X_1 - \mathbb{E}[X_1|\hat{X}^*_1]|\hat{X}^*_1) \tag{106}$$

$$\geq h(X_1) - h(X_1 - \mathbb{E}[X_1|\hat{X}^*_1]) \tag{107}$$

$$\geq h(X_1) - h(X_1 - \mathbb{E}[X_1|\hat{X}^G_1]) \tag{108}$$

$$= h(X_1) - h(X_1 - \mathbb{E}[X_1|\hat{X}^G_1]|\hat{X}^G_1) \tag{109}$$

$$= I(X_1; \hat{X}^G_1), \tag{110}$$

where

- equation 105 follows because $\hat{X}^*_1$ is a function of $X^*_{r,1}$;

- equation 106 follows because $\mathbb{E}[X_1|\hat{X}^*_1]$ is a deterministic function of $\hat{X}^*_1$;

- equation 108 follows because for a given covariance matrix in equation 84, the Gaussian distribution maximizes the differential entropy;

- equation 109 follows because the MMSE is uncorrelated from the data and since the random variables are Gaussian, the MMSE would be independent of the data.

Next, consider the rate constraint of the second step as the following

$$R_2 \geq I(X_2; X_{r,2}^* | X_{r,1}^*) \tag{111}$$
$$= h(X_2 | X_{r,1}^*) - h(X_2 | X_{r,1}^*, X_{r,2}^*) \tag{112}$$
$$\geq h(X_2 | X_{r,1}^*) - h(X_2 | \hat{X}_1^*, \hat{X}_2^*) \tag{113}$$
$$\geq h(X_2 | X_{r,1}^*) - h(X_2 | \hat{X}_1^G, \hat{X}_2^G) \tag{114}$$
$$= h(\rho_1 X_1 + N_1 | X_{r,1}^*) - h(X_2 | \hat{X}_1^G, \hat{X}_2^G) \tag{115}$$
$$\geq \frac{1}{2} \log \left( \rho_1^2 2^{2h(X_1 | X_{r,1}^*)} + 2^{2h(N_1)} \right) - h(X_2 | \hat{X}_1^G, \hat{X}_2^G) \tag{116}$$
$$\geq \frac{1}{2} \log \left( \rho_1^2 2^{-2R_1} 2^{2h(X_1)} + 2^{2h(N_1)} \right) - h(X_2 | \hat{X}_1^G, \hat{X}_2^G), \tag{117}$$

where

- equation 113 follows because $\hat{X}_1^*$ and $\hat{X}_2^*$ are deterministic functions of $X_{r,1}^*$ and $(X_{r,1}^*, X_{r,2}^*)$, respectively;

- equation 114 follows because for a given covariance matrix in equation 85, the Gaussian distribution maximizes the differential entropy;

- equation 116 follows from entropy power inequality (EPI) (see pp. 22 of El Gamal & Kim (2011));

- equation 117 follows from equation 104.

Similarly, consider the rate constraint of the third frame as the following,

$$R_3 \geq I(X_3; \hat{X}_3^* | X_{r,1}^*, X_{r,2}^*) \tag{118}$$
$$= h(X_3 | X_{r,1}^*, X_{r,2}^*) - h(X_3 | X_{r,1}^*, X_{r,2}^*, \hat{X}_3^*) \tag{119}$$
$$\geq h(X_3 | X_{r,1}^*, X_{r,2}^*) - h(X_3 | \hat{X}_1^*, \hat{X}_2^*, \hat{X}_3^*) \tag{120}$$
$$\geq h(X_3 | X_{r,1}^*, X_{r,2}^*) - h(X_3 | \hat{X}_1^G, \hat{X}_2^G, \hat{X}_3^G) \tag{121}$$
$$= h(\rho_2 X_2 + N_2 | X_{r,1}^*, X_{r,2}^*) - h(X_3 | \hat{X}_1^G, \hat{X}_2^G, \hat{X}_3^G) \tag{122}$$
$$\geq \frac{1}{2} \log \left( \rho_2^2 2^{2h(X_2 | X_{r,1}^*, X_{r,2}^*)} + 2^{2h(N_2)} \right) - h(X_3 | \hat{X}_1^G, \hat{X}_2^G, \hat{X}_3^G) \tag{123}$$
$$\geq \frac{1}{2} \log \left( \rho_2^2 2^{-2R_2} 2^{2h(X_2 | X_{r,1}^*)} + 2^{2h(N_2)} \right) - h(X_3 | \hat{X}_1^G, \hat{X}_2^G, \hat{X}_3^G) \tag{124}$$
$$\geq \frac{1}{2} \log \left( \rho_1^2 \rho_2^2 2^{-2R_1 - 2R_2} 2^{2h(X_1)} + \rho_2^2 2^{-2R_2} 2^{2h(N_1)} + 2^{2h(N_2)} \right) - h(X_3 | \hat{X}_1^G, \hat{X}_2^G, \hat{X}_3^G). \tag{125}$$

Next, we look at the distortion constraint.

*Distortion Constraint*: The choices in equation 84–equation 86 imply that

$$D_j \geq \mathbb{E}[\|X_j - \hat{X}_j^*\|^2] = \mathbb{E}[\|X_j - \hat{X}_j^G\|^2], \qquad j = 1, 2, 3. \tag{126}$$

Finally, we look at the perception constraint

*Perception Constraint*:

Define the following distribution

$$P_{U^*V^*} := \arg \inf_{\substack{\tilde{P}_{UV}: \\ \tilde{P}_U = P_{X_1} \\ \tilde{P}_V = P_{\hat{X}_1^*}}} \mathbb{E}_{\tilde{P}}[\|U - V\|^2]. \tag{127}$$

Now, define $P_{U^G V^G}$ to be a Gaussian joint distribution with the following covariance matrix

$$\text{cov}(U^G, V^G) = \text{cov}(U^*, V^*). \tag{128}$$

Then, we have the following set of inequalities:

$$P_1 \geq W_2^2(P_{X_1}, P_{\hat{X}_1^*}) \tag{129}$$

$$= \inf_{\substack{\tilde{P}_{UV}: \\ \tilde{P}_U = P_{X_1} \\ \tilde{P}_V = P_{\hat{X}_1^*}}} \mathbb{E}_{\tilde{P}}[\|U - V\|^2] \tag{130}$$

$$= \mathbb{E}[\|U^* - V^*\|^2] \tag{131}$$

$$= \mathbb{E}[\|U^G - V^G\|^2] \tag{132}$$

$$\geq W_2^2(P_{U^G}, P_{V^G}) \tag{133}$$

$$= \inf_{\substack{\hat{P}_{UV}: \\ \hat{P}_U = P_{U^G} \\ \hat{P}_V = P_{V^G}}} \mathbb{E}_{\hat{P}}[\|U - V\|^2] \tag{134}$$

$$= \inf_{\substack{\hat{P}_{UV}: \\ \hat{P}_U = P_{X_1} \\ \hat{P}_V = P_{\hat{X}_1^G}}} \mathbb{E}_{\hat{P}}[\|U - V\|^2] \tag{135}$$

$$= W_2^2(P_{X_1}, P_{\hat{X}_1^G}), \tag{136}$$

where

- equation 131 follows from the definition in equation 127;

- equation 132 follows from equation 128 which implies that $(U^*, V^*)$ and $(U^G, V^G)$ have the same second-order statistics;

- equation 135 follows because $P_{V^G} = P_{\hat{X}_1^G}$ which is justified in the following. First, notice that both $P_{V^G}$ and $P_{\hat{X}_1^G}$ are Gaussian distributions. Denote the variance of $V^G$ by $\sigma_{V^G}^2$ and recall that the variance of $\hat{X}_1^G$ is denoted by $\sigma_{\hat{X}_1^G}^2$. According to equation 128, $\sigma_{V^G}^2$ is equal to the variance of $V^*$. Also, from equation 127, we know that $P_{V^*} = P_{\hat{X}_1^*}$, hence the variances of $V^*$ and $\hat{X}_1^*$ are the same. On the other side, according to equation 84, we know that the variance of $\hat{X}_1^*$ is equal to $\sigma_{\hat{X}_1^G}^2$. Thus, we conclude that $\sigma_{\hat{X}_1^G}^2 = \sigma_{V^G}^2$, which yields $P_{V^G} = P_{\hat{X}_1^G}$. A similar argument shows that $P_{U^G} = P_{X_1}$.

A similar argument holds for the perception constraint of the second and third steps for both PLFs.

Thus, we have proved the set of Gaussian auxiliary random variables $(\hat{X}_1^G, \hat{X}_2^G, \hat{X}_3^G)$ given in equation 87–equation 89 where the coefficients are chosen according to distortion-perception constraints, provides an outer bound to $\overline{RDP}$ which is the set of all tuples $(\mathsf{R}, \mathsf{D}, \mathsf{P})$ such that

$$R_1 \geq I(X_1; \hat{X}_1^G), \tag{137}$$

$$R_2 \geq \frac{1}{2} \log \left( \rho_1^2 2^{-2R_1} 2^{2h(X_1)} + 2^{2h(N_1)} \right) - h(X_2|\hat{X}_1^G, \hat{X}_2^G), \tag{138}$$

$$R_3 \geq \frac{1}{2} \log \left( \rho_1^2 \rho_2^2 2^{-2R_1 - 2R_2} 2^{2h(X_1)} + \rho_2^2 2^{-2R_2} 2^{2h(N_1)} + 2^{2h(N_2)} \right) - h(X_3|\hat{X}_1^G, \hat{X}_2^G, \hat{X}_3^G), \tag{139}$$

$$D_j \geq \mathbb{E}[\|X_j - \hat{X}_j^G\|^2], \qquad j = 1, 2, 3, \tag{140}$$

$$P_j \geq W_2^2(P_{X_1 \dots X_j}, P_{\hat{X}_1^G \dots \hat{X}_j^G}). \tag{141}$$

Now, we need to show that the above RDP region is also an inner bound to $\overline{RDP}$. This is simply verified by the following choice. In iRDP region of equation 27–equation 35, choose the following:

$$X_{r,j} = \hat{X}_j = \hat{X}_j^G, \qquad j = 1, 2, 3, \tag{142}$$

where $(\hat{X}_1^G, \hat{X}_2^G, \hat{X}_3^G)$ satisfy equation 87–equation 89 with coefficients chosen according to distortion-perception constraints. The lower bounds on distortion and perception constraints in equation 140 and equation 141 are immediately achieved by this choice. Now, we will look at the rate constraints. The achievable rate constraint of the first step can be written as follows

$$R_1 \geq I(X_1; \hat{X}_1^G), \tag{143}$$

which immediately coincides with equation 137. The achievable rate of the second step can be written as follows

$$R_2 \geq I(X_2; \hat{X}_2^G | \hat{X}_1^G) \tag{144}$$
$$= h(X_2 | \hat{X}_1^G) - h(X_2 | \hat{X}_1^G, \hat{X}_2^G) \tag{145}$$
$$= h(\rho_1 X_1 + N_1 | \hat{X}_1^G) - h(X_2 | \hat{X}_1^G, \hat{X}_2^G) \tag{146}$$
$$= \frac{1}{2} \log(\rho_1^2 2^{2h(X_1|\hat{X}_1^G)} + 2^{2h(N_1)}) - h(X_2 | \hat{X}_1^G, \hat{X}_2^G) \tag{147}$$
$$\geq \frac{1}{2} \log\left(\rho_1^2 2^{-2R_1} 2^{2h(X_1)} + 2^{2h(N_1)}\right) - h(X_2 | \hat{X}_1^G, \hat{X}_2^G), \tag{148}$$

where

- equation 147 follows because EPI holds with "equality" for jointly Gaussian distributions (see pp. 22 of El Gamal & Kim (2011));

- equation 148 follows from equation 138.

Thus, the bound in equation 148 coincides with equation 117. A similar argument holds for the achievable rate of the third frame.

Notice that the above proof (both converse and achievability) can be extended to $T$ frames using the sequential analysis that was presented. Thus, without loss of optimality, one can restrict to the jointly Gaussian distributions and identity functions $\eta_1(.)$ and $\eta_2(.,.)$ in iRDP region $\overline{RDP}$.

## C  Low-rate Regime for the First Frame

In this section, we prove the following theorem when the first frame is compressed at a low rate. The rate of the second frame is an arbitrary nonnegative value.

**Theorem C.1** *Let $R_1 = \epsilon$ for a sufficiently small $\epsilon > 0$ and $R_2$ be an arbitrary nonnegative rate where $R_2 \gg \epsilon$. The achievabale distortions for the second frame, $D_{2,SA}(\epsilon, R_2)$ (for 0-PLF-SA), $D_{2,FMD}(\epsilon, R_2)$ (for 0-PLF-FMD) and $D_{2,JD}(\epsilon, R_2)$ (for 0-PLF-JD) are given by*

$$D_{2,SA}(\epsilon, R_2) = 2\sigma^2(1 - \sqrt{1 - 2^{-2R_2}}) + O(\sqrt{\epsilon}), \tag{149}$$
$$D_{2,FMD}(\epsilon, R_2) = 2\sigma^2(1 - \sqrt{1 - 2^{-2R_2} + \rho^2 2\epsilon \ln 2}) + O(\epsilon), \tag{150}$$
$$D_{2,JD}(\epsilon, R_2) = 2\sigma^2(1 - \sqrt{1 - \rho^2}\sqrt{1 - 2^{-2R_2}} - \rho^2\sqrt{2\epsilon \ln 2}) + O(\epsilon). \tag{151}$$

To prove the above theorem, we first recall the optimization problems of the Gauss-Markov source model. Then, we will look at each PLF separately; 0-PLF-SA, 0-PLF-FMD, and 0-PLF-JD. For each of these PLFs, we discuss the second step and provide the analysis of the third step for completeness.

Recall the RDP region of the Gauss-Markov model which is the set of all tuples $(\mathsf{R}, \mathsf{D}, \mathsf{P})$ such that

$$R_1 \geq I(X_1; \hat{X}_1), \tag{152a}$$
$$R_2 \geq I(X_2; \hat{X}_2 | \hat{X}_1), \tag{152b}$$
$$R_3 \geq I(X_3; \hat{X}_3 | \hat{X}_1, \hat{X}_2), \tag{152c}$$
$$D_j \geq \mathbb{E}[\|X_j - \hat{X}_j\|^2], \tag{152d}$$
$$P_j \geq \phi_j(P_{\hat{X}_1 \ldots \hat{X}_{j-1} X_j}, P_{\hat{X}_1 \ldots \hat{X}_{j-1} \hat{X}_j}), \qquad j = 1, 2, 3, \tag{152e}$$

for some auxiliary random variables $(\hat{X}_1, \hat{X}_2, \hat{X}_3)$ which satisfy the following Markov chains

$$\hat{X}_1 \rightarrow X_1 \rightarrow (X_2, X_3), \ \hat{X}_2 \rightarrow (X_2, \hat{X}_1) \rightarrow (X_1, X_3),$$
$$\hat{X}_3 \rightarrow (X_3, \hat{X}_1, \hat{X}_2) \rightarrow (X_1, X_2). \tag{153}$$

For the Gauss-Markov source model, the reconstructions that satisfy the Markov chains in equation 153 can be generally written as follows

$$\hat{X}_1 = \nu X_1 + Z_1, \tag{154}$$
$$\hat{X}_2 = \omega_1 \hat{X}_1 + \omega_2 X_2 + Z_2, \tag{155}$$
$$\hat{X}_3 = \tau_1 \hat{X}_1 + \tau_2 \hat{X}_2 + \tau_3 X_3 + Z_3, \tag{156}$$

where $\hat{X}_j \sim \mathcal{N}(0, \hat{\sigma}_j^2)$ for $j = 1, 2$, $Z_1$, $Z_2$ and $Z_3$ are independent of $X_1$, $(\hat{X}_1, X_2)$ and $(\hat{X}_1, \hat{X}_2, X_3)$, respectively.

According to equation 23, the optimization program of the first step for all 0-PLFs is as follows

$$\min_{P_{\hat{X}_1 | X_1}} \mathbb{E}[\|X_1 - \hat{X}_1\|^2]$$
$$\text{s.t.} \quad I(X_1; \hat{X}_1) \leq R_1,$$
$$P_{X_1} = P_{\hat{X}_1}. \tag{157}$$

Using the choice in equation 154, the optimization program of the first step for simplifies as follows

$$\min_{\nu} 2\sigma^2(1 - \nu) \tag{158a}$$
$$\text{s.t.} \quad \nu^2 \leq (1 - 2^{-2R_1}), \tag{158b}$$

When $R_1 = \epsilon$ for a sufficiently small $\epsilon > 0$, the solution of the above program is as follows

$$D_{1,\text{SA}}(\epsilon) = D_{1,\text{FMD}}(\epsilon) = D_{1,\text{JD}}(\epsilon) = 2\sigma^2(1 - \sqrt{2\epsilon \ln 2}) + O(\epsilon), \tag{159}$$

where the optimal choice of $\nu$ is given by

$$\nu = \sqrt{1 - 2^{-2R_1}} = \sqrt{2\epsilon \ln 2} + O(\epsilon). \tag{160}$$

Next, consider the optimization programs for different steps and PLFs as follows.

## C.1   0-**PLF-SA**

In this section, we provide the optimization programs for different steps of 0-PLF-SA. For the second step, we are able to provide an approximate solution for the low compression rate, i.e., $R_1 = \epsilon$. For the third step, we plot the tradeoff in Fig. 12.

*Second Step:*

The optimization program of the second step is given as follows.

**Proposition C.2** *The optimization program of $0$-PLF-SA for the second step, given in equation 24, can be written as*

$$\min_{\omega_1,\omega_2} \quad 2\sigma^2 - 2\omega_1\rho\nu\sigma^2 - 2\omega_2\sigma^2 \tag{161a}$$

$$s.t. \quad \omega_2^2(1 - \rho^2\nu^2 2^{-2R_2}) \leq (1 - \omega_1^2 - 2\omega_1\omega_2\rho\nu)(1 - 2^{-2R_2}), \tag{161b}$$

$$\omega_1 + \nu\omega_2\rho = \rho\nu, \tag{161c}$$

$$\nu = \sqrt{1 - 2^{-2R_1}}. \tag{161d}$$

*Proof:* According to equation 24, the optimization problem of the second step is as follows,

$$\min_{P_{\hat{X}_2|X_2\hat{X}_1}} \quad \mathbb{E}[\|X_2 - \hat{X}_2\|^2]$$

$$\text{s.t.} \quad I(X_2; \hat{X}_2|\hat{X}_1) \leq R_2,$$

$$P_{\hat{X}_1 X_2} = P_{\hat{X}_1 \hat{X}_2}. \tag{162}$$

We proceed with simplifying the rate constraint as follows,

$$R_2 \geq I(X_2; \hat{X}_2|\hat{X}_1) \tag{163}$$

$$= h(\hat{X}_2|\hat{X}_1) - h(Z_2) \tag{164}$$

$$= h(\omega_2 X_2 + Z_2|\hat{X}_1) - h(Z_2) \tag{165}$$

$$= \frac{1}{2}\log 2^{-2h(Z_2)}\left(\omega_2^2 2^{2h(X_2|\hat{X}_1)} + 2^{2h(Z_2)}\right) \tag{166}$$

$$= \frac{1}{2}\log 2^{-2h(Z_2)}\left(\omega_2^2 2^{2h(\rho X_1 + N_1|\hat{X}_1)} + 2^{2h(Z_2)}\right) \tag{167}$$

$$= \frac{1}{2}\log 2^{-2h(Z_2)}\left(\omega_2^2(\rho^2 2^{2h(X_1|\hat{X}_1)} + 2^{2h(N_1)}) + 2^{2h(Z_2)}\right) \tag{168}$$

$$= \frac{1}{2}\log 2^{-2h(Z_2)}\left(\omega_2^2(\rho^2 2^{2h(X_1|\hat{X}_1)} + (1 - \rho^2)\sigma^2) + 2^{2h(Z_2)}\right) \tag{169}$$

$$\geq \frac{1}{2}\log 2^{-2h(Z_2)}\left(\omega_2^2(\rho^2\sigma^2 2^{-2R_1} + (1 - \rho^2)\sigma^2) + 2^{2h(Z_2)}\right), \tag{170}$$

where

- equation 164 and equation 165 follow from equation 155;

- equation 166 and equation 168 follow because Entropy Power Inequality (EPI) (see pp. 22 of El Gamal & Kim (2011)) holds with equality for Gaussian sources;

- equation 167 follows from equation 7 where $X_2 = \rho X_1 + N_1$;

- equation 170 follows from the rate constraint of the first step, i.e., $R_1 \geq I(X_1; \hat{X}_1)$.

By re-arranging the terms in equation 170 and applying the Gaussian choice in equation 155, we obtain the following set of inequalities

$$(\omega_2^2(\rho^2\sigma^2 2^{-2R_1} + (1 - \rho^2)\sigma^2))2^{-2R_2} \geq (1 - 2^{-2R_2})2^{2h(Z_2)} \tag{171}$$

$$= (1 - 2^{-2R_2}) \cdot (1 - \omega_1^2 - \omega_2^2 - 2\omega_1\omega_2\nu\rho)\sigma^2. \tag{172}$$

Considering that $\nu = \sqrt{1 - 2^{-2R_1}}$ and re-arranging the terms in the above inequality, we get the constraint in equation 161b.

The objective function in equation 161a can be obtained as follows,

$$\mathbb{E}[\|X_2 - \hat{X}_2\|^2] = 2\sigma^2 - 2\mathbb{E}[X_2\hat{X}_2] \tag{173}$$

$$= 2\sigma^2 - 2(\rho\nu\omega_1 + \omega_2)\sigma^2, \tag{174}$$

where the last equality follows from equation 154 and equation 155.

The derivation of the constraint in equation 161c is as follows. We multiply both sides of equation 154 and equation 155 by $X_2$ and $\hat{X}_1$, respectively, and take an expectation from both sides. Thus, we have

$$\mathbb{E}[X_2\hat{X}_1] = \nu\mathbb{E}[X_1X_2] = \nu\rho\sigma^2, \tag{175}$$
$$\mathbb{E}[\hat{X}_1\hat{X}_2] = \omega_1\sigma^2 + \omega_2\mathbb{E}[X_2\hat{X}_1]. \tag{176}$$

Notice that the perception constraint $P_{X_2\hat{X}_1} = P_{\hat{X}_2\hat{X}_1}$ implies that $\mathbb{E}[\hat{X}_1\hat{X}_2] = \mathbb{E}[X_2\hat{X}_1]$ which together with equation 175 and equation 176 yields the constraint in equation 161c. ∎

Now, we provide an approximate solution for the optimization program when the first frame is compressed at a low rate, i.e., $R_1 = \epsilon$ where $\epsilon$ is sufficiently small. In this case, we have

$$1 - 2^{-2R_1} = 2\epsilon\ln 2 + O(\epsilon^2), \tag{177}$$
$$\nu = \sqrt{2\epsilon\ln 2} + O(\epsilon), \tag{178}$$

so the optimization program of the second step in equation 161 simplifies as follows

$$\min_{\omega_1,\omega_2} \quad 2\sigma^2 - 2\omega_1\rho\sigma^2\sqrt{2\epsilon\ln 2 + O(\epsilon^2)} - 2\omega_2\sigma^2 \tag{179a}$$
$$\text{s.t.} \quad \omega_2^2(1 - \rho^2 2^{-2R_2}(2\epsilon\ln 2 + O(\epsilon^2))) \leq (1 - \omega_1^2 - 2\omega_1\omega_2\rho(2\epsilon\ln 2 + O(\epsilon^2)))(1 - 2^{-2R_2}), \tag{179b}$$
$$\omega_1 + \nu\omega_2\rho = \rho\nu. \tag{179c}$$

Notice that equation 179c and equation 178 imply that $\omega_1 = \Theta(\sqrt{\epsilon})$ which together with equation 179b yields the following

$$\omega_2 \leq \sqrt{1 - 2^{-2R_2}} + O(\sqrt{\epsilon}). \tag{180}$$

On the other side, plugging equation 179c into equation 179a and considering dominant terms, the program in equation 179 reduces to the following

$$\min_{\omega_2} \quad 2\sigma^2 - 2\omega_2\sigma^2 + O(\epsilon) \tag{181}$$
$$\text{s.t.} \quad \omega_2 \leq \sqrt{1 - 2^{-2R_2}} + O(\sqrt{\epsilon}). \tag{182}$$

Considering that the objective function of the above program is decreasing in $\omega_2$, the solution of this program is given by

$$\omega_2 = \sqrt{1 - 2^{-2R_2}} + O(\sqrt{\epsilon}). \tag{183}$$

Plugging the above into equation 179c, we get

$$\omega_1 = \rho\sqrt{2\epsilon\ln 2}(1 - \sqrt{1 - 2^{-2R_2}}) + O(\epsilon). \tag{184}$$

Thus, we have

$$\hat{X}_2 = \rho\sqrt{2\epsilon\ln 2}(1 - \sqrt{1 - 2^{-2R_2}})\hat{X}_1 + \sqrt{1 - 2^{-2R_2}}X_2 + Z_2, \tag{185}$$

where $Z_2 \sim \mathcal{N}(0, (2^{-2R_2} - \rho^2(1 - \sqrt{1 - 2^{-2R_2}})^2(2\epsilon\ln 2))\sigma^2)$ and the solution of optimization program is as follows

$$D_{2,\text{SA}}(\epsilon, R_2) = 2\sigma^2(1 - \sqrt{1 - 2^{-2R_2}}) + O(\sqrt{\epsilon}). \tag{186}$$

*Third Step:*

For the third step, we have the following optimization program.

**Proposition C.3** *The optimization program of* $0$*-PLF-SA for the third step can be written as follows*

$$\min_{\tau_1,\tau_2,\tau_3} 2\sigma^2 - 2\tau_3\sigma^2 - 2\tau_2\omega_2\rho\sigma^2 - 2\tau_2\omega_1\nu\rho^2\sigma^2 - 2\tau_1\nu\rho^2\sigma^2 \tag{187a}$$

$$s.t. : \tau_3^2\sigma^2(1 - 2^{-2R_3}(\rho^4 2^{-2R_1-2R_2} + \rho^2(1-\rho^2)2^{-2R_2} - \rho^2)) \le \sigma^2(1 - 2^{-2R_3})\Big(1 - \tau_1^2 - \tau_2^2 - 2\tau_1\tau_2\omega_1\nu$$

$$-2\tau_1\tau_2\omega_2\nu\rho - 2\tau_2\tau_3\omega_1\nu\rho^2 - 2\tau_2\tau_3\omega_2\rho - 2\tau_1\tau_3\nu\rho^2\Big), \tag{187b}$$

$$\rho^2\nu = \tau_1 + \tau_2\rho\nu + \tau_3\rho^2\nu, \tag{187c}$$

$$\omega_1\rho^2\nu + \rho\omega_2 = \tau_1\rho\nu + \tau_2 + \tau_3(\omega_1\rho^2\nu + \rho\omega_2), \tag{187d}$$

$$\nu = \sqrt{1 - 2^{-2R_1}}. \tag{187e}$$

*Proof:* According to equation 25, the optimization program of the third step is given as follows

$$\min_{P_{\hat{X}_3|X_3\hat{X}_1\hat{X}_2}} \mathbb{E}[\|X_3 - \hat{X}_3\|^2]$$
$$s.t. \quad I(X_3; \hat{X}_3|\hat{X}_1, \hat{X}_2) \le R_3,$$
$$P_{\hat{X}_1\hat{X}_2 X_3} = P_{\hat{X}_1\hat{X}_2\hat{X}_3}. \tag{188}$$

Using the above program, we first derive the rate expression in equation 187b. Consider the following set of inequalities

$$R_3 \ge I(X_3; \hat{X}_3|\hat{X}_1, \hat{X}_2) \tag{189}$$
$$= h(\hat{X}_3|\hat{X}_1, \hat{X}_2) - h(Z_3) \tag{190}$$
$$= h(\tau_3 X_3 + Z_3|\hat{X}_1, \hat{X}_2) - h(Z_3) \tag{191}$$
$$= \frac{1}{2}\log 2^{-2h(Z_3)}\left(\tau_3^2 2^{2h(X_3|\hat{X}_1,\hat{X}_2)} + 2^{2h(Z_3)}\right) \tag{192}$$
$$= \frac{1}{2}\log 2^{-2h(Z_3)}\left(\tau_3^2 2^{2h(\rho X_2+N_2|\hat{X}_1,\hat{X}_2)} + 2^{2h(Z_3)}\right) \tag{193}$$
$$= \frac{1}{2}\log 2^{-2h(Z_3)}\left(\tau_3^2(\rho^2 2^{2h(X_2|\hat{X}_1,\hat{X}_2)} + 2^{2h(N_2)}) + 2^{2h(Z_3)}\right) \tag{194}$$
$$= \frac{1}{2}\log 2^{-2h(Z_3)}\left(\tau_3^2(\rho^2 2^{2h(X_2|\hat{X}_1,\hat{X}_2)} + (1-\rho^2)\sigma^2) + 2^{2h(Z_3)}\right) \tag{195}$$
$$\ge \frac{1}{2}\log 2^{-2h(Z_3)}\left(\tau_3^2(\rho^2 2^{2h(X_2|\hat{X}_1)}2^{-2R_2} + (1-\rho^2)\sigma^2) + 2^{2h(Z_3)}\right) \tag{196}$$
$$= \frac{1}{2}\log 2^{-2h(Z_3)}\left(\tau_3^2(\rho^2 2^{2h(\rho X_1+N_1|\hat{X}_1)}2^{-2R_2} + (1-\rho^2)\sigma^2) + 2^{2h(Z_3)}\right) \tag{197}$$
$$= \frac{1}{2}\log 2^{-2h(Z_3)}\left(\tau_3^2(\rho^4 2^{-2R_2}2^{2h(X_1|\hat{X}_1)} + \rho^2(1-\rho^2)2^{-2R_2}\sigma^2 + (1-\rho^2)\sigma^2) + 2^{2h(Z_3)}\right) \tag{198}$$
$$\ge \frac{1}{2}\log 2^{-2h(Z_3)}\left(\tau_3^2(\rho^4\sigma^2 2^{-2R_1-2R_2} + \rho^2(1-\rho^2)2^{-2R_2}\sigma^2 + (1-\rho^2)\sigma^2) + 2^{2h(Z_3)}\right), \tag{199}$$

where

- equation 193 follows from equation 7 where $X_3 = \rho X_2 + N_2$;

- equation 194 and equation 198 follow from Entropy Power Inequality (EPI) (see pp. 22 in El Gamal & Kim (2011)) which holds with equality for Gaussian sources;

- equation 196 follows from the rate constraint $I(X_2; \hat{X}_2|\hat{X}_1) \leq R_2$ which yields $h(X_2|\hat{X}_2, \hat{X}_1) \geq h(X_2|\hat{X}_1) - R_2$;

- equation 199 follows from the rate constraint $I(X_1; \hat{X}_1) \leq R_1$ which yields $h(X_1) \geq h(X_1|\hat{X}_1) - R_1$.

Thus, re-arranging the terms in equation 199, we have

$$(\tau_3^2(\rho^2(1-\rho^2)\sigma^2 2^{-2R_2} + (1-\rho^2)\sigma^2))2^{-2R_3} \geq (1-2^{-2R_3})2^{2h(Z_3)} \tag{200}$$

$$= \sigma^2(1-2^{-2R_3})\Big(1 - \tau_1^2 - \tau_2^2 - \tau_3^2 - 2\tau_1\tau_2\omega_1 - 2\tau_1\tau_2\omega_2\rho$$

$$-2\tau_2\tau_3\omega_1\rho^2 - 2\tau_2\tau_3\omega_2\rho - 2\tau_1\tau_3\rho^2\Big). \tag{201}$$

The above constraint can be simplified as follows

$$\tau_3^2\sigma^2(1-\rho^2 2^{-2R_3} + \rho^2(1-\rho^2)2^{-2R_2}2^{-2R_3}) \geq \sigma^2(1-2^{-2R_3})\Big(1 - \tau_1^2 - \tau_2^2 - 2\tau_1\tau_2\omega_1 - 2\tau_1\tau_2\omega_2\rho$$

$$-2\tau_2\tau_3\omega_1\rho^2 - 2\tau_2\tau_3\omega_2\rho - 2\tau_1\tau_3\rho^2\Big), \tag{202}$$

which is the rate expression in equation 187b.

The derivation of the perception constraint in equation 187c is given in the following:

$$\rho^2\nu\sigma^2 = \mathbb{E}[X_3\hat{X}_1] \tag{203}$$

$$= \mathbb{E}[\hat{X}_3\hat{X}_1] \tag{204}$$

$$= \tau_1\sigma^2 + \tau_2\mathbb{E}[\hat{X}_2\hat{X}_1] + \tau_3\mathbb{E}[X_3\hat{X}_1] \tag{205}$$

$$= \tau_1\sigma^2 + \tau_2\mathbb{E}[X_2\hat{X}_1] + \tau_3\rho^2\mathbb{E}[X_1\hat{X}_1] \tag{206}$$

$$= \tau_1\sigma^2 + \tau_2\rho\mathbb{E}[X_1\hat{X}_1] + \tau_3\rho^2\mathbb{E}[X_1\hat{X}_1] \tag{207}$$

$$= \tau_1\sigma^2 + \tau_2\rho\nu\sigma^2 + \tau_3\rho^2\nu\sigma^2, \tag{208}$$

where

- equation 204 follows from 0-PLF-SA, i.e., $P_{\hat{X}_3\hat{X}_2\hat{X}_1} = P_{X_3\hat{X}_2\hat{X}_1}$ which implies that $\mathbb{E}[X_3\hat{X}_1] = \mathbb{E}[\hat{X}_3\hat{X}_1]$ for the Gauss-Markov source model;

- equation 205 follows from equation 156 where we multiply both sides with $\hat{X}_1$ and take an expectation over the distribution;

- equation 206 follows from 0-PLF-SA which implies that $\mathbb{E}[\hat{X}_2\hat{X}_1] = \mathbb{E}[X_2\hat{X}_1]$ and also from equation 7, we have $X_3 = \rho^2 X_1 + \rho N_1 + N_2$ where $(N_1, N_2)$ are independent of $\hat{X}_1$;

- equation 207 follows from equation 7 where $X_2 = \rho X_1 + N_1$ and $N_1$ is independent of $\hat{X}_1$.

Similarly, for derivation of equation 187d, we have

$$\omega_1\rho^2\nu\sigma^2 + \rho\omega_2\sigma^2 = \mathbb{E}[\hat{X}_2 X_3] \tag{209}$$

$$= \mathbb{E}[\hat{X}_2\hat{X}_3] \tag{210}$$

$$= \tau_1\mathbb{E}[\hat{X}_1\hat{X}_2] + \tau_2\sigma^2 + \tau_3\mathbb{E}[X_3\hat{X}_2] \tag{211}$$

$$= \tau_1\mathbb{E}[\hat{X}_1 X_2] + \tau_2\sigma^2 + \tau_3\mathbb{E}[X_3\hat{X}_2] \tag{212}$$

$$= \tau_1\rho\nu\sigma^2 + \tau_2\sigma^2 + \tau_3(\omega_1\rho^2\nu\sigma^2 + \rho\omega_2\sigma^2). \tag{213}$$

The distortion term in equation 187a can be derived as follows

$$\mathbb{E}[\|X_3 - \hat{X}_3\|^2] = \mathbb{E}[X_3^2] + \mathbb{E}[\hat{X}_3^2] - 2\mathbb{E}[X_3\hat{X}_3] \tag{214}$$

$$= 2\sigma^2 - 2\mathbb{E}[X_3\hat{X}_3] \tag{215}$$

$$= 2\sigma^2 - 2(\tau_1\mathbb{E}[\hat{X}_1 X_3] + \tau_2\mathbb{E}[\hat{X}_2 X_3] + \tau_3\sigma^2) \tag{216}$$

$$= 2\sigma^2 - 2(\tau_1\rho^2\mathbb{E}[\hat{X}_1 X_1] + \tau_2\rho\mathbb{E}[\hat{X}_2 X_2] + \tau_3\sigma^2) \tag{217}$$

$$= 2\sigma^2 - 2(\tau_1\rho^2\nu\sigma^2 + \tau_2\rho(\rho\nu\omega_1 + \omega_2)\sigma^2 + \tau_3\sigma^2), \tag{218}$$

where

- equation 215 follows because 0-PLF-SA implies that $P_{X_3} = P_{\hat{X}_3}$;

- equation 216 follows from equation 156 where $X_3 = \tau_1\hat{X}_1 + \tau_2\hat{X}_2 + \tau_3 X_3 + Z_3$;

- equation 217 follows from equation 7;

- equation 218 follows from equation 154 and equation 155.

This concludes the proof. ∎

The solution of the optimization program in Proposition C.3 is plotted in Fig. 12 for some values of the parameters.

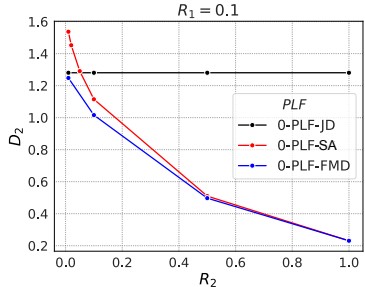 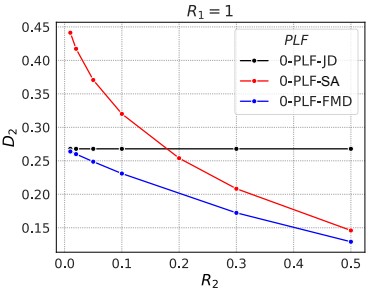

Figure 11: Distortion of the second frame versus its rate for the low-rate regime and $\rho = 1$.

## C.2   0-**PLF-FMD**

In this section, we propose the optimization program of 0-PLF-FMD for the second and third steps. We analytically solve the optimization problem of the second step and provide some numerical evaluations for the program of the third step.

*Second Step:*

The optimization program of the second step is similar to that of Proposition C.3 but with a difference that the condition equation 161c which preserves the joint distribution of $(\hat{X}_1, \hat{X}_2)$ is not needed for 0-PLF-FMD and only marginal distributions are fixed. We also use the following approximation for the rate of the first frame

$$1 - 2^{-2R_1} = 2\epsilon \ln 2 + O(\epsilon^2). \tag{219}$$

Thus, the optimization problem of the second step for 0-PLF-FMD is as follows

$$\min_{\omega_1,\omega_2} \quad 2\sigma^2 - 2\omega_1\rho\sigma^2\sqrt{2\epsilon \ln 2 + O(\epsilon^2)} - 2\omega_2\sigma^2 \tag{220a}$$

$$\text{s.t.} \quad \omega_2^2(1 - \rho^2 2^{-2R_2}(2\epsilon \ln 2 + O(\epsilon^2))) \leq (1 - \omega_1^2 - 2\omega_1\omega_2\rho(2\epsilon \ln 2 + O(\epsilon^2)))(1 - 2^{-2R_2}). \tag{220b}$$

Now, we proceed with solving the above optimization program analytically. Considering dominant terms of equation 220b, this condition reduces to the following

$$\omega_2^2 \leq (1 - \omega_1^2)(1 - 2^{-2R_2}). \tag{221}$$

Thus, the optimization program of equation 220 with considering the dominant terms reduces to the following

$$\min_{\omega_1,\omega_2} \quad 2\sigma^2 - 2\omega_1\rho\sigma^2\sqrt{2\epsilon \ln 2} - 2\omega_2\sigma^2 \tag{222a}$$

$$\text{s.t.} \quad \omega_2^2 \leq (1 - \omega_1^2)(1 - 2^{-2R_2}). \tag{222b}$$

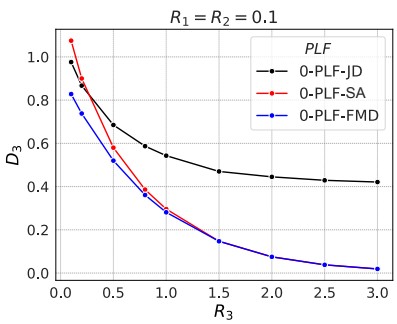 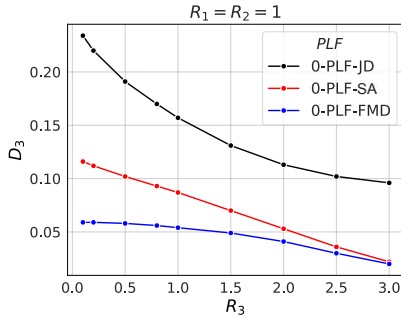

Figure 12: Distortion of the third frame versus its rate for the low-rate regime and $\rho = 1$.

The above program is convex, with a linear objective function in $\omega_1$ and $\omega_2$. Since the feasible region is convex, the solution lies on the boundary, i.e.,

$$\omega_2^2 = (1 - \omega_1^2)(1 - 2^{-2R_2}). \tag{223}$$

Substituting the above into equation 222a, with $\omega_2$ constrained to be nonnegative since a positive value increases the objective function, we obtain

$$\min_{\omega_1} 2\sigma^2(1 - \rho\omega_1\sqrt{2\epsilon\ln 2} - \sqrt{1 - \omega_1^2}\sqrt{1 - 2^{-2R_2}}). \tag{224}$$

Taking the derivative of the above expression with respect to $\omega_1$, we have

$$\frac{\omega_1}{\sqrt{1 - \omega_1^2}}\sqrt{1 - 2^{-2R_2}} = \rho\sqrt{2\epsilon\ln 2}, \tag{225}$$

which yields

$$\omega_1 = \frac{\rho\sqrt{2\epsilon\ln 2}}{\sqrt{1 - 2^{-2R_2} + \rho^2 2\epsilon\ln 2}}, \tag{226}$$

and

$$\omega_2 = \frac{1 - 2^{-2R_2}}{\sqrt{1 - 2^{-2R_2} + \rho^2(2\epsilon\ln 2)}}. \tag{227}$$

Thus, we get

$$\hat{X}_2 = \frac{\rho\sqrt{2\epsilon\ln 2}}{\sqrt{1 - 2^{-2R_2} + \rho^2 2\epsilon\ln 2}}\hat{X}_1 + \frac{1 - 2^{-2R_2}}{\sqrt{1 - 2^{-2R_2} + \rho^2(2\epsilon\ln 2)}}X_2 + Z_2, \tag{228}$$

where $Z_2 \sim \mathcal{N}(0, (1 - \omega_1^2 - \omega_2^2 - 2\rho\nu\omega_1\omega_2)\sigma^2)$ is a Gaussian random variable independent of $(\hat{X}_1, X_2)$, and the optimal distortion is given by

$$D_{2,\text{FMD}}(\epsilon, R_2) := 2\sigma^2(1 - \sqrt{1 - 2^{-2R_2} + \rho^2 2\epsilon\ln 2}) + O(\epsilon). \tag{229}$$

*Third Step:*

The optimization program of the third step for 0 0-PLF-FMD is similar to that of equation 187. The difference is that the conditions in equation 187c and equation 187d, which preserve the joint distributions of $(\hat{X}_1, \hat{X}_2, \hat{X}_3)$, are not required because for 0-PLF-FMD only the marginal distributions are fixed. Thus, we have the following optimization program for the third step

$$\min_{\tau_1,\tau_2,\tau_3} 2\sigma^2 - 2\tau_3\sigma^2 - 2\tau_2\omega_2\rho\sigma^2 - 2\tau_2\omega_1\nu\rho^2\sigma^2 - 2\tau_1\nu\rho^2\sigma^2 \tag{230a}$$

$$\text{s.t.}: \ \tau_3^2\sigma^2(1 - 2^{-2R_3}(\rho^4 2^{-2R_1 - 2R_2} + \rho^2(1 - \rho^2)2^{-2R_2} - \rho^2)) \leq \sigma^2(1 - 2^{-2R_3})\Big(1 - \tau_1^2 - \tau_2^2$$

$$- 2\tau_1\tau_2\omega_1\nu - 2\tau_1\tau_2\omega_2\nu\rho - 2\tau_2\tau_3\omega_1\nu\rho^2 - 2\tau_2\tau_3\omega_2\rho - 2\tau_1\tau_3\nu\rho^2\Big). \tag{230b}$$

The solution of the above optimization program is plotted for some values of parameters in Fig. 12.

### C.3 0-**PLF-JD**

In this section, we propose the optimization programs of 0-PLF-JD for the second and third steps. We analytically solve the optimization problem of the second frame and provide some numerical evaluations for the third step.

*Second Step:*

The optimization program of the second step is similar to that of Proposition C.2 with a difference that the condition in equation 161c is replaced by the corresponding condition of 0-PLF-JD which is $P_{X_1 X_2} = P_{\hat{X}_1 \hat{X}_2}$. This constraint implies that $\mathbb{E}[X_1 X_2] = \mathbb{E}[\hat{X}_1 \hat{X}_2]$ which together with equation 154 and equation 155 yields

$$\omega_1 + \nu \omega_2 \rho = \rho. \tag{231}$$

Thus, the optimization problem of the second step for 0-PLF-JD when $R_1 = \epsilon$ is as follows

$$\min_{\omega_1, \omega_2} \quad 2\sigma^2 - 2\omega_1 \rho \sigma^2 \sqrt{2\epsilon \ln 2 + O(\epsilon^2)} - 2\omega_2 \sigma^2 \tag{232a}$$

$$\text{s.t.} \quad \omega_2^2(1 - \rho^2 2^{-2R_2}(2\epsilon \ln 2 + O(\epsilon^2))) \leq (1 - \omega_1^2 - 2\omega_1 \omega_2 \rho \sqrt{2\epsilon \ln 2 + O(\epsilon^2)})(1 - 2^{-2R_2}) \tag{232b}$$

$$\omega_1 + \nu \omega_2 \rho = \rho. \tag{232c}$$

The constraint equation 232c implies that

$$\omega_1 = \rho - \rho \omega_2 \sqrt{2\epsilon \ln 2} + O(\epsilon). \tag{233}$$

Plugging the above into equation 232a and equation 232b, we get

$$\min_{\omega_2} \quad 2\sigma^2(1 - \rho^2 \sqrt{2\epsilon \ln 2} - \omega_2) + O(\epsilon) \tag{234a}$$

$$\text{s.t.} \omega_2 \leq \sqrt{1 - \rho^2}\sqrt{1 - 2^{-2R_2}} + O(\sqrt{\epsilon}). \tag{234b}$$

The objective function of the above program is decreasing in $\omega_2$, so the solution of the above program is given by

$$\omega_2 = \sqrt{1 - \rho^2}\sqrt{1 - 2^{-2R_2}} + O(\sqrt{\epsilon}). \tag{235}$$

Thus, we have

$$\hat{X}_2 = (\rho - \rho \omega_2 \sqrt{2\epsilon \ln 2})\hat{X}_1 + \sqrt{1 - \rho^2}\sqrt{1 - 2^{-2R_2}} X_2 + Z_2, \tag{236}$$

where $Z_2 \sim \mathcal{N}(0, ((1 - \rho^2)2^{-2R_2} - \rho^2\sqrt{1 - \rho^2}\sqrt{1 - 2^{-2R_2}}\sqrt{2\epsilon \ln 2})\sigma^2)$ is a Gaussian random variable independent of $(\hat{X}_1, X_2)$ and the optimal distortion is given by

$$D_{2,\text{JD}}(\epsilon, R_2) := 2\sigma^2(1 - \sqrt{1 - \rho^2}\sqrt{1 - 2^{-2R_2}} - \rho^2\sqrt{2\epsilon \ln 2}) + O(\epsilon). \tag{237}$$

*Third Step:*

The optimization program of the third step for 0-PLF-JD is similar to equation 187 but with a difference that the conditions in equation 187c and equation 187d are replaced by the corresponding conditions of 0-PLF-JD which is $P_{X_1 X_2 X_3} = P_{\hat{X}_1 \hat{X}_2 \hat{X}_3}$. This constraint implies that

$$\mathbb{E}[X_1 X_3] = \mathbb{E}[\hat{X}_1 \hat{X}_3], \tag{238}$$

$$\mathbb{E}[X_2 X_3] = \mathbb{E}[\hat{X}_2 \hat{X}_3]. \tag{239}$$

Considering equation 154–equation 156 together with the above conditions, we get

$$\rho^2 = \tau_1 + \tau_2 \rho + \tau_3 \rho^2 \nu, \tag{240}$$

$$\rho = \tau_1 \rho + \tau_2 + \tau_3(\omega_1 \rho^2 \nu + \rho \omega_2). \tag{241}$$

Thus, we have the following optimization program for the third step

$$\min_{\tau_1,\tau_2,\tau_3} 2\sigma^2 - 2\tau_3\sigma^2 - 2\tau_2\omega_2\rho\sigma^2 - 2\tau_2\omega_1\nu\rho^2\sigma^2 - 2\tau_1\nu\rho^2\sigma^2 \tag{242a}$$

$$\text{s.t.}: \ \tau_3^2\sigma^2(1 - 2^{-2R_3}(\rho^4 2^{-2R_1-2R_2} + \rho^2(1-\rho^2)2^{-2R_2} - \rho^2)) \le \sigma^2(1 - 2^{-2R_3})\Big(1 - \tau_1^2 - \tau_2^2 - 2\tau_1\tau_2\omega_1\nu$$

$$-2\tau_1\tau_2\omega_2\nu\rho - 2\tau_2\tau_3\omega_1\nu\rho^2 - 2\tau_2\tau_3\omega_2\rho - 2\tau_1\tau_3\nu\rho^2\Big), \tag{242b}$$

$$\rho^2 = \tau_1 + \tau_2\rho + \tau_3\rho^2\nu, \tag{242c}$$

$$\rho = \tau_1\rho + \tau_2 + \tau_3(\omega_1\rho^2\nu + \rho\omega_2). \tag{242d}$$

The solution of the above program is plotted in Fig. 12 for some values of parameters. For the case $R_1 = R_2 = 0.1$ (low compression rates) and a large range of rates $R_3$, the performances of 0-PLF-SA and 0-PLF-FMD are almost the same. For $R_1 = R_2 = 1$ (low compression rates), the distortion of 0-PLF-SA is significantly smaller than that of 0-PLF-JD for all values of $R_3$, and for a large enough $R_3$, it performs similar to 0-PLF-FMD.

## D  High-Rate Regime for the First Frame

In this section, we first prove the following theorem where the first frame is compressed at a high rate, i.e., $R_1 \to \infty$. The rates of all subsequent frames are assumed to be small, i.e., $R_j = \epsilon$ for sufficiently small $\epsilon > 0$ and $j \in \{2, \ldots, T\}$. Then, we provide proofs for the achievable reconstructions of 0-PLF-FMD as outlined in Table 5.

**Theorem D.1** *Let $R_1 \to \infty$ and $R_j = \epsilon$ for sufficiently small $\epsilon > 0$ and $j \in \{2, \ldots, T\}$. An achievable reconstruction under 0-PLF-SA in $j$th frame ($j \in \{1, \ldots, T\}$) is given by*

$$\hat{X}_j = \rho^{j-1}\hat{X}_1 + \sum_{i=1}^{j-1} O(\sqrt{\epsilon})N_i + \sum_{i=2}^{j-2} O(\sqrt{\epsilon})Z_{i,SA} + O(\sqrt{\epsilon})Z_{j-1,SA} + Z_{j,SA}, \tag{243}$$

*where $Z_{j,SA}$ is a Gaussian random noise independent of $(\{N_i\}_{i=1}^{j-1}, \{Z_{i,SA}\}_{i=2}^{j-1})$, with mean zero and variance $(1 - \rho^{2(j-1)} + O(\epsilon))\sigma^2$, and the distortion is as follows*

$$D_{j,SA}(\infty, \underbrace{\epsilon, \ldots, \epsilon}_{j-1}) = 2(1 - \rho^{2(j-1)} - O(\sqrt{\epsilon}))\sigma^2 + O(\epsilon), \tag{244}$$

*and an achievable reconstruction under 0-PLF-JD in $j$th frame is given by*

$$\hat{X}_j = \rho^{j-1}\hat{X}_1 + \sum_{i=1}^{j-1} O(\sqrt{\epsilon})N_i + \sum_{i=2}^{j-2} O(\sqrt{\epsilon})Z_{i,JD} + \rho Z_{j-1,JD} + Z_{j,JD}, \tag{245}$$

*where $Z_{j,JD}$ is a Gaussian random noise independent of $(\{N_i\}_{i=1}^{j-1}, \{Z_{i,JD}\}_{i=2}^{j-1})$ with mean zero and variance given in Section D.2, and the distortion is as follows*

$$D_{j,JD}(\infty, \underbrace{\epsilon, \ldots, \epsilon}_{j-1}) = 2\left(1 - \rho^{2(j-1)} - O(\sqrt{\epsilon})\right)\sigma^2 + O(\epsilon). \tag{246}$$

Before proving the above theorem, we present an important observation based on equation 243 and equation 245. As seen from equation 243, the 0-PLF-SA scheme incorporates a small portion—on the order of $O(\sqrt{\epsilon})$—of the noise from the previous reconstruction $Z_{j-1,SA}$. In contrast, 0-PLF-JD includes a significantly larger portion, denoted by $\rho$, of the previous reconstruction's noise. Particularly at low rates, this noise may be propagated through subsequent reconstructions. This effect becomes more pronounced when a high rate is allocated to the third frame, as illustrated in Table 2.

To prove the above theorem, we consider each PLF separately. We provide the analysis for the second, third and fourth frames. We then use an induction to derive the achievable reconstruction for $j$th frame. Notice that the solutions for the second and third frames are also presented in Table 5.

### D.1   0-**PLF-SA**

In this section, we introduce the optimization programs of the second, third and fourth steps for 0-PLF-SA and provide the solutions for them. The results are further extended to $T$ frames by induction. Similar to equation 155–equation 156, we can generally write the achievable reconstructions of as follows

$$\hat{X}_1 = X_1, \tag{247}$$
$$\hat{X}_2 = \omega_1 \hat{X}_1 + \omega_2 X_2 + Z_{2,\text{SA}}, \tag{248}$$
$$\hat{X}_3 = \tau_1 \hat{X}_1 + \tau_2 \hat{X}_2 + \tau_3 X_3 + Z_{3,\text{SA}}, \tag{249}$$

where $Z_{2,\text{SA}}$ and $Z_{3,\text{SA}}$ are Gaussian random variables independent of $(\hat{X}_1, X_2)$ and $(\hat{X}_1, \hat{X}_2, X_3)$, respectively.

*Second Step:*

The optimization program of the second step for 0-PLF-SA is similar to that of Proposition C.2 but with a difference that $\nu = 1$ since we have a high compression rate for the first frame. Thus, the optimization program of the second step is as follows

$$\min_{\omega_1, \omega_2} \quad 2\sigma^2 - 2\omega_1 \rho \sigma^2 - 2\omega_2 \sigma^2 \tag{250a}$$
$$\text{s.t.} \quad \omega_2^2(1 - \rho^2 2^{-2R_2}) \le (1 - \omega_1^2 - 2\omega_1\omega_2\rho)(1 - 2^{-2R_2}), \tag{250b}$$
$$\omega_1 + \omega_2 \rho = \rho. \tag{250c}$$

For the second frame, the achievable reconstruction is given as follows (see Table 2 in Salehkalaibar et al. (2023))

$$\hat{X}_2 = (\rho - \rho\sqrt{2\epsilon \ln 2})\hat{X}_1 + \sqrt{2\epsilon \ln 2} X_2 + Z_{2,\text{SA}}, \tag{251}$$

where $Z_{2,\text{SA}} \sim \mathcal{N}(0, (1 - \rho^2 + O(\epsilon))\sigma^2)$ is independent of $(\hat{X}_1, X_2)$ and $\hat{X}_1 = X_1$ and the distortion is given as follows

$$D_{2,\text{SA}}(\infty, \epsilon) := 2(1 - \rho^2 - (1 - \rho^2)\sqrt{2\epsilon \ln 2})\sigma^2. \tag{252}$$

*Third Step:*

The optimization program of the third step is similar to that of Proposition C.3 with $\nu = 1$ due to high rate assumption for the first frame. Thus, we have the following optimization program

$$\min_{\tau_1, \tau_2, \tau_3} 2\sigma^2 - 2\tau_3\sigma^2 - 2\tau_2\omega_2\rho\sigma^2 - 2\tau_2\omega_1\rho^2\sigma^2 - 2\tau_1\rho^2\sigma^2 \tag{253a}$$
$$\text{s.t.}: \quad \tau_3^2(1 - 2^{-2R_3}(\rho^4 2^{-2R_1 - 2R_2} + \rho^2(1 - \rho^2)2^{-2R_2} - \rho^2)) \le (1 - 2^{-2R_3})\Big(1 - \tau_1^2 - \tau_2^2 - 2\tau_1\tau_2\omega_1$$
$$-2\tau_1\tau_2\omega_2\rho - 2\tau_2\tau_3\omega_1\rho^2 - 2\tau_2\tau_3\omega_2\rho - 2\tau_1\tau_3\rho^2\Big), \tag{253b}$$
$$\rho^2 = \tau_1 + \tau_2\rho + \tau_3\rho^2, \tag{253c}$$
$$\omega_1\rho^2 + \rho\omega_2 = \tau_1\rho + \tau_2 + \tau_3(\omega_1\rho^2 + \rho\omega_2). \tag{253d}$$

**Case of $R_3 \to \infty$:** In this case, the solution of the optimization problem is trivially given by $\hat{X}_3 = X_3$ since it satisfies the 0-PLF-SA in the third frame, namely, $P_{\hat{X}_3\hat{X}_2\hat{X}_1} = P_{X_3\hat{X}_2\hat{X}_1}$. This case is shown in Table 2.

**Case of $R_3 = R_2 = \epsilon$:** We provide an approximation for the solution of the program in equation 253 when $R_j = \epsilon$. Consider the following

$$1 - 2^{-2R_j} = 2\epsilon \ln 2 + O(\epsilon^2), \qquad j \in \{2, 3\}. \tag{254}$$

We use this approximation for $R_2$, $R_3$, the fact that $R_1 \to \infty$, and consider the dominant terms of equation 253b to simplify it as follows

$$(1 - \rho^4)\tau_3^2 \le (1 - \tau_1^2 - \tau_2^2)(2\epsilon \ln 2). \tag{255}$$

So, the optimization program in equation 253 simplifies as follows

$$\min_{\tau_1,\tau_2,\tau_3} 2\sigma^2 - 2\tau_3\sigma^2 - 2\tau_2\omega_2\rho\sigma^2 - 2\tau_2\omega_1\rho^2\sigma^2 - 2\tau_1\rho^2\sigma^2 \tag{256a}$$

$$\text{s.t.}: \quad (1-\rho^4)\tau_3^2 \leq (1-\tau_1^2-\tau_2^2)(2\epsilon\ln 2), \tag{256b}$$

$$\rho^2 = \tau_1 + \tau_2\rho + \tau_3\rho^2, \tag{256c}$$

$$\omega_1\rho^2 + \rho\omega_2 = \tau_1\rho + \tau_2 + \tau_3(\omega_1\rho^2 + \rho\omega_2). \tag{256d}$$

The optimization problem described above is convex, and its objective function is linear in $\tau_1, \tau_2, \tau_3$. This implies that the solution lies on the boundary of the feasible region. In particular, the condition in equation 256b holds with equality:

$$(1-\rho^4)\tau_3^2 = (1-\tau_1^2-\tau_2^2)(2\epsilon\ln 2). \tag{257}$$

As will be seen later, the term $1-\tau_1^2-\tau_2^2$ indeed includes a constant term, which ensures that the right-hand side of equation 257 is of order $\Theta(\epsilon\ln 2)$. This implies that $\tau_3 = \Theta(\sqrt{\epsilon\ln 2})$. Combined with equation 256c and equation 256d, this suggests that $\tau_1$, $\tau_2$, and $\tau_3$ can generally be expressed as

$$\tau_1 = K_1 + \delta_1\sqrt{2\epsilon\ln 2}, \tag{258}$$

$$\tau_2 = K_2 + \delta_2\sqrt{2\epsilon\ln 2}, \tag{259}$$

$$\tau_3 = \delta_3\sqrt{2\epsilon\ln 2}, \tag{260}$$

where $K_1$, $K_2$, $\delta_1$, $\delta_2$, and $\delta_3$ are constants to be determined. Substituting equation 258–equation 260 into equation 253c–equation 253d yields the following set of equations:

$$\rho^2 = K_1 + \rho K_2, \tag{261a}$$

$$\rho^3 = K_1\rho + K_2, \tag{261b}$$

$$0 = \delta_1 + \rho\delta_2 + \rho^2\delta_3, \tag{261c}$$

$$-\rho^3 + \rho = \rho\delta_1 + \delta_2 + \rho^3\delta_3. \tag{261d}$$

Note that equation 261a–equation 261b yield

$$K_1 = \rho^2, \tag{262}$$

$$K_2 = 0. \tag{263}$$

These constant terms which correspond to leading terms of $\tau_1$ and $\tau_2$ contribute to the dominant terms in equation 257. Plugging the values of $K_1$ and $K_2$ into equation 257, we obtain the following result

$$\delta_3 = 1. \tag{264}$$

Substituting the above into equation 261c–equation 261d, we get

$$\delta_1 = -2\rho^2, \tag{265}$$

$$\delta_2 = \rho. \tag{266}$$

Thus, considering equation 258–equation 260, equation 262–equation 266 and equation 249, we get the following achievable reconstruction

$$\hat{X}_3 = (\rho^2 - 2\rho^2\sqrt{2\epsilon\ln 2})\hat{X}_1 + \rho\sqrt{2\epsilon\ln 2}\hat{X}_2 + \sqrt{2\epsilon\ln 2}X_3 + Z_{3,\text{SA}}, \tag{267}$$

where $Z_{3,\text{SA}} \sim \mathcal{N}(0, (1-\rho^4+O(\epsilon))\sigma^2)$ and the distortion is given by

$$D_{3,\text{SA}}(\infty, \epsilon, \epsilon) := 2(1 - \rho^4 - (1-\rho^4)\sqrt{2\epsilon\ln 2})\sigma^2. \tag{268}$$

Plugging equation 251 into equation 267 yields the following

$$\hat{X}_3 = (\rho^2 - \rho^2\sqrt{2\epsilon\ln 2})\hat{X}_1 + \sqrt{2\epsilon\ln 2}X_3 + \rho\sqrt{2\epsilon\ln 2}Z_{2,\text{SA}} + Z_{3,\text{SA}}. \tag{269}$$

Table 5: Achievable reconstructions and distortions for $R_1 \to \infty$ and $R_2 = R_3 = \epsilon$.

| | Second Step | Third Step |
|---|---|---|
| 0-PLF-FMD ($\sqrt{\epsilon} \ll \rho < 1$) | $\hat{X}_2 = (1 - O(\epsilon))\hat{X}_1 + O(\epsilon)X_2 + Z_{2,\mathrm{FMD}}$ 
 $Z_{2,\mathrm{FMD}} \sim \mathcal{N}(0, O(\epsilon)\sigma^2)$ 
 $D_{2,\mathrm{FMD}}^\infty = 2(1 - \rho - O(\epsilon))\sigma^2$ Table 2 in Salehkalaibar et al. (2023) | $\hat{X}_3 = (1 - O(\epsilon))\hat{X}_1 + O(\epsilon)X_2 + O(\epsilon)X_3 + Z_{2,\mathrm{FMD}} + Z_{3,\mathrm{FMD}}$ 
 $Z_{3,\mathrm{FMD}} \sim \mathcal{N}(0, O(\epsilon)\sigma^2)$ 
 $D_{3,\mathrm{FMD}}^\infty = 2(1 - \rho^2 - O(\epsilon))\sigma^2$ (Appendix D.3) |
| 0-PLF-FMD ($0 < \rho \ll \sqrt{\epsilon}$) | $\hat{X}_2 = O(\sqrt{\epsilon})X_2 + Z'_{2,\mathrm{FMD}}$ 
 $Z'_{2,\mathrm{FMD}} \sim \mathcal{N}(0, (1 - O(\epsilon))\sigma^2)$ 
 $D_{2,\mathrm{FMD}}^\infty = 2\sigma^2(1 - O(\sqrt{\epsilon}))$ (Appendix D.3) | $\hat{X}_3 = O(\sqrt{\epsilon})X_3 + Z'_{3,\mathrm{FMD}}$ 
 $Z'_{3,\mathrm{FMD}} \sim \mathcal{N}(0, (1 - O(\epsilon))\sigma^2)$ 
 $D_{3,\mathrm{FMD}}^\infty = 2\sigma^2(1 - O(\sqrt{\epsilon}))$ (Appendix D.3) |
| 0-PLF-JD | $\hat{X}_2 = (\rho - O(\sqrt{\epsilon}))\hat{X}_1 + O(\sqrt{\epsilon})X_2 + Z_{2,\mathrm{JD}}$ 
 $Z_{2,\mathrm{JD}} \sim \mathcal{N}(0, (1 - \rho^2 + O(\epsilon))\sigma^2)$ 
 $D_{2,\mathrm{JD}}^\infty = 2\sigma^2(1 - \rho^2 - O(\sqrt{\epsilon}))$ Table 2 in Salehkalaibar et al. (2023) | $\hat{X}_3 = \rho^2\hat{X}_1 + O(\sqrt{\epsilon})N_1 + O(\sqrt{\epsilon})N_2 + \rho Z_{2,\mathrm{JD}} + Z_{3,\mathrm{JD}}$ 
 $Z_{3,\mathrm{JD}} \sim \mathcal{N}(0, (1 - \rho^2 + O(\epsilon))\sigma^2)$ 
 $D_{3,\mathrm{JD}}^\infty = 2\sigma^2(1 - \rho^4 - O(\sqrt{\epsilon}))$ (Appendix D.2) |
| 0-PLF-SA | $\hat{X}_2 = (\rho - O(\sqrt{\epsilon}))\hat{X}_1 + O(\sqrt{\epsilon})X_2 + Z_{2,\mathrm{SA}}$ 
 $Z_{2,\mathrm{SA}} = Z_{2,\mathrm{JD}}$ 
 $D_{2,\mathrm{SA}}^\infty = D_{2,\mathrm{JD}}^\infty$ (Appendix D.1) | $\hat{X}_3 = \rho^2\hat{X}_1 + O(\sqrt{\epsilon})N_1 + O(\sqrt{\epsilon})N_2 + O(\sqrt{\epsilon})Z_{2,\mathrm{SA}} + Z_{3,\mathrm{SA}}$ 
 $Z_{3,\mathrm{SA}} \sim \mathcal{N}(0, (1 - \rho^4 + O(\epsilon))\sigma^2)$ 
 $D_{3,\mathrm{SA}}^\infty = 2\sigma^2(1 - \rho^4 - O(\sqrt{\epsilon}))$ (Appendix D.1) |

Using equation 7, the expression in equation 269 can be written as the following

$$\hat{X}_3 = \rho^2\hat{X}_1 + \rho\sqrt{2\epsilon\ln 2}N_1 + \sqrt{2\epsilon\ln 2}N_2 + \rho\sqrt{2\epsilon\ln 2}Z_{2,\mathrm{SA}} + Z_{3,\mathrm{SA}}. \tag{270}$$

This result is shown in Table 5.

*Fourth Step:* We derive the optimization program of the fourth frame and solve it. For the fourth frame, we write the achievable reconstruction as follows

$$\hat{X}_4 = \lambda_1\hat{X}_1 + \lambda_2\hat{X}_2 + \lambda_3\hat{X}_3 + \lambda_4 X_4 + Z_{4,\mathrm{SA}}, \tag{271}$$

where $Z_{4,\mathrm{SA}}$ is a Gaussian random variable independent of $(\hat{X}_1, \hat{X}_2, \hat{X}_3, X_4)$ with mean zero and its variance will be determined later.

**Proposition D.2** *The optimization program of the fourth step for* 0*-PLF-SA when the first frame has a high compression rate, is given as follows*

$$\min_{\lambda_1, \lambda_2, \lambda_3, \lambda_4} 2\sigma^2(1 - \lambda_4 - \lambda_3\rho\tau_3 - \lambda_3\rho^2\tau_2\omega_2 - \lambda_3\rho^3\tau_2\omega_1 - \lambda_3\rho^3\tau_1 - \lambda_2\rho^3\omega_1 - \lambda_2\rho^2\omega_2 - \lambda_1\rho^3) \tag{272a}$$

$$s.t. : 2^{-2R_4}\sigma^2(\lambda_4^2\rho^6 2^{-2R_3 - 2R_2 - 2R_1} + \lambda_4^2\rho^4 2^{-2R_3 - 2R_2}(1 - \rho^2) + \lambda_4^2\rho^2 2^{-2R_3}(1 - \rho^2) + \lambda_4^2(1 - \rho^2))$$
$$\leq 2^{2h(Z_{4,SA})}(1 - 2^{-2R_4}), \tag{272b}$$

$$\rho^3 = \lambda_1 + \rho\lambda_2 + \rho^2\lambda_3 + \rho^3\lambda_4, \tag{272c}$$

$$\rho^2(\rho\omega_1 + \omega_2) = \rho\lambda_1 + \lambda_2 + \rho(\rho\omega_1 + \omega_2)\lambda_3 + \rho^2(\rho\omega_1 + \omega_2)\lambda_4, \tag{272d}$$

$$\rho(\rho^2\tau_1 + \rho(\rho\omega_1 + \omega_2)\tau_2 + \tau_3) = \rho^2\lambda_1 + \rho(\rho\omega_1 + \omega_2)\lambda_2 + \lambda_3 + \rho(\rho^2\tau_1 + \rho(\rho\omega_1 + \omega_2)\tau_2 + \tau_3)\lambda_4. \tag{272e}$$

*Proof:* An extension of equation 23–equation 25 to the fourth step yields the following optimization program

$$\min_{P_{\hat{X}_4|X_4\hat{X}_1\hat{X}_2\hat{X}_3}} \mathbb{E}[\|X_4 - \hat{X}_4\|^2]$$
$$\text{s.t.} \quad I(X_4; \hat{X}_4|\hat{X}_1, \hat{X}_2, \hat{X}_3) \leq R_4,$$
$$P_{\hat{X}_1\hat{X}_2\hat{X}_3 X_4} = P_{\hat{X}_1\hat{X}_2\hat{X}_3\hat{X}_4}. \tag{273}$$

The perception constraints in equation 272c–equation 272e are derived based on 0-PLF-SA which states that $P_{\hat{X}_4\hat{X}_3\hat{X}_2\hat{X}_1} = P_{X_4\hat{X}_3\hat{X}_2\hat{X}_1}$. This implies that $\mathbb{E}[\hat{X}_4\hat{X}_1] = \mathbb{E}[X_4\hat{X}_1]$, $\mathbb{E}[\hat{X}_4\hat{X}_2] = \mathbb{E}[X_4\hat{X}_2]$ and $\mathbb{E}[\hat{X}_4\hat{X}_3] = \mathbb{E}[X_4\hat{X}_3]$. These constraints together with equation 154–equation 156, equation 271 lead to constraints

equation 272c–equation 272e. To derive the rate constraint, consider the following set of inequalities

$$I(X_4; \hat{X}_4 | \hat{X}_1, \hat{X}_2, \hat{X}_3) \tag{274}$$

$$= h(\hat{X}_4 | \hat{X}_1, \hat{X}_2, \hat{X}_3) - h(Z_{4,\text{SA}}) \tag{275}$$

$$= h(\lambda_4 X_4 + Z_{4,\text{SA}} | \hat{X}_1, \hat{X}_2, \hat{X}_3) - h(Z_{4,\text{SA}}) \tag{276}$$

$$= \frac{1}{2} \log 2^{-2h(Z_{4,\text{SA}})} \left( \lambda_4^2 2^{2h(X_4 | \hat{X}_1, \hat{X}_2, \hat{X}_3)} + 2^{2h(Z_{4,\text{SA}})} \right) \tag{277}$$

$$= \frac{1}{2} \log 2^{-2h(Z_{4,\text{SA}})} \left( \lambda_4^2 \rho^2 2^{2h(X_3 | \hat{X}_1, \hat{X}_2, \hat{X}_3)} + \lambda_4^2 2^{2h(N_3)} + 2^{2h(Z_{4,\text{SA}})} \right) \tag{278}$$

$$\geq \frac{1}{2} \log 2^{-2h(Z_{4,\text{SA}})} \left( \lambda_4^2 \rho^2 2^{-2R_3} 2^{2h(X_3 | \hat{X}_1, \hat{X}_2)} + \lambda_4^2 2^{2h(N_3)} + 2^{2h(Z_{4,\text{SA}})} \right) \tag{279}$$

$$= \frac{1}{2} \log 2^{-2h(Z_{4,\text{SA}})} \left( \lambda_4^2 \rho^4 2^{-2R_3} 2^{2h(X_2 | \hat{X}_1, \hat{X}_2)} + \lambda_4^2 \rho^2 2^{-2R_3} 2^{2h(N_2)} + \lambda_4^2 2^{2h(N_3)} + 2^{2h(Z_{4,\text{SA}})} \right) \tag{280}$$

$$\geq \frac{1}{2} \log 2^{-2h(Z_{4,\text{SA}})} \left( \lambda_4^2 \rho^4 2^{-2R_3 - 2R_2} 2^{2h(X_2 | \hat{X}_1)} + \lambda_4^2 \rho^2 2^{-2R_3} 2^{2h(N_2)} + \lambda_4^2 2^{2h(N_3)} + 2^{2h(Z_{4,\text{SA}})} \right) \tag{281}$$

$$= \frac{1}{2} \log 2^{-2h(Z_{4,\text{SA}})} \left( \lambda_4^2 \rho^6 2^{-2R_3 - 2R_2} 2^{2h(X_1 | \hat{X}_1)} + \lambda_4^2 \rho^4 2^{-2R_3 - 2R_2} 2^{2h(N_1)} + \lambda_4^2 \rho^2 2^{-2R_3} 2^{2h(N_2)} + \lambda_4^2 2^{2h(N_3)} \right.$$
$$\left. + 2^{2h(Z_{4,\text{SA}})} \right) \tag{282}$$

$$\geq \frac{1}{2} \log 2^{-2h(Z_{4,\text{SA}})} \left( \lambda_4^2 \rho^6 2^{-2R_3 - 2R_2 - 2R_1} \sigma^2 + \lambda_4^2 \rho^4 2^{-2R_3 - 2R_2} 2^{2h(N_1)} + \lambda_4^2 \rho^2 2^{-2R_3} 2^{2h(N_2)} + \lambda_4^2 2^{2h(N_3)} \right.$$
$$\left. + 2^{2h(Z_{4,\text{SA}})} \right), \tag{283}$$

where

- equation 277 follows from EPI (see pp. 22 in El Gamal & Kim (2011)) which holds with equality for Gaussian sources;

- equation 279, equation 281 and equation 283 follow from the rate constraints $R_3 \geq I(X_3; \hat{X}_3 | \hat{X}_1, \hat{X}_2)$, $R_2 \geq I(X_2; \hat{X}_2 | \hat{X}_1)$ and $R_1 \geq I(X_1; \hat{X}_1)$, respectively;

- equation 280 and equation 282 follow from equation 7 where $X_3 = \rho X_2 + N_2$ and $X_2 = \rho X_1 + N_1$, respectively, and the fact that EPI holds with equality for Gaussian sources.

Re-arranging the terms in equation 283, we get to the constraint in equation 272b. The objective function in equation 272a is obtained by the expansion of $\mathbb{E}[\|X_4 - \hat{X}_4\|^2]$ using equation 248, equation 249 and equation 271. ∎

Now, we provide the approximate solution of the optimization program in equation 272 when $R_2 = R_3 = R_4 = \epsilon$ for sufficiently small $\epsilon > 0$. Using the following approximation

$$1 - 2^{-2R_j} = 2\epsilon \ln 2 + O(\epsilon^2), \tag{284}$$

and considering the dominant terms of equation 272b, the solution of the optimization program is given by

$$\min_{\lambda_1, \lambda_2, \lambda_3, \lambda_4} 2\sigma^2 (1 - \lambda_4 \sigma^2 - 2\lambda_3 \rho \tau_3 - \lambda_3 \rho^2 \tau_2 \omega_2 - \lambda_3 \rho^3 \tau_2 \omega_1 - \lambda_3 \rho^3 \tau_1 - \lambda_2 \rho^3 \omega_1 - \lambda_2 \rho^2 \omega_2 - \lambda_1 \rho^3) \tag{285a}$$

$$\text{s.t.} : \lambda_4^2 (1 - \rho^6) \leq (1 - \lambda_1^2 - \lambda_2^2 - \lambda_3^2)(2\epsilon \ln 2), \tag{285b}$$

$$\rho^3 = \lambda_1 + \rho \lambda_2 + \rho^2 \lambda_3 + \rho^3 \lambda_4, \tag{285c}$$

$$\rho^2(\rho \omega_1 + \omega_2) = \rho \lambda_1 + \lambda_2 + \rho(\rho \omega_1 + \omega_2)\lambda_3 + \rho^2(\rho \omega_1 + \omega_2)\lambda_4, \tag{285d}$$

$$\rho(\rho^2 \tau_1 + \rho(\rho \omega_1 + \omega_2)\tau_2 + \tau_3) = \rho^2 \lambda_1 + \rho(\rho \omega_1 + \omega_2)\lambda_2 + \lambda_3 + \rho(\rho^2 \tau_1 + \rho(\rho \omega_1 + \omega_2)\tau_2 + \tau_3)\lambda_4. \tag{285e}$$

We now proceed to solve the above program. Since it is a convex optimization problem and the objective function is linear, the solution lies on the boundary of the feasible region. In particular, the constraint in equation 285b is satisfied with equality, i.e.,

$$\lambda_4^2(1 - \rho^6) = (1 - \lambda_1^2 - \lambda_2^2 - \lambda_3^2)(2\epsilon \ln 2). \tag{286}$$

As will be shown later, the expression $1 - \lambda_1^2 - \lambda_2^2 - \lambda_3^2$ contains a constant term that ensures the right-hand side of equation 286 is of order $\Theta(2\epsilon \ln 2)$. Consequently, $\lambda_4 = \Theta(\sqrt{2\epsilon \ln 2})$. Taking this into account along with the conditions in equation 285c–equation 285e, we observe that the parameters $\lambda_j$ can be generally expressed as

$$\lambda_j = K_j + \delta_j \sqrt{2\epsilon \ln 2}, \qquad j \in \{1, 2, 3\}, \tag{287}$$

$$\lambda_4 = \delta_4 \sqrt{2\epsilon \ln 2}, \tag{288}$$

where $K_j$, $\delta_j$ are constants to be determined. We substitute the parameters in equation 287–equation 288 into equation 285c–equation 285e to get the following set of equations

$$\rho^3 = K_1 + \rho K_2 + \rho^2 K_3, \tag{289}$$

$$\rho^4 = \rho K_1 + K_2 + \rho^3 K_3, \tag{290}$$

$$\rho^5 = \rho^2 K_1 + \rho^3 K_2 + K_3. \tag{291}$$

Solving this system yields $K_1 = \rho^3$, $K_2 = K_3 = 0$. Note that the constant terms $\{K_j\}_{j=1}^3$ dominate in equation 286 which leads to

$$\delta_4 = 1. \tag{292}$$

Thus, the optimization problem in equation 285, considering only the dominant terms, reduces to:

$$\min_{\delta_j, j=1:4} 2(1 - \rho^6 - (1 + \rho^5 \delta_3 + \rho^4 \delta_2 + \rho^3 \delta_1)\sqrt{2\epsilon \ln 2})\sigma^2 \tag{293a}$$

$$\text{s.t.} : 0 = \delta_1 + \rho\delta_2 + \rho^2\delta_3 + \rho^3, \tag{293b}$$

$$\rho^2(1 - \rho^2) = \rho\delta_1 + \delta_2 + \rho^3\delta_3 + \rho^4, \tag{293c}$$

$$\rho(1 - \rho^4) = \rho^2\delta_1 + \rho^3\delta_2 + \delta_3 + \rho^5. \tag{293d}$$

Solving the above optimization problem, we get

$$\delta_2 = \rho^2, \qquad \delta_3 = \rho, \qquad \delta_1 = -3\rho^3, \qquad \delta_4 = 1. \tag{294}$$

In summary, considering equation 271 with equation 287–equation 288, equation 292, equation 294, we get the following reconstruction

$$\hat{X}_4 = (\rho^3 - 3\rho^3\sqrt{2\epsilon \ln 2})\hat{X}_1 + \rho^2\sqrt{2\epsilon \ln 2}\hat{X}_2 + \rho\sqrt{2\epsilon \ln 2}\hat{X}_3 + \sqrt{2\epsilon \ln 2}X_4 + Z_{4,\text{SA}}. \tag{295}$$

Plugging equation 251 and equation 267 into the above expression, we get

$$\hat{X}_4 = \rho^3\hat{X}_1 + \rho^2\sqrt{2\epsilon \ln 2}N_1 + \rho\sqrt{2\epsilon \ln 2}N_2 + N_3 + \rho^2\sqrt{2\epsilon \ln 2}Z_{2,\text{SA}} + \rho\sqrt{2\epsilon \ln 2}Z_{3,\text{SA}} + Z_{4,\text{SA}}, \tag{296}$$

where $Z_{4,\text{SA}}$ has variance $(1 - \rho^6 + O(\epsilon))\sigma^2$ and the distortion is given by

$$D_{4,\text{SA}}^\infty = 2(1 - \rho^6 - \sqrt{2\epsilon \ln 2}(1 - \rho^6))\sigma^2 + O(\epsilon). \tag{297}$$

Now, we use an induction to derive the achievable reconstruction of $j$th frame.

*$j$th Step*:

By applying induction and extending the above analysis (covering the second, third, and fourth frames) to the $j$th frame, we obtain the following achievable reconstruction for the $j$th frame

$$\hat{X}_j = \rho^{j-1}\hat{X}_1 + \sqrt{2\epsilon\ln 2}\sum_{i=1}^{j-1}\rho^{j-1-i}N_i + \sqrt{2\epsilon\ln 2}\sum_{i=2}^{j-1}\rho^{j-i}Z_{i,\mathrm{SA}} + Z_{j,\mathrm{SA}}, \tag{298}$$

where $Z_{j,\mathrm{SA}} \sim \mathcal{N}(0,(1 - \rho^{2(j-1)} + O(\epsilon))\sigma^2)$ is a Gaussian random variable independent of $(\hat{X}_1,\{N_i\}_{i=1}^{j-1},\{Z_{i,\mathrm{SA}}\}_{i=2}^{j-1})$ and the distortion is given by

$$D_{j,\mathrm{SA}}^{\infty} = 2\sigma^2(1 - \rho^{2(j-1)} - \sqrt{2\epsilon\ln 2}(1-\rho^2)\sum_{i=1}^{j-1}\rho^{2(j-1-i)}) + O(\epsilon). \tag{299}$$

## D.2   0-**PLF-JD**

*Second Step:* When the first frame is compressed at a high rate, the optimization program in the second step for 0-PLF-JD closely resembles that in equation 250 for 0-PLF-SA, and its solution is given by equation 251.

*Third Step:*

The optimization program of the third step for 0-PLF-JD is similar to equation 253, except that the perception constraints in equation 253c–equation 253d are replaced by the following

$$\rho^2 = \tau_1 + \tau_2\rho + \tau_3\rho^2, \tag{300}$$

$$\rho = \tau_1\rho + \tau_2 + \tau_3(\omega_1\rho^2 + \rho\omega_2). \tag{301}$$

These equations follow from the condition $P_{X_1 X_2 X_3} = P_{\hat{X}_1 \hat{X}_2 \hat{X}_3}$, which implies that the second-order statistics are preserved. In particular, $\mathbb{E}[\hat{X}_1\hat{X}_3] = \mathbb{E}[X_1 X_3] = \rho^2\sigma^2$ and $\mathbb{E}[\hat{X}_2\hat{X}_3] = \mathbb{E}[X_2 X_3] = \rho\sigma^2$. Accordingly, the optimization problem for the third step of 0-PLF-JD, assuming that the first frame is encoded at a high rate, can be written as follows:

$$\min_{\tau_1,\tau_2,\tau_3}\ 2\sigma^2 - 2\tau_3\sigma^2 - 2\tau_2\omega_2\rho\sigma^2 - 2\tau_2\omega_1\rho^2\sigma^2 - 2\tau_1\rho^2\sigma^2 \tag{302a}$$

$$\text{s.t.}:\ \tau_3^2(1 - 2^{-2R_3}(\rho^4 2^{-2R_1-2R_2} + \rho^2(1-\rho^2)2^{-2R_2} - \rho^2)) \leq (1 - 2^{-2R_3})\Big(1 - \tau_1^2 - \tau_2^2$$
$$-2\tau_1\tau_2\omega_1 - 2\tau_1\tau_2\omega_2\rho - 2\tau_2\tau_3\omega_1\rho^2 - 2\tau_2\tau_3\omega_2\rho - 2\tau_1\tau_3\rho^2\Big), \tag{302b}$$

$$\rho^2 = \tau_1 + \tau_2\rho + \tau_3\rho^2, \tag{302c}$$

$$\rho = \tau_1\rho + \tau_2 + \tau_3(\omega_1\rho^2 + \rho\omega_2). \tag{302d}$$

**Case of $R_3 \to \infty$:** In this case, the optimization problem in equation 302 simplifies to the following:

$$\min_{\tau_1,\tau_2,\tau_3}\ 2\sigma^2 - 2\tau_3\sigma^2 - 2\tau_2\omega_2\rho\sigma^2 - 2\tau_2\omega_1\rho^2\sigma^2 - 2\tau_1\rho^2\sigma^2 \tag{303a}$$

$$\text{s.t.}:\ \tau_3^2 \leq 1 - \tau_1^2 - \tau_2^2 - 2\tau_1\tau_2\omega_1 - 2\tau_1\tau_2\omega_2\rho - 2\tau_2\tau_3\omega_1\rho^2 - 2\tau_2\tau_3\omega_2\rho - 2\tau_1\tau_3\rho^2, \tag{303b}$$

$$\rho^2 = \tau_1 + \tau_2\rho + \tau_3\rho^2, \tag{303c}$$

$$\rho = \tau_1\rho + \tau_2 + \tau_3(\omega_1\rho^2 + \rho\omega_2). \tag{303d}$$

The above optimization program is convex and the objective function is linear, so the solution lies on the boundary of feasible region. In particular, we have

$$\tau_3^2 = 1 - \tau_1^2 - \tau_2^2 - 2\tau_1\tau_2\omega_1 - 2\tau_1\tau_2\omega_2\rho - 2\tau_2\tau_3\omega_1\rho^2 - 2\tau_2\tau_3\omega_2\rho - 2\tau_1\tau_3\rho^2, \tag{304}$$

$$\rho^2 = \tau_1 + \tau_2\rho + \tau_3\rho^2, \tag{305}$$

$$\rho = \tau_1\rho + \tau_2 + \tau_3(\omega_1\rho^2 + \rho\omega_2). \tag{306}$$

So, based on the above set of equations and the fact that $\omega_1$ and $\omega_2$ include constant terms as well as terms of order $O(\sqrt{2\epsilon\ln 2})$, $\tau_1$, $\tau_2$ and $\tau_3$ can be expressed in the general form $\tau_1 = K_1 + \delta_1\sqrt{2\epsilon\ln 2}$, $\tau_2 = K_2 + \delta_2\sqrt{2\epsilon\ln 2}$

and $\tau_3 = K_3 + \delta_3\sqrt{2\epsilon \ln 2}$. Substituting these expressions into equation 303c–equation 303d yields the following set of equations

$$\rho^2 = K_1 + \rho K_2 + \rho^2 K_3, \tag{307}$$

$$\rho = \rho K_1 + K_2 + \rho^3 K_3, \tag{308}$$

$$0 = \delta_1 + \rho \delta_2 + \rho^2 \delta_3, \tag{309}$$

$$0 = \rho \delta_1 + \delta_2 + K_3(\rho - \rho^3) + \delta_3 \rho^3. \tag{310}$$

The above set of equations yields the following

$$K_2 = \rho, \tag{311}$$

$$K_1 = -\rho^2 K_3, \tag{312}$$

$$\delta_2 = -\rho K_3. \tag{313}$$

Also, the constraint in equation 304 yields the following for the first-order terms:

$$K_3^2 = 1 - K_1^2 - K_2^2 - 2K_1 K_2 \rho - 2K_2 K_3 \rho^3 - 2K_1 K_3 \rho^2. \tag{314}$$

Substituting the expressions from equation 311 and equation 312 into the equation above, we obtain

$$K_3 = \frac{1}{\sqrt{1 + \rho^2}}, \tag{315}$$

and

$$K_1 = -\frac{\rho^2}{\sqrt{1 + \rho^2}}. \tag{316}$$

Additionally, from equation 313, we have

$$\delta_2 = -\frac{\rho}{\sqrt{1 + \rho^2}}. \tag{317}$$

Since in this optimization problem all first-order coefficients ($K_1$, $K_2$, and $K_3$) are nonzero, the third reconstruction can be expressed as:

$$\hat{X}_3 = (-\frac{\rho^2}{\sqrt{1 + \rho^2}} + O(\sqrt{\epsilon}))\hat{X}_1 + (\rho - O(\sqrt{\epsilon}))\hat{X}_2 + (\frac{1}{\sqrt{1 + \rho^2}} + O(\sqrt{\epsilon}))X_3 + Z_{3,\mathrm{JD}}, \tag{318}$$

where $Z_{3,\mathrm{JD}} \sim \mathcal{N}(0, O(\sqrt{\epsilon})\sigma^2)$. Finally, since $\hat{X}_1 = X_1$ due to the high-rate assumption, the reconstruction can be further simplified as:

$$\hat{X}_3 = (\rho - O(\sqrt{\epsilon}))\hat{X}_2 + \frac{1}{\sqrt{1 + \rho^2}}(\rho N_1 + N_2 + O(\sqrt{\epsilon})) + Z_{3,\mathrm{JD}}. \tag{319}$$

This result is shown in Table 2.

**Case of $R_3 = R_2 = \epsilon$:** Similar to equation 285, we focus on the dominant terms of the constraint in equation 302b and formulate the following optimization problem:

$$\min_{\tau_1, \tau_2, \tau_3} \quad 2\sigma^2 - 2\tau_3 \sigma^2 - 2\tau_2 \omega_2 \rho \sigma^2 - 2\tau_2 \omega_1 \rho^2 \sigma^2 - 2\tau_1 \rho^2 \sigma^2 \tag{320a}$$

$$\text{s.t.}: \quad (1 - \rho^4)\tau_3^2 \le (1 - \tau_1^2 - \tau_2^2)(2\epsilon \ln 2), \tag{320b}$$

$$\rho^2 = \tau_1 + \tau_2 \rho + \tau_3 \rho^2, \tag{320c}$$

$$\rho = \tau_1 \rho + \tau_2 + \tau_3(\omega_1 \rho^2 + \rho \omega_2). \tag{320d}$$

The above program is convex and the objective function is linear, so the solution lies on the boundary of the feasible region. In particular, we have:

$$(1 - \rho^4)\tau_3^2 = (1 - \tau_1^2 - \tau_2^2)(2\epsilon \ln 2). \tag{321}$$

As will be shown later, the expression $1 - \tau_1^2 - \tau_2^2$ includes a constant term that ensures the right-hand side of equation 321 is of the order $\Theta(2\epsilon \ln 2)$. This implies that $\tau_3 = \Theta(\sqrt{2\epsilon \ln 2})$. Together with equation 320c–equation 320d, this suggests that $\tau_1, \tau_2, \tau_3$ can be expressed as $\tau_1 = K_1 + \delta_1 \sqrt{2\epsilon \ln 2}$, $\tau_2 = K_2 + \delta_2 \sqrt{2\epsilon \ln 2}$ and $\tau_3 = \delta_3 \sqrt{2\epsilon \ln 2}$ where $K_j, \delta_j$ are constants to be determined. Substituting these expressions into equation 320c–equation 320d, we obtain the following equations:

$$\rho^2 = K_1 + \rho K_2, \tag{322a}$$

$$\rho = K_1 \rho + K_2, \tag{322b}$$

$$0 = \delta_1 + \rho \delta_2 + \rho^2 \delta_3, \tag{322c}$$

$$0 = \rho \delta_1 + \delta_2 + \rho^3 \delta_3. \tag{322d}$$

Notice that equation 322a and equation 322b yield $K_1 = 0$ and $K_2 = \rho$. Note that the constant terms of $\{\tau_j\}_{j=1}^2$ (i.e., $\{K_j\}_{j=1}^2$) contribute to the dominant terms of the inequality equation 321. As a result, we obtain the following condition

$$\delta_3 = \frac{1}{\sqrt{1 + \rho^2}}. \tag{323}$$

The optimization program in equation 320 can thus be simplified as:

$$\min_{\delta_1, \delta_2, \delta_3} 2\sigma^2 (1 - \rho^4 - (\frac{1}{\sqrt{1 + \rho^2}} + \delta_1 \rho^2 + \delta_2 \rho^3 + \rho^2 - \rho^4) \sqrt{2\epsilon \ln 2}) \tag{324a}$$

$$\text{s.t. } 0 = \delta_1 + \rho \delta_2 + \frac{\rho^2}{\sqrt{1 + \rho^2}}, \tag{324b}$$

$$0 = \rho \delta_1 + \delta_2 + \frac{\rho^3}{\sqrt{1 + \rho^2}}. \tag{324c}$$

Solving the above optimization program, we get

$$\delta_2 = 0, \quad \delta_1 = -\frac{\rho^2}{\sqrt{1 + \rho^2}}. \tag{325}$$

Thus, by substituting the values of $\tau_j$ (expressed in terms of $\delta_j$ and $K_j$ as derived above) into equation 249, we obtain

$$\hat{X}_3 = \rho \hat{X}_2 - \frac{\rho^2}{\sqrt{1 + \rho^2}} \sqrt{2\epsilon \ln 2} \hat{X}_1 + \frac{1}{\sqrt{1 + \rho^2}} \sqrt{2\epsilon \ln 2} X_3 + Z_{3,\text{JD}}, \tag{326}$$

where $Z_{3,\text{JD}} \sim \mathcal{N}(0, (1 - \rho^2 + O(\epsilon))\sigma^2)$ is independent of $(\hat{X}_1, \hat{X}_2, X_3)$. Plugging equation 251 into the above expression yields the following

$$\hat{X}_3 = \left(\rho^2 - (\rho^2 + \frac{\rho^2}{\sqrt{1 + \rho^2}})\sqrt{2\epsilon \ln 2}\right) \hat{X}_1 + \rho \sqrt{2\epsilon \ln 2} X_2 + \frac{\sqrt{2\epsilon \ln 2}}{\sqrt{1 + \rho^2}} X_3 + \rho Z_{2,\text{JD}} + Z_{3,\text{JD}}, \tag{327}$$

where the distortion is given as follows

$$D_{3,\text{JD}}(\infty, \epsilon, \epsilon) := 2(1 - \rho^4 - (1 - \rho^2)(\rho^2 + \sqrt{1 + \rho^2})\sqrt{2\epsilon \ln 2})\sigma^2 + O(\epsilon). \tag{328}$$

Using equation 7, equation 327 can be further simplified as follows

$$\hat{X}_3 = \rho^2 \hat{X}_1 + \left(\rho + \frac{\rho}{\sqrt{1 + \rho^2}}\right)\sqrt{2\epsilon \ln 2} N_1 + \frac{1}{\sqrt{1 + \rho^2}}\sqrt{2\epsilon \ln 2} N_2 + \rho Z_{2,\text{JD}} + Z_{3,\text{JD}}. \tag{329}$$

This case is shown in Table 5.

*Fourth Step:*

The optimization program for the fourth step in the case of 0-PLF-JD closely resembles the one described in Proposition D.2, except that conditions equation 285c–equation 285e are replaced by the corresponding constraints for 0-PLF-JD, given by

$$\mathbb{E}[\hat{X}_4\hat{X}_3] = \mathbb{E}[X_4\hat{X}_3], \ \mathbb{E}[\hat{X}_4\hat{X}_2] = \mathbb{E}[X_4\hat{X}_2], \ \mathbb{E}[\hat{X}_4\hat{X}_1] = \mathbb{E}[X_4\hat{X}_1]. \tag{330}$$

Using expressions equation 155–equation 156 and equation 271, these conditions can be further simplified as follows:

$$\rho^3 = \lambda_1 + \rho\lambda_2 + \rho^2\lambda_3 + \rho^3\lambda_4, \tag{331}$$
$$\rho^2 = \rho\lambda_1 + \lambda_2 + \rho\lambda_3 + \rho^2(\rho\omega_1 + \omega_2)\lambda_4, \tag{332}$$
$$\rho = \rho^2\lambda_1 + \rho\lambda_2 + \lambda_3 + \rho(\rho^2\tau_1 + \rho(\rho\omega_1 + \omega_2)\tau_2 + \tau_3)\lambda_4. \tag{333}$$

Therefore, focusing on the dominant terms, the optimization problem for the fourth step of 0-PLF-JD, assuming the first frame is compressed at a high rate, is given by

$$\min_{\lambda_1,\lambda_2,\lambda_3,\lambda_4} 2\sigma^2(1 - \lambda_4 - \lambda_3\rho\tau_3 - \lambda_3\rho^2\tau_2\omega_2 - \lambda_3\rho^3\tau_2\omega_1 - \lambda_3\rho^3\tau_1 - \lambda_2\rho^3\omega_1 - \lambda_2\rho^2\omega_2 - \lambda_1\rho^3) \tag{334a}$$

$$\text{s.t.} \lambda_4^2(1 - \rho^6) \leq (1 - \lambda_1^2 - \lambda_2^2 - \lambda_3^2 + O(\epsilon))(2\epsilon\ln 2), \tag{334b}$$
$$\rho^3 = \lambda_1 + \rho\lambda_2 + \rho^2\lambda_3 + \rho^3\lambda_4, \tag{334c}$$
$$\rho^2 = \rho\lambda_1 + \lambda_2 + \rho\lambda_3 + \rho^2(\rho\omega_1 + \omega_2)\lambda_4, \tag{334d}$$
$$\rho = \rho^2\lambda_1 + \rho\lambda_2 + \lambda_3 + \rho(\rho^2\tau_1 + \rho(\rho\omega_1 + \omega_2)\tau_2 + \tau_3)\lambda_4. \tag{334e}$$

The above program is convex and the objective function is linear, so the solution lies on the boundary of the feasible region. Particularly, equation 334b holds with equality, i.e.,

$$\lambda_4^2(1 - \rho^6) = (1 - \lambda_1^2 - \lambda_2^2 - \lambda_3^2 + O(\epsilon))(2\epsilon\ln 2), \tag{335}$$

which implies that $\lambda_4 = \Theta(\sqrt{2\epsilon\ln 2})$. In fact, as will be confirmed later, the expression $1 - \lambda_1^2 - \lambda_2^2 - \lambda_3^2$ includes a constant term that ensures the right-hand side of the above condition is of order $\Theta(\sqrt{2\epsilon\ln 2})$, reinforcing that $\lambda_4 = \Theta(\sqrt{2\epsilon\ln 2})$. Considering this fact together with conditions equation 334c–equation 334e, we can generally write $\lambda_j = K_j + \delta_j\sqrt{2\epsilon\ln 2}$ for $j \in \{1, 2, 3\}$ and $\lambda_4 = \delta_4\sqrt{2\epsilon\ln 2}$ where $K_j, \delta_j$ are constants to be determined. Plugging these expressions into equation 334c–equation 334e, we get the following set of equations

$$\rho^3 = K_1 + \rho K_2 + \rho^2 K_3, \tag{336}$$
$$\rho^2 = \rho K_1 + K_2 + \rho K_3, \tag{337}$$
$$\rho = \rho^2 K_1 + \rho K_2 + K_3, \tag{338}$$
$$0 = \delta_1 + \rho\delta_2 + \rho^2\delta_3 + \rho^3\delta_4, \tag{339}$$
$$0 = \rho\delta_1 + \delta_2 + \rho\delta_3 + \rho^4\delta_4, \tag{340}$$
$$0 = \rho^2\delta_1 + \rho\delta_2 + \delta_3 + \rho^5\delta_4. \tag{341}$$

From the first three equations, we obtain $K_1 = K_2 = 0$, $K_3 = \rho$. Since the constant terms $K_j$ contribute to the dominant part of equation 335, it simplifies to

$$\delta_4 = \sqrt{\frac{1 - \rho^2}{1 - \rho^6}}. \tag{342}$$

The optimization program in equation 334 further reduces to the following

$$\min_{\delta_1,\delta_2,\delta_3,\delta_4} 2(1 - \rho^6 - (\sqrt{\frac{1-\rho^2}{1-\rho^6}} + \delta_1\rho^3 + \delta_2\rho^4 + \delta_3\rho^5 + \rho^2 - \rho^6)\sqrt{2\epsilon\ln 2})\sigma^2 \tag{343a}$$

$$\text{s.t.} \quad 0 = \delta_1 + \rho\delta_2 + \rho^2\delta_3 + \rho^3\sqrt{\frac{1-\rho^2}{1-\rho^6}}, \tag{343b}$$

$$0 = \rho\delta_1 + \delta_2 + \rho\delta_3 + \rho^4\sqrt{\frac{1-\rho^2}{1-\rho^6}}, \tag{343c}$$

$$0 = \rho^2\delta_1 + \rho\delta_2 + \delta_3 + \rho^5\sqrt{\frac{1-\rho^2}{1-\rho^6}}. \tag{343d}$$

Solving the above optimization program, we get

$$\delta_1 = -\rho^3\sqrt{\frac{1-\rho^2}{1-\rho^6}}, \quad \delta_2 = \delta_3 = 0. \tag{344}$$

In summary, by substituting these values of $\lambda_j$ expressed in terms of $K_j$ and $\delta_j$ into equation 271, the achievable reconstruction is

$$\hat{X}_4 = -\rho^3\sqrt{\frac{1-\rho^2}{1-\rho^6}}\sqrt{2\epsilon\ln 2}\hat{X}_1 + \rho\hat{X}_3 + \sqrt{\frac{1-\rho^2}{1-\rho^6}}\sqrt{2\epsilon\ln 2}X_4 + Z_{4,\text{JD}}, \tag{345}$$

where $Z_{4,\text{JD}} \sim \mathcal{N}(0, (1 - \rho^2 + \rho^4 - \rho^6 + O(\epsilon))\sigma^2)$ is a Gaussian random variable independent of $(\hat{X}_1, \hat{X}_3, X_4)$. Now, we plug equation 251 and equation 267 into the above expression and we get

$$\hat{X}_4 = \rho^3\hat{X}_1 + \left(\rho^2 + \rho^2\sqrt{\frac{1-\rho^2}{1-\rho^6}}\right)\sqrt{2\epsilon\ln 2}N_1 + \left(\rho + \rho\sqrt{\frac{1-\rho^2}{1-\rho^6}}\right)\sqrt{2\epsilon\ln 2}N_2 + \sqrt{\frac{1-\rho^2}{1-\rho^6}}\sqrt{2\epsilon\ln 2}N_3$$
$$+\rho^2\sqrt{2\epsilon\ln 2}Z_{2,\text{JD}} + \rho Z_{3,\text{JD}} + Z_{4,\text{JD}}, \tag{346}$$

where the distortion is given by

$$D_{4,\text{JD}}^\infty := 2\sigma^2\left(1 - \rho^6 - \sqrt{2\epsilon\ln 2}(1-\rho^2)\left(\sqrt{\frac{1-\rho^6}{1-\rho^2}} + \rho^2 - \rho^6\right)\right) + O(\epsilon). \tag{347}$$

*jth Step:*

Using induction and extension of the above analysis for the $j$-th frame yields the following achievable reconstruction

$$\hat{X}_j = \rho^{j-1}\hat{X}_1 + \sqrt{2\epsilon\ln 2}\left(1 + \sqrt{\frac{1-\rho^2}{1-\rho^{2(j-1)}}}\right)\sum_{i=1}^{j-2}\rho^{j-1-i}N_i + \sqrt{\frac{1-\rho^2}{1-\rho^{2(j-1)}}}\sqrt{2\epsilon\ln 2}N_{j-1}$$
$$+\sqrt{2\epsilon\ln 2}\sum_{i=2}^{j-2}\rho^i Z_{j-i,\text{JD}} + \rho Z_{j-1,\text{JD}} + Z_{j,\text{JD}}, \tag{348}$$

where $Z_{j,\text{JD}}$ is a Gaussian random variable independent of $(\{N_i\}_{i=1}^{j-1}, \{Z_{i,\text{JD}}\}_{i=2}^{j-1})$ with mean zero and the following variance

$$\mathbb{E}[Z_{j,\text{JD}}^2] = \begin{cases} ((1-\rho^2)\sum_{i=0}^{\frac{j}{2}-1}\rho^{4i} + O(\epsilon))\sigma^2 & \text{if } j \text{ is even,} \\ ((1-\rho^2)\sum_{i=0}^{\frac{j-1}{2}-1}\rho^{4i} + O(\epsilon))\sigma^2 & \text{if } j \text{ is odd,} \end{cases} \tag{349}$$

and the distortion is given by

$$D_{j,\text{JD}}(\infty, \epsilon, \ldots, \epsilon) := 2\sigma^2 \left( 1 - \rho^{2(j-1)} - \sqrt{2\epsilon \ln 2}(1 - \rho^2) \left( \sqrt{\frac{1 - \rho^{2(j-1)}}{1 - \rho^2}} + \sum_{i=1}^{j-2} \rho^{2(j-1-i)} \right) \right) + O(\epsilon). \tag{350}$$

### D.3   0-**PLF-FMD**

In this section, we provide the optimization programs for the second and third steps of 0-PLF-FMD and solve them. These results were presented in the first and second rows of Table 5. Recall that for the Gauss-Markov source model, the reconstructions exploit the structure in equation 154–equation 156.

*Second Step:*

For the second step, similarly to equation 155, the achievable reconstruction can be expressed as

$$\hat{X}_2 = \omega_1 \hat{X}_1 + \omega_2 X_2 + Z_{2,\text{FMD}}, \tag{351}$$

where $Z_{2,\text{FMD}}$ is independent of $(\hat{X}_1, X_2)$ and notice that $\hat{X}_1 = X_1$ since we have high compression rate for the first frame. The optimization program of the second step is similar to that of Proposition C.2, but with $\nu = 1$ and the perception constraint in equation 161c, which ensures preservation of the joint distribution of $(\hat{X}_1, \hat{X}_2)$ is removed. Instead, only the marginal distribution is constrained. Consequently, the optimization program for the second step of 0-PLF-FMD is given by

$$\min_{\omega_1, \omega_2} \quad 2\sigma^2 - 2\omega_1 \rho \sigma^2 - 2\omega_2 \sigma^2 \tag{352a}$$

$$\text{s.t. } \omega_2^2(1 - \rho^2 2^{-2R_2}) \leq (1 - \omega_1^2 - 2\omega_1 \omega_2 \rho)(1 - 2^{-2R_2}). \tag{352b}$$

The solution of the above program when $R_2 = \epsilon$ (for a sufficiently small $\epsilon$) is given by (see Table 2 in Salehkalaibar et al. (2023))

$$\hat{X}_2 = (1 - \frac{(1 + \rho^2)2\epsilon \ln 2}{2\rho^2})\hat{X}_1 + \frac{2\epsilon \ln 2}{\rho}X_2 + Z_{2,\text{FMD}}, \tag{353}$$

where $Z_{2,\text{FMD}} \sim \mathcal{N}(0, (\frac{1-\rho^2}{\rho^2})2\sigma^2 \epsilon \ln 2)$ is independent of $(\hat{X}_1, X_2)$.

Notice that when $\rho = \Theta(\sqrt{\epsilon})$, the term $\frac{(1+\rho^2)2\epsilon \ln 2}{2\rho^2}$ becomes a constant. In this case, the approximation in equation 353 is not valid anymore. This case should be handled separately as follows.

**Case of** $0 < \rho \ll \sqrt{\epsilon}$**:** In this case, considering the dominant terms of equation 352, this program reduces to the following

$$\min_{\omega_1, \omega_2} \quad 2\sigma^2 - 2\omega_2 \sigma^2 \tag{354a}$$

$$\text{s.t.} \quad \omega_2^2 \leq (1 - \omega_1^2)(2\epsilon \ln 2). \tag{354b}$$

The solution of the above program is as follows

$$\omega_1 = 0, \tag{355}$$

$$\omega_2 = \sqrt{2\epsilon \ln 2}. \tag{356}$$

Thus, the reconstruction of the second step can be written as follows

$$\hat{X}_2 = \sqrt{2\epsilon \ln 2}X_2 + Z'_{2,\text{FMD}}, \tag{357}$$

where $Z'_{2,\text{FMD}} \sim \mathcal{N}(0, (1 - 2\epsilon \ln 2)\sigma^2)$ is independent of $X_2$.

*Third Step:*

For the third step, similar to equation 156, we write the achievable reconstruction as

$$\hat{X}_3 = \tau_1 \hat{X}_1 + \tau_2 \hat{X}_2 + \tau_3 X_3 + Z_{3,\text{FMD}}, \tag{358}$$

where $Z_{3,\text{FMD}}$ is a Gaussian random variable independent of $(\hat{X}_1, \hat{X}_2, X_3)$. The optimization program of the third step is similar to that of Proposition C.3 but with $\nu = 1$ and when the constraints in equation 187c and equation 187d which preserve the joint distribution of $P_{\hat{X}_1 \hat{X}_2 \hat{X}_3}$ are removed and only the marginal distributions are fixed. Thus, we get the following optimization program

$$\min_{\tau_1, \tau_2, \tau_3} 2\sigma^2 - 2\tau_3 \sigma^2 - 2\tau_2 \omega_2 \rho \sigma^2 - 2\tau_2 \omega_1 \rho^2 \sigma^2 - 2\tau_1 \rho^2 \sigma^2 \tag{359a}$$

$$\text{s.t.:} \quad \tau_3^2 (1 - 2^{-2R_3}(\rho^4 2^{-2R_1 - 2R_2} + \rho^2 (1 - \rho^2) 2^{-2R_2} - \rho^2)) \le (1 - 2^{-2R_3})\Big(1 - \tau_1^2 - \tau_2^2 - 2\tau_1 \tau_2 \omega_1$$
$$- 2\tau_1 \tau_2 \omega_2 \rho - 2\tau_2 \tau_3 \omega_1 \rho^2 - 2\tau_2 \tau_3 \omega_2 \rho - 2\tau_1 \tau_3 \rho^2\Big). \tag{359b}$$

**Case of $R_3 \to \infty$:** In this case, the solution of the optimization is trivially given by $\hat{X}_3 = X_3$ since it satisfies 0-PLF-FMD in the third frame which is $P_{\hat{X}_3} = P_{X_3}$. This case is shown in Table 2.

**Case of $R_3 = R_2 = \epsilon$:** We use the following approximation

$$1 - 2^{-2R_j} = 2\epsilon \ln 2 + O(\epsilon^2), \qquad j \in \{2, 3\}. \tag{360}$$

Thus, considering the dominant terms of the constraint in equation 359b, we have

$$(1 - \tau_1^2 - \tau_2^2 - 2\tau_1 \tau_2 \omega_1 - 2\tau_1 \tau_2 \omega_2 \rho - 2\tau_2 \tau_3 \omega_1 \rho^2 - 2\tau_2 \tau_3 \omega_2 \rho - 2\tau_1 \tau_3 \rho^2)(2\epsilon \ln 2) \ge (1 - \rho^4)\tau_3^2. \tag{361}$$

Thus, for the third frame, we have the following optimization program,

$$\min_{\tau_1, \tau_2, \tau_3} 2\sigma^2 - 2\tau_3 \sigma^2 - 2\tau_2 \omega_2 \rho \sigma^2 - 2\tau_2 \omega_1 \rho^2 \sigma^2 - 2\tau_1 \rho^2 \sigma^2 \tag{362a}$$

$$\text{s.t. } (1 - \tau_1^2 - \tau_2^2 - 2\tau_1 \tau_2 \omega_1 - 2\tau_1 \tau_2 \omega_2 \rho - 2\tau_2 \tau_3 \omega_1 \rho^2 - 2\tau_2 \tau_3 \omega_2 \rho - 2\tau_1 \tau_3 \rho^2)(2\epsilon \ln 2) \ge (1 - \rho^4)\tau_3^2. \tag{362b}$$

The above optimization program is convex and the objective function is linear, the optimal solution occurs when the constraint equation 362b holds with equality, i.e.,

$$(1 - \tau_1^2 - \tau_2^2 - 2\tau_1 \tau_2 \omega_1 - 2\tau_1 \tau_2 \omega_2 \rho - 2\tau_2 \tau_3 \omega_1 \rho^2 - 2\tau_2 \tau_3 \omega_2 \rho - 2\tau_1 \tau_3 \rho^2)(2\epsilon \ln 2) = (1 - \rho^4)\tau_3^2. \tag{363}$$

Note that according to equation 353, $\omega_1$ and $\omega_2$ consists of constant terms as well as terms of order $O(2\epsilon \ln 2)$. The left-hand side of equation 363 is itself of order $O(2\epsilon \ln 2)$. This implies that $\tau_3$ cannot include a constant term. Based on these observations and the structure of equation 363, the dominant components of $\tau_1$ and $\tau_2$ can be expressed in the following general form

$$\tau_1 = K_1 + \delta_1 (2\epsilon \ln 2), \tag{364}$$
$$\tau_2 = K_2 + \delta_2 (2\epsilon \ln 2), \tag{365}$$
$$\tau_3 = \delta_3 (2\epsilon \ln 2), \tag{366}$$

where $K_j$, $\delta_j$ are constants to be determined. Substituting the above expressions into the objective function in equation 362a and the constraint in equation 363, and retaining only the dominant terms, leads to the following optimization program:

$$\min_{\tau_1, \tau_2, \tau_3} 2\sigma^2 (1 - (K_1 + K_2)\rho^2 - 2\epsilon \ln 2(1 + \delta_3 + (\delta_1 + \delta_2)\rho^2 + \frac{K_2}{2}(1 - \rho^2))) \tag{367a}$$

$$\text{s.t. } (1 - (K_1 + K_2)^2 - 2(2\epsilon \ln 2)(K_1 \delta_1 + K_2 \delta_2 + K_1 \delta_3 \rho^2 + \delta_3 K_2 \rho^2 + K_1 K_2 + K_1 \delta_2 + K_2 \delta_1 - K_1 K_2 \frac{1 + \rho^2}{2\rho^2}))$$
$$= (1 - \rho^4)\delta_3^2 (2\epsilon \ln 2). \tag{367b}$$

Note that the constant terms on the left-hand side of equation 367b must be zero, as there are no constant terms on the right-hand side, i.e.,

$$K_1 + K_2 = 1. \tag{368}$$

Furthermore, by equating the terms of order $O(2\epsilon \ln 2)$ on both sides, we obtain the following equation

$$K_1 K_2 (\frac{1}{\rho^2} - 1) - 2\delta_1 - 2\delta_2 = (1 - \rho^4)\delta_3^2 + 2\rho^2 \delta_3. \tag{369}$$

Considering equation 368–equation 369, one can rewrite the program in equation 367 as follows

$$\min_{\delta_3, K_2} 2\sigma^2 (1 - \rho^2 - 2\epsilon \ln 2(1 + (1 - \rho^4)\delta_3 - \frac{1}{2}(1 - \rho^4)\delta_3^2 \rho^2 + \frac{K_2(2 - K_2)(1 - \rho^2)}{2})). \tag{370}$$

The solution to the above program is obtained by setting the gradient to zero, which yields the following

$$K_2 = 1, \tag{371}$$
$$\delta_3 = \frac{1}{\rho^2}. \tag{372}$$

Plugging the above into equation 368–equation 369, we have

$$K_1 = 0, \tag{373}$$
$$\delta_1 + \delta_2 = \frac{1 - \rho^4}{2\rho^4}. \tag{374}$$

Combining equation 371–equation 374 with equation 353 and 351 and retaining only the dominant terms, we obtain

$$\hat{X}_3 = (1 - (1 - \frac{1 - \rho^2}{2\rho^4})2\epsilon \ln 2)\hat{X}_1 + \frac{2\epsilon \ln 2}{\rho}X_2 + \frac{2\epsilon \ln 2}{\rho^2}X_3 + Z_{2,\text{FMD}} + Z_{3,\text{FMD}}, \tag{375}$$

where the distortion is given by

$$D_{3,\text{FMD}}(\infty, \epsilon, \epsilon) = 2\sigma^2 (1 - \rho^2 - 2\epsilon \ln 2(1 + \frac{3 + \rho^2 - 2\rho^4}{2\rho^2})). \tag{376}$$

This case is shown in Table 5.

**Case of** $0 < \rho \ll \sqrt{\epsilon}$: In this case, considering the dominant terms of equation 362, the program reduces to the following:

$$\min_{\tau_1, \tau_2, \tau_3} \quad 2\sigma^2 - 2\tau_3 \sigma^2 \tag{377a}$$
$$\text{s.t. } (1 - \tau_1^2 - \tau_2^2)(2\epsilon \ln 2) \geq \tau_3^2. \tag{377b}$$

The solution of the above program is simply given by

$$\tau_1 = 0, \tag{378}$$
$$\tau_2 = 0, \tag{379}$$
$$\tau_3 = \sqrt{2\epsilon \ln 2}. \tag{380}$$

Thus, the reconstruction is given by

$$\hat{X}_3 = \sqrt{2\epsilon \ln 2} X_3 + Z'_{3,\text{FMD}}, \tag{381}$$

where $Z'_{3,\text{FMD}} \sim \mathcal{N}(0, (1 - 2\epsilon \ln 2)\sigma^2)$ is independent of $X_3$.

The achievable reconstructions derived in this section are summarized in Table 5 for the first three frames.

# E   Experimental Setup Details

As described in Section 5, our experimental setup is based on the one proposed in Salehkalaibar et al. (2023). We briefly describe our setup as follows.

## E.1   Video Compressor

In our experiments, we use both traditional and neural-based video compression methods. As a classical baseline, we adopt the H.264 codec for traditional video compression, with results shown in Fig. 14b. For neural compressors, we use two models to compress frame data: the scale-space flow model (Agustsson et al., 2020b) as presented in Salehkalaibar et al. (2023), and the DCVC-HEM framework. Scale-space flow model is adopted as it allows us to efficiently learn the statistical characteristics of the source distribution without relying on pre-trained optical flow estimators. To control the bit rate, we vary the dimension of the latent representation while keeping the quantization interval fixed at 2. Dithered quantization is applied to simulate the common randomness in our setting, following Zhang et al. (2021). For each frame $X_j$, the encoder-decoder pair is optimized based on the latent representations obtained from previously optimized frames.

## E.2   Loss Function

Our theoretical formulation involves solving a constrained optimization problem, which is intractable in practice due to the complexity and non-convexity introduced by neural networks. To circumvent this, we instead adopt a Lagrangian approximation and optimize the following objective:

$$\min \mathbb{E}[\|X_j - \hat{X}_j\|^2] + \lambda \phi_j(P_{\hat{X}_1 \ldots \hat{X}_{j-1} X_j}, P_{\hat{X}_1 \ldots \hat{X}_{j-1} \hat{X}_j}),$$

where the first term represents the mean squared error (MSE) distortion between the original frame $X_j$ and its reconstruction $\hat{X}_j$, and the second term represents the perceptual constraint in our proposed PLF-SA. It measures the divergence between the true and generated frame distributions conditioned on the previous reconstructions. The hyperparameter $\lambda$ balances the trade-off between distortion and perceptual quality, with larger values imposing a stronger perception constraint. Similar to prior work, we use the WGAN (Gulrajani et al., 2017) to approximate the perception term.

## E.3   Training Details

We adopt two primary training setups for experiments on the UVG and MovingMNIST datasets. For evaluation on the UVG dataset, the neural compressors are trained on $256 \times 256$ patches from the Vimeo-90K dataset (Xue et al., 2019). In the case of MovingMNIST dataset, each encoder-decoder pair is trained directly on digit sequences generated from the dataset itself, with digit trajectories constructed as described in Section E.5. Training on Vimeo-90K requires approximately two days on a single NVIDIA A100 GPU, while MovingMNIST training completes in roughly one day. For each bitrate regime, we first pre-train the model to minimize MMSE loss, followed by fine-tuning with the joint distortion–perception loss, which we found to be more stable and effective than optimizing the full objective end-to-end. For optimization, we use the `RMSProp` optimizer (Graves, 2014) for MovingMNIST experiments, and the `Adam` optimizer (Kingma & Ba, 2017) for training on Vimeo-90K.

As shown in Section 5.1, DCVC-HEM (Li et al., 2022) is adopted for comparisons. For the MovingMNIST dataset, since DCVC-HEM is not specifically trained for low-bitrate scenarios ($R_1 = \epsilon$) on this dataset, we fine-tune the pre-trained DCVC-HEM to ensure a fair comparison. During fine-tuning, the quantization scales are adjusted to $\{q_{\text{I-frame}} = 3.5, q_{\text{P-frame}} = 1.5\}$ to enhance its compression performance under low-bitrate settings. In the high-bitrate scenario ($R_1 = \infty$) on MovingMNIST, where the bitrate for second frame is fixed at $R_2 = 2$ bits across all PLF models, achieving this bitrate with DCVC-HEM is challenging. To address this, we directly input the second-frame reconstruction results from 0-PLF-SA into DCVC-HEM to produce the reconstruction of the third frame. For UVG dataset, we use the pre-trained DCVC-HEM

---

**Algorithm 1** *Random Trajectory* sequence generation.

---

1: **inputs:** maximum step size $S$, sequence length $N$, frame size $F$, digit size $D$.
2: $sequence \leftarrow [\,]$
3: $(x, y) \sim U(0, F - D)$
4: $sequence[1] \leftarrow gen\_frame((x, y))$
5: **for** $frame \in \{2, ..., N\}$ **do**
6: $\quad (d_x, d_y) \sim U(-S, S)$
7: $\quad$ **if** $(y < 0)$ **then**
8: $\quad\quad y \leftarrow 0$
9: $\quad\quad d_y \sim U(0, S)$
10: $\quad$ **else if** $(y > F - D)$ **then**
11: $\quad\quad y \leftarrow F - D$
12: $\quad\quad d_y \sim U(-S, 0)$
13: $\quad$ **end if**
14: $\quad$ **if** $(x < 0)$ **then**
15: $\quad\quad x \leftarrow 0$
16: $\quad\quad d_x \sim U(0, S)$
17: $\quad$ **else if** $(x > F - D)$ **then**
18: $\quad\quad x \leftarrow F - D$
19: $\quad\quad d_x \sim U(-S, 0)$
20: $\quad$ **end if**
21: $\quad (x, y) \leftarrow (x, y) + (d_x, d_y)$
22: $\quad sequence[frame] \leftarrow gen\_frame((x, y))$
23: **end for**
24: **return** $sequence$

---

checkpoint without additional fine-tuning. All results presented in Section 5 ensure that the average per-frame bitrate of DCVC-HEM is slightly greater than or equal to the bitrate settings of the proposed PLF-SA models.

### E.4 Perceptual Quality Evaluations

To evaluate the perceptual quality of different compressors, we first use the widely adopted LPIPS metric (Zhang et al., 2018), computed as

$$d(X_i, \hat{X}_i) = \frac{1}{HW} \sum_{h,w} \|f(X_i)_{h,w} - f(\hat{X}_i)_{h,w}\|_2^2,$$

where $f(\cdot)$ denotes a pretrained, fixed deep network used to extract spatial features. For the UVG dataset, we follow standard setting and use an ImageNet-pretrained VGG network as the feature extractor. For the MovingMNIST dataset, the domain gap between ImageNet and MNIST datasets causes the VGG net to overlook meaningful feature differences, even when digit identities change. To address this, we train two 4-layer convolutional networks on the second and third frames of MovingMNIST, each for 10 epochs using the `Adam` optimizer. Once trained, these models serve as feature extractors and we compute LPIPS metric between embeddings of source and reconstructed frames on MovingMNIST.

In addition to LPIPS, we also adopt Fréchet Video Distance (FVD) and Kernel Video Distance (KVD) to evaluate the perceptual quality. FVD (Unterthiner et al., 2019) is a video-level extension of the well-known Fréchet Inception Distance (FID), which models the distribution of entire video sequences in a learned feature space and compares the real and generated video distributions via the Fréchet distance. KVD (Unterthiner et al., 2019) is a variant of FVD which uses the learned features of I3D network to compute distance instead of traditional polynomial kernel. For both metrics, we use the I3D network pretrained on Kinetics-400 as the feature extractor, following established best practices in generative video modeling.

---

**Algorithm 2** *Consistent Trajectory* sequence generation.
---
1: **inputs:** maximum step size $S$, sequence length $N$, frame size $F$, digit size $D$.
2: $sequence \leftarrow [\,]$
3: $(x, y) \sim U(0, F - D)$
4: $(d_x, d_y) \sim U(-S, S)$
5: **for** $frame \in \{1, ..., N\}$ **do**
6: $\quad (x, y) \leftarrow (x, y) + (d_x, d_y)$
7: $\quad sequence[frame] \leftarrow gen\_frame((x, y))$
8: **end for**
9: **return** $sequence$

---

Table 6: Perceptual loss comparisons on the UVG dataset. First frame is compressed at low-rate $R_1 = 0.144$ bpp and second/third frames are compressed at high-rate $R_2 = 4.632$ bpp, $R_3 = 4.632$ bpp.

|  | DCVC-HEM | PLF-JD | PLF-SA | PLF-FMD |
|---|---|---|---|---|
| FVD ↓ | 158.99 | 330.09 | 26.57 | 30.86 |
| KVD ↓ | 64.97 | 147.33 | 13.34 | 13.70 |

### E.5 MovingMNIST Digit Trajectory

This subsection describes the algorithms developed to generate digit trajectories for the MovingMNIST experiments. Section 5 addresses the two main rate regimes discussed in our work. First, we describe our *Random Trajectory* algorithm, utilized when the first frame $X_1$ is encoded with a low rate (Section 4.1). Following that, we discuss *Consistent Trajectory* algorithm, applied to experiments where the first frame $X_1$ is encoded with a high rate (Section 4.2).

Algorithm 1 describes how *Random Trajectory* generates a MovingMNIST sequence. The required inputs are the maximum step size $S$, sequence length $N$, frame size $F$, and digit size $D$. We first sample the initial digit position $(x, y)$ from a uniform distribution $U(0, F - D)$, generating frame $X_1$ by placing the digit in the sampled initial position (lines $3 - 4$). For the subsequent frames $X_2, \ldots, X_N$, we check if the moving digit has reached the frame boundaries (lines $7, 10, 14, 17$). We then sample the vertical and horizontal shifts $(d_x, d_y)$ accordingly (lines $6, 9, 12, 15, 19$). The shift is then applied to the current position $(x, y)$, and the frame is generated by placing the digit in the updated position (lines $21 - 22$). This conditional sampling strategy guarantees that the digit "bounces" in the opposite direction if the margins are reached, keeping the digit always in-frame. In Section 5, we utilize $S = 5$, $N = 3$, $F = 64$, and $D = 32$ for the regime with a low rate at the first frame (Section 4.1).

Algorithm 2 displays the *Constant Trajectory* MovingMNIST sequence generation. Given the same set of inputs as Algorithm 1, we sample a starting position $(x, y) \sim U(0, F - D)$ and a spatial frame-wise shift $(d_x, d_y)$ (lines $3 - 4$). For every frame, the same pair $(d_x, d_y)$ is applied to the current $(x, y)$ position to generate the next frame (lines $5 - 7$). The conditional sampling strategy is not utilized, with digits possibly reaching and crossing the frame boundaries. Utilizing the same shift $(d_x, d_y)$ across frames and not applying any direction changes close to the frame edges provide a frame-wise consistent trajectory across the whole sequence. This characteristic enables the trajectory analysis conducted in Section 5 (Fig. 6) for the rate regime with $X_1$ encoded with a high rate (Section 4.2). We utilize sequence length $N = 3$, frame size $F = 64$, and digit size $D = 32$. For sharp movements (Fig. 6a), we have a maximum step size $S = 20$. For slow movements (Fig. 6b), we utilize maximum step size $S = 5$.

## F   Supplementary Results

In this section, We display additional results to the experimental discussion in Section 5.

First, in Tab. 6, we present a quantitative comparison of perceptual quality on the UVG dataset. All models are evaluated under the same rate setting: the first frame is compressed at a low rate ($R_1 = 0.144$ bpp), while

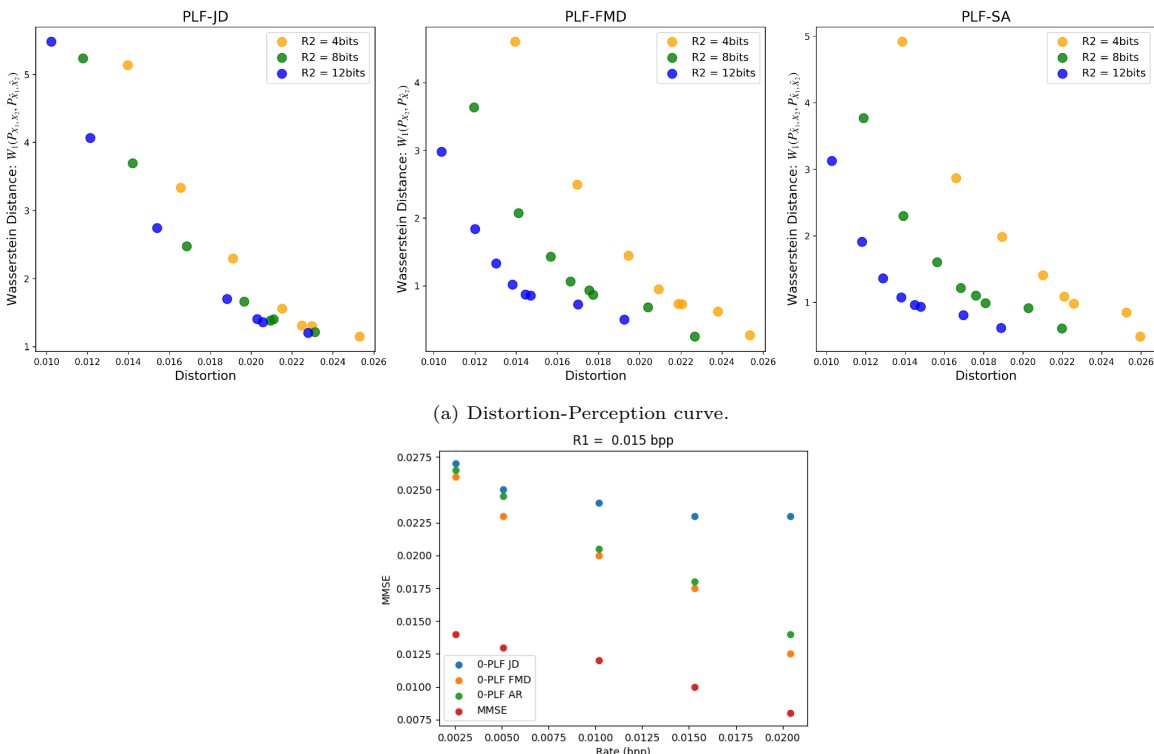

(a) Distortion-Perception curve.

(b) Rate-Distortion curve.

Figure 13: (a) Distortion-Perception curve and (b) Rate-Distortion curve for second frame on the MovingMNIST dataset with different perception loss functions. First frame is compressed at low-rate $R_1 = 12$ bits.

the second and third frames are encoded at higher rates ($R_2 = R_3 = 4.632$ bpp). We report two standard perceptual quality metrics: Fréchet Video Distance (FVD) and Kernel Video Distance (KVD). Among all models, our proposed PLF-SA achieves the best performance, yielding the lowest FVD (26.57) and KVD (13.34), outperforming all other models. PLF-FMD follows closely, while PLF-JD and DCVC-HEM exhibit considerably higher scores, indicating weaker perceptual consistency due to low rate setting. These results highlight the strength of PLF-SA in preserving perceptual quality under low rate allocations in realistic video data.

In Fig. 13, we analyze the impact of different perceptual loss functions on both the distortion-perception and rate-distortion trade-offs for the second frame on the MovingMNIST dataset. The first frame is encoded at a low rate of $R_1 = 12$ bits. Specifically, Fig. 13a shows the distortion-perception curve across different bitrate levels ($R_2 = 4, 8, 12$ bits). PLF-SA and PLF-FMD demonstrate comparable performance, maintaining a favorable balance between distortion and perception. In contrast, PLF-JD consistently underperforms, exhibiting higher Wasserstein distances for the same distortion levels. Fig. 13b presents the rate-distortion curves for each model. As expected, the MMSE baseline achieves the lowest distortion at every bitrate, as it is trained purely with a rate-distortion objective without any perception constraint. Among the perception models, PLF-FMD achieves the best rate-distortion trade-off, with PLF-SA closely following. PLF-JD again performs the worst. These findings validate the effectiveness of PLF-SA in adapting to rate-distortion-perception trade-offs.

Furthermore, additional reconstruction results are provided in Fig. 14, Fig. 15 and Fig. 16. In Fig. 14, we include comparisons with the traditional codec H.264. Fig. 14a shows the reconstruction results on the UVG dataset. It highlights that PLF-JD suffers from a noticeable color tone shift across frames, indicating the error permanence issue under low bitrate. In contrast, both PLF-SA and PLF-FMD maintain stable color fidelity. Fig. 14b depicts results on the MovingMNIST dataset under the same setting as in Fig.6. While H.264 preserves motion trajectories of digits in both low and high temporal correlation cases, it fails to reconstruct clear digit shapes compared to PLF-based methods. Fig. 15 presents additional results for the 4-frame architecture with a slightly different bitrate setting, showing that PLF-JD exhibits direction errors and PLF-SA correcting them, while PLF-FMD fails to reconstruct correct digits. Finally, Fig.16 presents more reconstruction results on the UVG dataset under the same low bitrate setting as in Fig. 14a ($R_1 = 0.144$

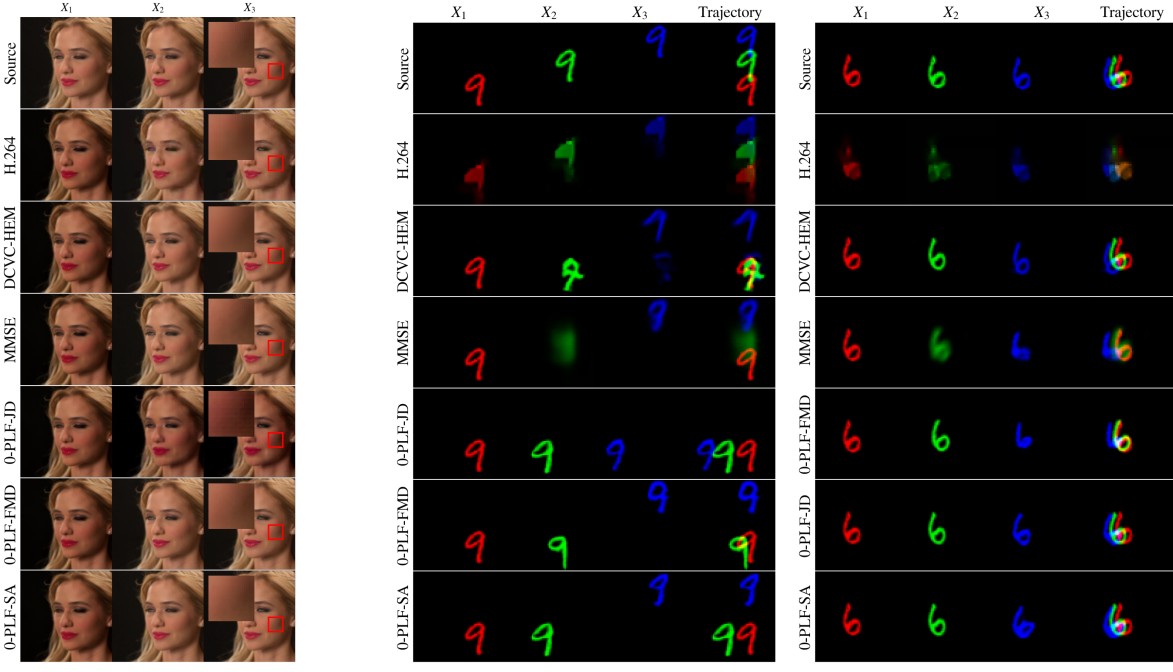

Figure 14: (a) The reconstruction results on the UVG dataset with highlighting the color tone changing of PLF-JD. First frame is compressed at $R_1 = 0.144$ bpp. (b) The reconstruction results on the MovingMNIST dataset with added results of H.264 where $R_1 = \infty$ bits, $R_2 = 2$ bits and $R_3 = 12$ bits. Left: Low correlation. Right: High correlation.

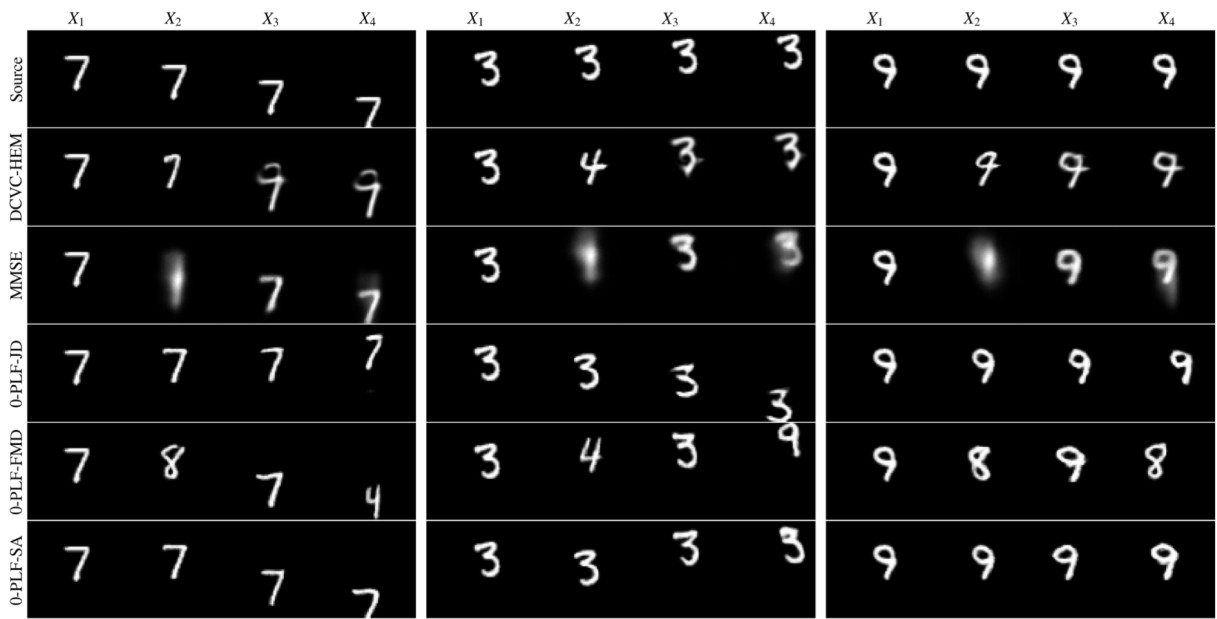

Figure 15: Additional reconstruction results on the MovingMNIST dataset using the $\infty$-$R_2$-$R_3$-$R_4$ 4-frame architecture, with $R_2 = 2$ bits, $R_3 = 12$ bits, and $R_4 = 2$ bits. Consistent with the results observed previously, both PLF-JD and PLF-SA produce direction errors, and PLF-SA is able to correct them. PLF-FMD cannot obviously preserve the temporal correlation as it reconstructs incorrect digits.

bpp, $R_2 = R_3 = 4.632$ bpp). These results further validate the robustness of PLF-SA and PLF-FMD in handling low-rate reference frames, consistent with our earlier findings in Fig. 4.

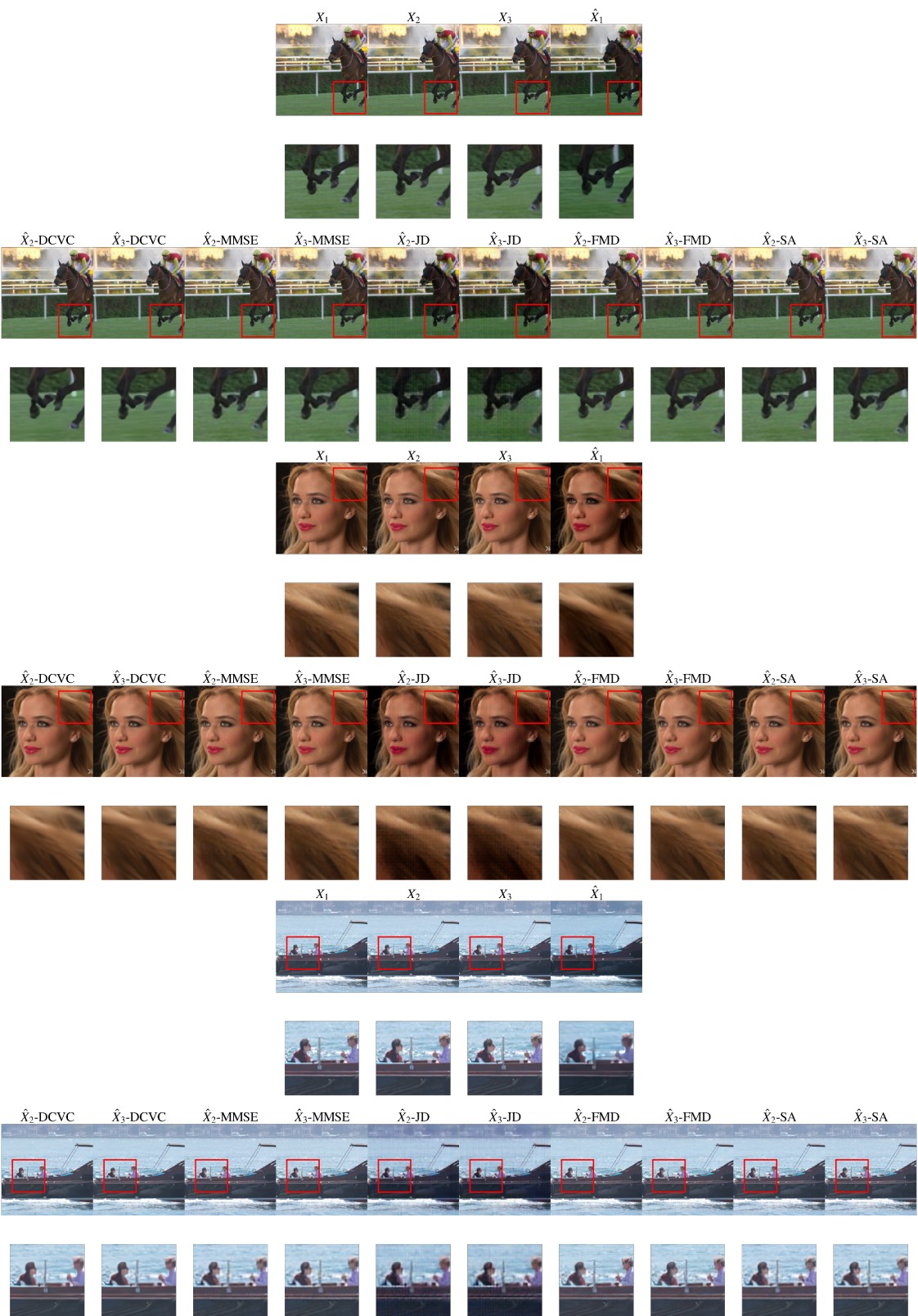

Figure 16: Additional reconstruction results on the UVG dataset. First frame is compressed at low-rate $R_1 = 0.144$ bpp and second/third frame are compressed at high-rate $R_2 = 4.632$ bpp, $R_3 = 4.632$ bpp.

