# OpenReview forum: "On Self-Adaptive Perception Loss Function for Sequential Lossy Compression"
_TMLR — Rejected by TMLR_

### Review · Reviewer_ixyj · 2025-11-11

**Summary Of Contributions:**

This paper studies causal, low-latency lossy video compression system under an MSE distortion measure together with a perceptual loss function (PLF). Building on earlier work that proposed PLF-JD (joint distribution over all frames) and PLF-FMD (frame-wise marginals), the authors introduce a new self-adaptive perception loss function (PLF-SA) that compares the joint distribution between the current source frame and previous reconstructions, rather than previous original frames.

Strengths:
1. The proposed PLF-SA is a simple but well-motivated modification of prior PLFs that conditions on past reconstructions, leading to an intuitively self-adaptive behavior between PLF-FMD and PLF-JD.
2. The information-theoretic analysis is solid. The author formalizes an information rate–distortion–perception (RDP) region for first-order Markov sources under PLF-SA. It shows that, at sufficiently large rates, the operational RDP region can be approximated by this information-theoretic region.

Weaknesses:
1. Salehkalaibar et al. (2023) presented the first systematic study of the properties of PLF-FMD and PLF-JD, and several core ideas and parts of the theoretical analysis are inherited from Salehkalaibar et al. (2023). As a result, the technical novelty of the present work appears limited.
2. I have some concerns about the experimental setting:
a. Experiments are conducted only on a limited set of datasets (1-digit MovingMNIST, KTH, UVG), which is not sufficient to convincingly demonstrate the effectiveness and robustness of the proposed method. Some widely used datasets in learned video compression, such as MCL-JCV, and HEVC Class B, C, D, E datasets, should take into consideration.
b. I am skeptical about the PSNR and LPIPS results in Table 3. The table reports PSNR and LPIPS for each model at a single fixed bitrate, which raises the question of whether the compared models are truly aligned in terms of bitrate. The authors should instead report full RD/RP curves and BD-rate/BD-distortion comparisons to ensure a fair evaluation.
c. The experiments in Sections 5.2.1 and 5.2.2 are conducted only on 3-frame videos. Results on longer sequences would be more reliable and informative.

**Additional Comments:**

see Summary Of Contributions.

**Audience:**

Yes

**Audience Explanation:**

I think this paper intersects information theory, perceptual metrics, and learned video compression, which is highly relevant to a nontrivial subset of TMLR’s readership.

**Broader Impact Concerns:**

No major ethical concerns.

**Claims And Evidence:**

No

**Claims Explanation:**

While the theoretical part of the paper appears clearly presented, I do not think that the overall claims of the submission are supported by sufficiently convincing empirical evidence due to the insufficient experimental results.

**Requested Changes:**

1. Clarify novelty and positioning: Clearly state what is conceptually and technically new about PLF-SA compared with prior PLF work (especially Salehkalaibar et al., 2023), and which results are mainly extensions of existing analyses.

2. Strengthen experiments: Provide more convincing empirical evidence by using additional datasets, reporting fair rate-matched RD/RP curves (e.g., with BD-rate) instead of single rate points.

---

> ### Author Response · Authors · 2025-12-08
>
> We thank the reviewer for acknowledging the strength of the theoretical results. We have added several new experiments, including comparison with LPIPS (Fig. 1), FloLPIPS evaluation (Table 4), fourth-frame results (Fig. 1 and Fig. 7), RD curves for UVG (1080p), and additional results on Vimeo-90K (Figure 5).
>
>
>
> >  Clarify novelty and positioning: Clearly state what is conceptually and technically new about PLF-SA compared with prior PLF work (especially Salehkalaibar et al., 2023), and which results are mainly extensions of existing analyses.
>
> Thanks for the question. We would like to emphasize that the results in this paper are substantially different from those in Salehkalaibar et al., 2023. Below we summarize the main contributions of the paper, highlighting how they are different from prior work.
>
> First, Theorem 3.3, which compares the operational RDP region with the information-theoretic RDP region, requires a new analysis because PLF-SA introduces a fundamentally different perception loss. Since PLF-SA is defined through matching different distributions compared with JD and FMD, we needed to re-establish the inner and outer bounds and verify that the difference constraints are preserved under the relevant Markov conditions.
>
> Next, Theorem 3.4 on the optimality of Gaussian reconstructions for Gaussian sources also needed to be proved from scratch. The reason is that the new perception loss must remain valid while simultaneously respecting the Markov structure and all rate-distortion constraints.
>
> For the low-rate regime in Section 4.1, we derive the optimal distortions for PLF-SA by solving a new optimization problem that is entirely different from those associated with JD or FMD. Notably, PLF-SA does not suffer from the error-propagation phenomenon, a result that is theoretically derived and experimentally validated. Importantly, these optimization problems are solved for steps beyond three frames (see Appendix C), which makes the analysis substantially more involved because each stage depends on the solutions of the previous ones.
>
> In Section 4.2, on temporal preservation, the optimization becomes challenging when the first frame is encoded at a high rate. Again, due to the sequential nature of the problem, each step requires solving a new constrained optimization problem conditioned on the previous results (details in Appendix D).
>
> Finally, the experimental results in Section 5 demonstrate the theoretical findings across multiple datasets, including UVG, Vimeo, KTH, and Moving MNIST, covering a range of rate regimes and different PLFs. These results further confirm the distinct behavior and advantages of PLF-SA compared with JD and FMD.
>
>
> > Comment: Strengthen experiments: Provide more convincing empirical evidence by using additional datasets, reporting fair rate-matched RD/RP curves (e.g., with BD-rate) instead of single rate points
>
> Thank you for the suggestion. Following your recommendation, we now report full rate-PSNR curves on both the UVG-1080p and Vimeo-90K validation sets (Figure 5) to provide a fair, rate-matched comparison. From these curves, we observe that under the different distribution-level perceptual constraints, FMD yields the lowest PSNR, with SA closely matching its behavior, while JD produces the lowest PSNR. These results offer stronger empirical support for our theoretical claims.

---

### Review · Reviewer_ZYxX · 2025-11-13

**Summary Of Contributions:**

The paper investigates perceptual optimization for low-latency sequential video compression and identifies key limitations in the existing rate–distortion–perception framework: PLF-FMD fails to capture temporal structure, whereas PLF-JD suffers from error permanence by propagating early-frame artifacts. To address these issues, the authors propose PLF-SA, a self-adaptive perceptual loss that aligns the current frame’s distribution with that of previously reconstructed frames. Through theoretical analysis on Gauss–Markov sources and experiments on MovingMNIST and UVG, the work demonstrates that PLF-SA can correct early-frame errors at low bitrates while effectively leveraging temporal correlations at higher bitrates, resulting in more coherent and stable reconstructions.

**Audience:**

Yes

**Audience Explanation:**

This paper analyzes perceptual optimization in neural video coding from a theoretical perspective. The insight is directly relevant to understanding and advancing learning systems.

**Broader Impact Concerns:**

No broader impact concerns.

**Claims And Evidence:**

No

**Claims Explanation:**

The experimental validation on temporal consistency relies almost entirely on qualitative visual examples from very small, low-resolution, 3-frame toy datasets, rather than on systematic quantitative evaluations (e.g., temporal metrics, error-decay curves, or large-scale benchmarks).

Furthermore, the comparisons against DCVC-HEM are not methodologically fair, since the baseline uses an MS-SSIM-optimized model while the proposed method uses MSE+GAN training. MSE-optimized DCVC-HEM should be included.

The paper also lacks clarity on bitrate control (e.g., how 0.144 bpp is precisely achieved with SQ+entropy models).

Overall, the empirical evidence provided does not sufficiently or convincingly substantiate the manuscript’s theoretical claims.

**Requested Changes:**

Major:

1. The authors emphasize that the perceptual quality is fundamentally a distribution-matching problem under the RDP framework. While this view is well supported theoretically, it would strengthen the paper to explicitly relate PLF to practical implementations. In particular, several neural codecs [1,2] successfully leverage deterministic perceptual losses such as LPIPS. A direct comparison or discussion of PLF (distribution-level) versus LPIPS-style losses (feature-space matching) would help clarify the practical advantages and trade-offs.
2. The theoretical RDP analysis assumes access to all past original frames $x_1 ,..., x_j$ for the encoder. In contrast, the implemented scale-space flow codec operates causally and only observes previously reconstructed frames $\hat{x}_{j-1}$. This structural gap may influence both the interpretation and practical realization of PLF-SA. An explicit discussion of how this mismatch affects the applicability of the theory would improve the clarity and realism of the claims.
3. The paper states that PLF-SA behaves similarly to PLF-FMD at low $R_1$ and approaches PLF-JD in the high-$R_1$ / low-$R_2$ regime. At present, these statements are supported primarily by qualitative visualizations. Given that the work positions PLF-SA as a theory-driven contribution, it would be valuable to include quantitative evaluations. For example, use motion aware FloLPIPS [3] or other temporal related metrics.
4. Although the theoretical development is general, the experiments use only 3-frame sequences at relatively low resolutions (64×64 or 256×256). This setup may not adequately reflect the long-term behavior of causal video compression, such as “resilience to error permanence” or temporal consistency. Incorporating experiments on longer sequences or higher-resolution datasets would provide a more convincing validation of the proposed properties.
5. The paper frequently references the first-frame bitrate R1=0.144 bpp. With architectures such as scale-space flow or DCVC-HEM, which rely on uniform quantization and arithmetic coding, achieving such a precise bitrate is generally impossible. It would be helpful to clarify whether this value refers to an average measured bitrate or a target bitrate, and how such precision is achieved in practice.
Additionally, more details on the bitrate configuration for PLF-SA / PLF-FMD / PLF-JD would improve clarity.
6. The comparison against DCVC-HEM may not be entirely aligned. The authors use the official MS-SSIM-optimized checkpoint of DCVC-HEM, which is not trained for MSE, while the proposed model includes MSE pre-training followed by GAN/perceptual fine-tuning. Since MS-SSIM-optimized codecs typically trade PSNR for structural quality, evaluating PSNR alone may disadvantage DCVC-HEM. Providing a matched comparison (e.g., MSE-trained DCVC-HEM, or including MS-SSIM) would lead to a more balanced assessment.
7. In figure 4, the $\hat{x}_1$ introduce color deviation, including for DCVC-HEM. It would be helpful to clarify why DCVC-HEM is not provided with its own encoded I-frame, rather than inheriting the color-distorted version.


Citation:

[1] Yang, Ren, Radu Timofte, and Luc Van Gool. "Perceptual Learned Video Compression with Recurrent Conditional GAN." IJCAI. 2022.

[2] Qi, Linfeng, et al. "Generative latent coding for ultra-low bitrate image and video compression." IEEE Transactions on Circuits and Systems for Video Technology (2025).

[3] Danier, Duolikun, Fan Zhang, and David Bull. "FloLPIPS: A bespoke video quality metric for frame interpolation." 2022 Picture Coding Symposium (PCS). IEEE, 2022.

---

> ### Author Response · Authors · 2025-12-08
>
> We thank the reviewer for the detailed feedback and suggestions. We have added several new experiments, including comparison with LPIPS (Fig. 1), FloLPIPS evaluation (Table 4), fourth-frame results (Fig. 1 and Fig. 7), RD curves for UVG (1080p), and an additional dataset, Vimeo-90K (Figure 5).
>
> > comparison of PLF to practical perceptual losses such as LPIPS.
>
> While LPIPS is often used as a perceptual loss, its optimization effect is essentially feature-space distortion minimization and does not explicitly enforce explicit distributional alignment between source and reconstruction. To clarify this distinction, we additionally trained LPIPS-based models on MovingMNIST. The results (now included in Figure 1, “LPIPS” row) show that LPIPS behaves similarly to pixel-MMSE trained and does not provide perceptual improvements in this setting. This supports our claim that PLF (Wasserstein distance) operates at the distribution level as required by the RDP framework, while LPIPS enforces only pointwise feature similarity.
>
> > The theory assumes access to past original frames, while the implementation uses past reconstructions. Clarify this mismatch
>
> To align the implementation as closely as possible with the theory, our implementation trains an independent SSF codec per frame, and we convert the residual-prediction scheme into a conditional coding architecture (via input concatenation). This ensures that during training and inference, each stage is conditioned on previously reconstructed frames. Additionally, we clarify that this fully conditioned architecture is used solely to match the theoretical formulation. In practical implementation, the SSF need not depend on all past frames. One may remove this dependency and impose the distribution constraint only within the PLF loss module.
>
> > Claims about PLF-SA’s behavior rely on qualitative results; quantitative temporal metrics such as FloLPIPS [3] would strengthen the paper.
>
> Thanks for bringing this up. According to this comment, we have added the FloLPIPS metric in Table 4 to have a quantitative evaluation of temporal preservation.
>
> > Experiments use short, low-resolution sequences; longer or higher-resolution results would better validate temporal behavior.
>
> In Figure 6 of the original submission, we already included 4-frame results on MovingMNIST, and we have further updated Figure 1 to show additional 4-frame experiments on different rate regions. These results remain consistent with the theoretical predictions and support the claimed different temporal behavior of different PLFs.
> For higher resolution, we report the full rate-PSNR curves for the second frame on both the UVG-1080 and Vimeo-90K (448×256) validation sets (Figure 5). The performance of PLF-SA remains consistent with the theoretical predictions, even on 1080p real-world datasets.
>
> > Clarify how the precise first-frame bitrate R_1=0.144 is obtained and how rates are configured across PLFs.
>
> Thank you for the question. The reported first-frame bitrate corresponds to the average measured bitrate produced by our offline I-frame compressor. Specifically, to simulate an extremely low-rate first-frame scenario in sequential coding, we used a conditional diffusion image compressor to generate first-frame reconstructions, as diffusion models naturally have a perceptual distribution constraint. Here, 0.144 bpp is the empirical average over all I-frames.
> For the subsequent frames in Figure 1(b) and Figure 4 (UVG results), PLF-SA / FMD / JD all operate under infinite rate of second frame; no compression is applied to align with the R1-Inf-Inf theoretical formulation.
>
>
> > The DCVC-HEM comparison may not be aligned since the official checkpoint is MS-SSIM-optimized, unlike your MSE/GAN-fine-tuned pipeline.
>
>
> Thank you for raising this point. We chose the MS-SSIM optimized DCVC-HEM because MS-SSIM imposes a perceptual constraint, making it more appropriate for evaluating distortion-perception trade-offs. However, as noted in the paper, this model does not generalize well to the extremely low-rate MovingMNIST setting due to the domain gap between its training data and ours. To ensure a relatively fair comparison, we further fine-tuned DCVC-HEM with an MSE loss on MovingMNIST. This produces a more balanced and dataset-consistent comparison.
>
>  > In Fig. 4, why does DCVC-HEM inherit the color-distorted I-frame instead of encoding its own?
>
>
> In Figure 4, we intentionally use the same low-rate I-frame with color deviation for all methods, including DCVC-HEM. This setup is designed to systematically study error permanence under a controlled and comparable first-frame distortion. Traditional neural codecs typically allocate a high bitrate to the I-frame, which makes it difficult to simulate a meaningful first-frame error for DCVC-HEM alone. Using a shared low-rate I-frame allows us to isolate and compare how different models propagate or attenuate this initial error. We will clarify this motivation in the final version.

---

### Review · Reviewer_2Lwb · 2025-11-25

**Summary Of Contributions:**

The paper under review aims to reduce the shortcomings of current perceptual loss metrics for lossy video compression such as PLF-JD (which imposes constraint on joint distribution of reconstructed frames) and PLF-FMD (which imposes contraint on marginal distributions only). These shortcomings have been identified and theoretically analyzed in Salehkalaibar et al, "On the Choice of Perception Loss Function for Learned Video Compression", 2023; the manuscript under review proposes a PLF-SA metric which reduces these issues to a certain degree. The theoretical foundation of the manuscript is generally carried over from Salehkalaibar et al. and focuses on study of Gaussian-Markov sources where rate-distortion-perception tradeoff can be evaluated analytically. Given this theoretical motivation, experiments on a few video datasets are performed and it is demostrated that the PLF-SA metric reduces certain artifacts associated with PLF-JD and PLF-FMD metrics.

The manuscript has a very large overlap from Salehkalaibar et al., particularly the theoretical results and proofs in the Appendix. The manuscript is reasonably well-written even though it has shortcomings, but I haven't been able to identify critical technical faults in the presentation. Certain proofs, particularly in the Appendix, need more clarification as I did not find them fully convincing. For example, on page 17 in the Appendix I see lossless compression of (possibly continuous) random variables $X_{r,1}, ..., X_{r,3}$ with prefix-free code without any mention of e.g. quantization. The article also contains a number of language errors and typos, some of which I mention.

**Audience:**

Yes

**Audience Explanation:**

The topic of optimal choice of perceptual loss functions for video compression is of interest to the community. Previous findings have identified shortcomings of metrics such as PLF-JD and PLF-FMD and the current manuscript attempts to resolve them via PLF-SA.

**Broader Impact Concerns:**

I don't have concerns about ethical implications of the work.

**Claims And Evidence:**

No

**Claims Explanation:**

The manuscript is reasonably well-written even though it has shortcomings, but I haven't been able to identify critical technical faults in the presentation. Certain proofs, particularly in the Appendix, need more clarification as I did not find them fully convincing. For example, on page 17 in the Appendix I see lossless compression of (possibly continuous) random variables $X_{r,1}, ..., X_{r,3}$ with prefix-free code without any mention of e.g. quantization. Other concerns are listed under Required Changes.

**Requested Changes:**

Critical changes are marked with !!! below.

Page 3: Notation \hat \mathcal X_j not defined before Eq. 2. Should it be just $\mathcal X_j$ instead, since it is just an image?

Page 3: "The reconstruction of each frame j should satisfy a certain distortion" - reconstruction can not satisfy a distortion, since it is a loss but not a constraint

Page 3: "We call this PLF as self-adaptive (SA)", "which is called as self-adaptive zero-perception loss", "be called as perception loss function based" - no need for "as" in any of these here or elsewhere in the manuscript

Page 3: "perception metric capturing the divergence between them. We call this PLF as self-adaptive (SA)" - it is not clear what mathematical restrictions the perception metric should satisfy at this point (in particular, why we call it self-adaptive)?

Page 3: In Eq. 3, the conditions is $\phi_j(..., ...) = 0$, however in Eq. 3 we deduce equality of distributions for all $j=1,...,T$. Mathematically, it is not clear what is going on here. Do we say that if for all $j=1,...,T \ \phi_j(..., ...) = 0$ implies that distributions match?

!!! Page 3: "self-adaptive zero-perception loss (0-PLF-SA)" - it is not clear at this point what the distinction betweel 0-PLF-SA and SA is. The text at times suggests that 0-PLF-SA is in fact a loss function (Page 5, it is said "0-PLF-SA metric") and in other cases we talk about "0-PLF condition".

!!! Page 3: Definition 2.1: RDP tuple not defined, not clear which sets R, D, P belong to. Furthermore, the definition of RDP region seems to belong to Section 3 of the paper.

Page 4: The notation $\mathcal{RDP}^o$ for closure of a set is somewhat uncommon, since in general topology the standard notation for set closure is $\overline{X}$ or sometimes $\mathrm{Cl } X$.

!!! Page 4: The definition of Gauss-Markov sources lacks the condition that N_j are Gaussian.

!!! Page 4: Equation (8): It would appear from this definition already that $W_2^2(P_{\hat X_1, ..., \hat X_{j-1}, X_j}, P_{\hat X_1, ..., \hat X_j}) \geq W_2^2(P_{X_j}, P_{\hat X_j})$, i.e., PLF-SA >= PLF-FMD (by applying push-forwards of projections mappings onto last coordinate which is 1-Lipshitz). I think it would be good to put some statements about the PLF-SA and PLF-FMD & PLF-JD relationship here.

Page 4: Definition 3.1. The random variables X_{r, j} lack any sort of explanation, whereas at least some context is given in Salehkalaibar et al. 2023.

!!! Page 4: Eq. 11: I is missing in the definiton of R_3

Page 5: "then the distribution $P_{\hat X_1 X_2}$ in the 0-PLF-SA metric, enforces independence between ..." - not well written, distribution doesn't enforce independence

!!! Page 5: Eq. 23a, 24a, 25a: How do we now that these minima actually exist, i.e., that the infinum is attained at certain point? I imagine we use the Gauss-Markov model reducing optimization to closed subset of \mathbb R^N, but this needs to be explained and some justification is needed for the fact that the constraints in the definition do not lead to an empty set.

Page 5-6: Corollary 3.5: Proof of Corollary 3.5 should be more explicit.

!!! Page 6: Our main object of interest, for Gauss-Markov sources in partiucalar, is the operational RDP region and not information RDP region. Therefore, I would suggest adding a corollary here expaining what can be said about operational RDP for Gauss-Markov source and how it relates to actual compression task.

!!! Page 7: Table 1, \omega_2 coefficient for 0-PLF-SA includes X_2 for some reason

Page 7: "we discuss that the choice" - we show that

Page 8: "a new perceptual metric that inherits advantages in both PLF-JD and PLF-FMD" - inherets advantages from

!!! Page 8: "the noise $Z_{2,JD}$ introduced in the second step continues to propagate in the third step through the term $\rho Z_{2,JD}$ in the expression for $\hat X_3$." - there is no explicit term $\rho Z_{2, JD}$ in the expression for $\hat X_3$.

Page 8: "high compression rate for the initial frame (R_1 \to \infty)"  - I am a bit confused here. Higher information rate R means lower rate of compression? Please check the usages of terms compression rate and information rate throughout the manuscript.

Page 10: "Furthermore, PSNR and Perceptual metric comparisons on UVG dataset are presented" - perceptual metric

Figure 6: "both PLF-JD and PLF-SA produce direction errors, and PLF-SA is able to correct them" - I would say "but PLF-SA is able to eventually correct them"

!!! Page 12: Conclusions: This discussion misses the weaknesses of PLF-SA as a metric. E.g., some of the tables demostrate that PLF-FMD might give higher PSNR, and in the Appendix it is shown that PLF-FMD achieves the best rate-distortion trade-off

Page 15: "the set of all tuples (R, D, P) which satisfy the following (R, D, P) satisfying" - bad writing

Page 16: "Fix the joint distribution P_{X_r} according to equation 40 which constructs the codebook, given by" - it is not clear what is role of Eq. 40 here, please be more specific.

!!! Page 16: "X_{r,1}, whose distribution needs to be preserved according to equation 41" - it is not clear what is meant here by preserving distribution of X_{r, 1} and what role is played by Eq. 41. Similar remark about X_{r,2} below.

!!! Page 17: "to represent X_{r,1}, losslessly" - I am a bit confused since we are talking about lossless compression of variable X_{r,1} which can in principle be continuous according to the RDP definition, and lossless compression is not possible here without quantization even with finite differential entropy

Page 17: "This shows that R + log(R + 1) + 5 \in R^o(D, P)" - technically, we also use conditions 27-29 here. Also, do we strictly need the closure op in the definition of R^o(D, P) for this conclusion to hold?

Page 18: "the decoding functions of the first and second steps imply that" - decoding functions do not imply that smth is true.

!!! Page 18: "The Markov chain in equation 33 is more restricted comparing to equation 77" - it is not clear why Eq. 77 is compared to Eq. 33 and not Eq.34; after all Eq. 76 and Eq. 33 match. "Moreover, the Markov chain equation 34 does not exist in" - not clear either. This part is very poorly explained.

!!! Page 19: Lemma A.3: repeated argument about "imposing the Markov chain" is not clear. Random variables X_{r,j} have been defined above, so one needs to explain how these definitions are changed to satisfy the complete set of Markov chain conditions.

!!! Page 21: Eq. 105 == Eq. 106: I suspect that we use the fact that everything is jointly Gaussian here since it is not true in general, please elaborate.

!!! Page 22: Eq. 117, 125: Is the proof for rates actually complete? These equations are quite different 19a-19c in Theorem 3.3 that we aim to prove.

Page 26 and onwards: Sometimes h is used to denote entropy instead of H, and it doesn't match the convention of using H for discrete case and h for differential.

Page 26: Eq. 171/172: Not clear why this is true.

Page 28: "which holds which equality" - holds with equality

Page 33: "achievable reconstruction of 0-PLF-SA" - 0-PLF is a condition, not something we reconstruct. So I would say "under 0-PLF" for "for 0-PLF".

Page 47: "For optimization, we use the RMSProp optimizer (Graves, 2014) for MovingMNIST experiments, and the Adam optimizer (Kingma & Ba, 2017) for training on Vimeo-90K." - why was optimizer changed?

---

> ### Author Response · Authors · 2025-12-08
>
> We thank the reviewer for recognizing the quality of the writing and the substantial differences from Salehkalaibar et al., 2023. We revised the draft; new text is shown in blue.
>
> Technical Comments:
>
> > Page 4 (Def. 3.1). Role of $X_{r,j}$​.
>
> A paragraph has been added after Theorem 3.3 explaining the purpose of $X_{r,j}$ in the RDP region.
>
> > Page 5 (Eqs. 23–25): existence of minima.
>
> As shown in Theorem 3.4, for Gauss–Markov sources the optimizers may be restricted to jointly Gaussian test channels, reducing (23)–(25) to a finite-dimensional covariance parametrization with a compact feasible set. Since MSE is continuous, each minimum is attained. We added a clarifying remark after Definition 3.5.
>
> > Corollary 3.5: needs a more explicit proof.
>
> We expanded the proof and explicitly constructed the joint test channel.
>
> > Page 8: missing term  $\rho Z_{2,JD​}$ from the expression of $\hat{X}_3$.
>
> The explicit dependence of $\hat{X}\_3$  on  $Z\_{2,JD}$ is now shown.
>
> > Compression rate terminology.
>
> High compression rate refers to the regime where $R_1​$ becomes very large, rendering the constraint $I(X_1;\hat{X}_1)\leq R_1$ inactive. We have updated the terminology to use “compression rate’’ consistently throughout the draft.
>
> > Conclusions: PLF-SA vs PLF-FMD.
>
> We added the advantages of FMD to the conclusion, while noting that it still struggles with temporal preservation in some rate regimes.
>
> > Page 16-17: role of Eqs. 40-41, distribution of $X_{r,1}, X_{r,2}$
>
> The text has been rewritten to clarify the construction of $P_{\mathsf{X}_r}$​​ and the meaning of preserving the induced distributions.
>
> > Page 17: Lossless representation of $X_{r,1}$.
>
> We thank the reviewer for the observation. We did not intend to suggest lossless compression of a continuous variable. In the SFRL construction, $X_{r,1}$ is mapped to a discrete index based on $(X_1, V_1)$, and only this index is encoded losslessly; the continuous value is not transmitted. We have rewritten the text to remove this ambiguity.
>
> > Page 17: Closure operation and Eqs. 27–29.
>
> Thanks for pointing this out. We now reference Eqs. 27–29 in the conclusion. While closure is not essential for this specific argument, it is required to ensure the region is closed.
>
> > Page 18: incorrect equation references and unclear Markov discussion.
>
>  We corrected the equation references and rewrote the surrounding text to improve clarity and flow.
>
> > Page 19: Lemma A.3: Markov chain argument unclear.
>
> We have rewritten the proof of Lemma A.3 to make the argument clear. The changes are shown in blue.
>
> > Page 21: why Eq. 105 = 106?
>
> This step follows because $\mathbb{E}[X_1| \hat{X}^{\star}_1]$  is a deterministic function of  $ \hat{X}^{\star}_1 $. We have added a justification for this step.
>
> >Page 22: completeness of rate proof.
>
> Eqs. 117 and 125 give lower bounds on the rates. Achievability for the Gaussian case using 19a–19c is shown later in Eqs. 143–148.
>
> > Entropy notation ($H$ vs. $h$).
>
> Corrected throughout: $h$ for differential entropy, $H$ for discrete.
>
> > Page 26: Eqs. 171–172.
>
> Eq. 171 follows from rearranging 170, and 172 from the Gaussian choice in 155. These justifications are now included before 171.
>
> > Optimizer choice.
>
> Thank you for the question. The optimizer choice reflects dataset complexity: MovingMNIST trains well with RMSProp, while Vimeo-90K benefits from Adam’s faster convergence. This is a practical choice and does not affect the method. A brief clarification has been added.
>
> > Page 6: suggestion to discuss operational RDP.
>
> As noted after Theorem 3.3, at sufficiently large rates the operational RDP region approximates the information-theoretic one. This follows directly from the theorem and holds for general sources, not only Gaussian.
>
> > Page 4: Eq. (8) and PLF-SA vs. PLF-FMD/JD.
>
> We appreciate this comment. The inequality indeed shows that SA is more restrictive than FMD. However, its comparison with JD is complicated and depends on the operating rate regime.
>
> Minor Editorial Comments:
>
> - $\hat{\mathcal{X}}_j​$ before Eq. 2: now set to $\mathcal{X}_j$​.
>
> - “Reconstruction should satisfy distortion’’ rephrased.
>
> - Removed unnecessary “as’’.
>
> - Clarified that $\phi_j=0$ for all j implies 0-PLF-SA.
>
> - Unified terminology around “perceptual loss function’’ for 0-PLF-SA.
>
> - Definition 2.1 moved to Section 3; tuple notation clarified.
>
> - Replaced $\mathcal{RDP}^o$ with closure notation $\overline{RDP}^o$.
>
> - Added that $N_j$ are Gaussian in the Gauss-Markov model.
>
> - Missing $I$ in $R_3$​ fixed.
>
> - Sentence “...0-PLF-SA metric, enforces independence” rewritten.
>
> - Table 1 typo in $\omega_2$ fixed.
>
> - “We discuss” → “we show”.
>
> - Applied “Inherits advantages from”.
>
> - “Perceptual Metric’’ → “perceptual loss”.
>
> - Fig. 7 wording updated per suggestion.
>
> - Typo “following (R,D,P) satisfying’’ fixed.
>
> - “Decoding functions imply’’ rephrased.
>
> - “Holds which equality’’ corrected.
>
> - “Achievable reconstruction of 0-PLF-SA’’ corrected to “under 0-PLF’’.

---

### Author Response · Authors · 2025-12-08

We would like to sincerely thank the action editor and reviewers for their careful evaluation of our work and for providing thoughtful and constructive feedback. The comments highlighted areas where additional experiments and further clarification of theoretical points would strengthen the paper. We have revised the draft accordingly, and the main updates are summarized below.

New experimental results:

 -  A comparison with LPIPS-style perceptual losses has been added in Figure 1.

 -  Temporal evaluation using the FloLPIPS metric is now included in Table 4.

 -  More results on fourth-frame reconstructions have been incorporated into Figure 1.

 -  Additional experiments on high-resolution datasets, including the Vimeo-90K validation set and the UVG-1080p dataset, have been added to broaden the empirical evaluation. The corresponding RD curves are presented in Figure 5.

Clarifications of theoretical statements:

 We have expanded several explanations, refined proofs, and added remarks addressing the reviewers’ questions about the RDP formulation, Markov assumptions, minimizer existence, …. All major additions and edits are marked in blue for easy reference.

We greatly appreciate the reviewers’ insights, which have helped improve the clarity, completeness, and empirical support of the paper.

---

### Decision · Action_Editor_g8xK · 2026-01-13

**Recommendation:** Reject

**Additional Comments:**

Reviewers suggest strengthening the evidence for realistic sequential video compression by evaluating substantially longer sequences and more clearly discussing generalization to practical settings. In addition, the authors need to clarify the rate control and fairness of the evaluation protocol.

**Audience:**

Yes

**Audience Explanation:**

Researchers interested in learned image/video compression and diffusion-based generative models may find the ideas of this paper interesting, even though the current evidence is not yet fully convincing.

**Claims And Evidence:**

No

**Claims Explanation:**

The experimental evidence does not sufficiently match the scope of the claims. Although additional experiments were provided in the revision, the evaluated sequence length remains far from realistic video compression settings, which makes it difficult to validate the claimed long-term temporal behavior and error permanence. In addition, the generalization of the proposed method to longer sequences is not empirically demonstrated, and the discussion on this limitation remains insufficient. These issues limit the clarity and strength of the evidence supporting the main claims.

**Resubmission Of Major Revision:**

The authors may consider submitting a major revision at a later time.